# Conditional independence testing under misspecified inductive biases

**Felipe Maia Polo**
Department of Statistics
University of Michigan
`felipemaiapolo@gmail.com`

**Yuekai Sun**
Department of Statistics
University of Michigan
`yuekai@umich.edu`

**Moulinath Banerjee**
Department of Statistics
University of Michigan
`moulib@umich.edu`

## Abstract

Conditional independence (CI) testing is a fundamental and challenging task in modern statistics and machine learning. Many modern methods for CI testing rely on powerful supervised learning methods to learn regression functions or Bayes predictors as an intermediate step; we refer to this class of tests as *regression-based* tests. Although these methods are guaranteed to control Type-I error when the supervised learning methods accurately estimate the regression functions or Bayes predictors of interest, their behavior is less understood when they fail due to misspecified inductive biases; in other words, when the employed models are not flexible enough or when the training algorithm does not induce the desired predictors. Then, we study the performance of regression-based CI tests under misspecified inductive biases. Namely, we propose new approximations or upper bounds for the testing errors of three regression-based tests that depend on misspecification errors. Moreover, we introduce the Rao-Blackwellized Predictor Test (RBPT), a regression-based CI test robust against misspecified inductive biases. Finally, we conduct experiments with artificial and real data, showcasing the usefulness of our theory and methods.

## 1 Introduction

Conditional independence (CI) testing is fundamental in modern statistics and machine learning (ML). Its use has become widespread in several different areas, from (i) causal discovery [12, 24, 34, 11] and (ii) algorithmic fairness [28], to (iii) feature selection/importance [5, 37] and (iv) transfer learning [25]. Due to its growing relevance across different sub-fields of statistics and ML, new testing methods with different natures, from regression to simulation-based tests, are often introduced.

Regression-based CI tests, *i.e.*, tests based on supervised learning methods, have become especially attractive in the past years due to (i) significant advances in supervised learning techniques, (ii) their suitability for high-dimensional problems, and (iii) their simplicity and easy application. However, regression-based tests usually depend on the assumption that we can accurately approximate the regression functions or Bayes predictors of interest, which is hardly true if (i) either the model classes are misspecified or if (ii) the training algorithms do not induce the desired predictors, *i.e.*, if we have misspecified inductive biases. Misspecified inductive biases typically lead to inflated Type-I error rates but also can cause tests to be powerless. Even though these problems can frequently arise in practical situations, more attention should be given to theoretically understanding the effects of misspecification on CI hypothesis testing. Moreover, current regression-based methods are usually not designed to be robust against misspecification errors, making CI testing less reliable. In this work, we study the performance of three major regression-based conditional independence tests under misspecified inductive biases and propose the Rao-Blackwellized Predictor Test (RBPT), which is more robust against misspecification.

37th Conference on Neural Information Processing Systems (NeurIPS 2023).

With more details, our main contributions are:

- We present new robustness results for three relevant regression-based conditional independence tests: (i) Significance Test of Feature Relevance (STFR) [7], (ii) Generalized Covariance Measure (GCM) test [31], and (iii) REgression with Subsequent Independence Test (RESIT) [42, 24, 12]. Namely, we derive approximations or upper bounds for the testing errors that explicitly depend on the level of misspecification.

- We introduce the Rao-Blackwellized Predictor Test (RBPT), a modification of the Significance Test of Feature Relevance (STFR) [7] test that is robust against misspecified inductive biases. In contrast with STFR and previous regression/simulation-based[1] methods, the RBPT does *not* require models to be correctly specified to guarantee Type-I error control. We develop theoretical results about the RBPT, and experiments show that RBPT is robust when controlling Type-I error while maintaining non-trivial power.

## 2   Preliminaries

**Conditional independence testing.** Let $(X, Y, Z)$ be a random vector taking values in $\mathcal{X} \times \mathcal{Y} \times \mathcal{Z} \subseteq \mathbb{R}^{d_X \times d_Y \times d_Z}$ and $\mathcal{P}$ be a fixed family of distributions on the measurable space $(\mathcal{X} \times \mathcal{Y} \times \mathcal{Z}, \mathcal{B})$, where $\mathcal{B} = \mathcal{B}(\mathcal{X} \times \mathcal{Y} \times \mathcal{Z})$ is the Borel $\sigma$-algebra. Let $(X, Y, Z) \sim P$ and assume $P \in \mathcal{P}$. If $\mathcal{P}_0 \subset \mathcal{P}$ is the set of distributions in $\mathcal{P}$ such that $X \perp\!\!\!\perp Y \mid Z$, the problem of conditional independence testing can be expressed in the following way:

$$H_0 : P \in \mathcal{P}_0 \qquad\qquad H_1 : P \in \mathcal{P} \backslash \mathcal{P}_0$$

In this work, we also write $H_0 : X \perp\!\!\!\perp Y \mid Z$ and $H_1 : X \not\!\perp\!\!\!\perp Y \mid Z$. We assume throughout that we have access to a dataset $\mathcal{D}^{(n+m)} = \{(X_i, Y_i, Z_i)\}_{i=1}^{n+m}$ independent and identically distributed (i.i.d.) as $(X, Y, Z)$, where $\mathcal{D}^{(n+m)}$ splits into a test set $\mathcal{D}_{te}^{(n)} = \{(X_i, Y_i, Z_i)\}_{i=1}^{n}$ and a training set $\mathcal{D}_{tr}^{(m)} = \{(X_i, Y_i, Z_i)\}_{i=n+1}^{n+m}$. For convenience, we use the training set to fit models and the test set to conduct hypothesis tests, even though other approaches are possible.

**Misspecified inductive biases in modern statistics and machine learning.** Traditionally, misspecified inductive biases in statistics have been linked to the concept of model misspecification and then strictly related to the chosen model classes. For instance, if the best (Bayes) predictor for $Y$ given $X$, $f^*$, is a non-linear function of $X$, but we use a linear function to predict $Y$, then we say our model is misspecified because $f^*$ is not in the class of linear functions. In modern machine learning and statistics, however, it is known that the training algorithm also plays a crucial role in determining the trained model. For example, it is known that training overparameterized neural networks using stochastic gradient descent bias the models towards functions with good generalization [14, 33]. In addition, D'Amour et al. [9] showed that varying hyperparameter values during training could result in significant differences in the patterns learned by the neural network. The researchers found, for instance, that models with different random initializations exhibit varying levels of out-of-distribution accuracy in predicting skin health conditions for different skin types, indicating that each model learned distinct features from the images. The sensitivity of the trained model concerning different training settings suggests that *even models capable of universal approximation may not accurately estimate the target predictor* if the training biases do not induce the functions we want to learn.

We present a toy experiment to empirically demonstrate how the training algorithm can prevent us from accurately estimating the target predictor even when the model class is correctly specified, leading to invalid significance tests. We work in the context of a high-dimensional (overparameterized) regression with a training set of 250 observations and 500 covariates. We use the Generalized Covariance Model (GCM) test[2] [31] to conduct the CI test. The data are generated as

$$Z \sim N(0, I_{500}), \ X \mid Z \sim N(\beta_X^\top Z, 1), \text{ and } Y \mid X, Z \sim N(\beta_Y^\top Z, 1),$$

where the first five entries of $\beta_X$ are set to 20, and the remaining entries are zero, while the last five entries of $\beta_Y$ are set to 20, and the remaining entries are zero. This results in $X$ and $Y$ being conditionally independent given $Z$ and depending on $Z$ only through a small number of entries. Additionally, $\mathbb{E}[X \mid Z] = \beta_X^\top Z$ and $\mathbb{E}[Y \mid Z] = \beta_Y^\top Z$, indicating that the linear model class is correctly specified. To

---

[1]Simulation-based tests usually rely on estimating conditional distributions.

[2]See Appendix A.3 for more details

perform the GCM test, we use LASSO ($\|\cdot\|_1$ penalization term added to empirical squared error) and the minimum-norm least-squares solution to fit linear models that predict $X$ and $Y$ given $Z$. In this problem, the LASSO fitting approach provides the correct inductive bias since $\beta_X$ and $\beta_Y$ are sparse.

We set the significance level to $\alpha = 10\%$ and estimate the Type-I error rate for 100 different training sets. Figure 1 provides the Type-I error rate empirical distribution and illustrates that, despite using the same model class for both fitting methods, the training algorithm induces an undesired predictor in the minimum-norm case, implying an invalid test most of the time. On the other hand, the LASSO approach has better Type-I error control. In Appendix A, we give a similar example but using the Significance Test of Feature Relevance (STFR) [7].

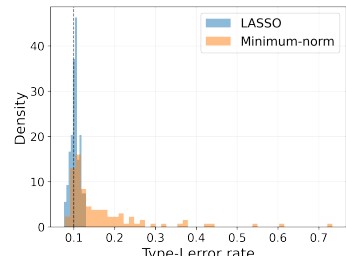

Figure 1: Type-I error rate is contingent on the training algorithm and not solely on the model classes. Unlike the minimum-norm solution, the LASSO fit gives the correct inductive bias in high-dimensional regression, providing better Type-I error control.

In this work, we formalize the idea of misspecified inductive biases in the following way. Assume that a training algorithm $\mathcal{A}$ is used to choose a model $\hat{g}^{(m)} = \mathcal{A}(\mathcal{D}_{tr}^{(m)})$ from the class $\mathcal{G}^{(m)}$. We further assume that the sequence $(\hat{g}^{(m)})_{m \in \mathbb{N}}$ converges to a limiting model $g^*$ in a relevant context-dependent sense. We use different notions of convergence depending on the specific problem under consideration, which will be clear in the following sections. We say that $g^*$ "carries misspecified inductive biases" if it does not equal the target Bayes predictor or regression function $f^*$. There are two possible reasons for $g^*$ carrying misspecified biases: either the limiting model class is small and does not include $f^*$, or the training algorithm $\mathcal{A}$ cannot find the best possible predictor, even asymptotically.

**Notation.** We write $\mathbb{E}_P$ and $\mathsf{Var}_P$ for the expectation and variance of statistics computed using i.i.d. copies of $(X, Y, Z) \sim P$. Consequently, $\mathbb{P}_P(A) = \mathbb{E}_P \mathbb{1}_A$, where $\mathbb{1}_A$ is the indicator of an event $A$. If $\mathbb{E}_P$ and $\mathsf{Var}_P$ are conditioned on some other statistics, we assume those statistics are also computed using i.i.d. samples from $P$. As usual, $\Phi$ is the $N(0,1)$ distribution function. If $(a_m)_{m \in \mathbb{N}}$ and $(b_m)_{m \in \mathbb{N}}$ are sequences of scalars, then $a_m = o(b_m)$ is equivalent to $a_m/b_m \to 0$ as $m \to \infty$ and $a_m = b_m + o(1)$ means $a_m - b_m = o(1)$. If $(V^{(m)})_{m \in \mathbb{N}}$ is a sequence of random variables, where $V^{(m)}$ as constructed using i.i.d. samples of $P^{(m)} \in \mathcal{P}$ for each $m$, then (i) $V^{(m)} = o_p(1)$ means that for every $\varepsilon > 0$ we have $\mathbb{P}_{P^{(m)}}(|V^{(m)}| > \varepsilon) \to 0$ as $m \to \infty$, (ii) $V^{(m)} = \mathcal{O}_p(1)$ means that for every $\varepsilon > 0$ there exists a $M > 0$ such that $\sup_{m \in \mathbb{N}} \mathbb{P}_{P^{(m)}}(|V^{(m)}| > M) < \varepsilon$, (iii) $V^{(m)} = a_m + o_p(1)$ means $V^{(m)} - a_m = o_p(1)$, (iv) $V^{(m)} = o_p(a_m)$ means $V^{(m)}/a_m = o_p(1)$, and (v) $V^{(m)} = \mathcal{O}_p(a_m)$ means $V^{(m)}/a_m = \mathcal{O}_p(1)$. Finally, let $(V_P^{(m)})_{m \in \mathbb{N}, P \in \mathcal{P}}$ be a family of random variables that distributions explicitly depend on $m \in \mathbb{N}$ and $P \in \mathcal{P}$. We give an example to clarify what we mean by "explicitly" depending on a specific distribution. Let $V_P^{(m)} = \frac{1}{m} \sum_{i=1}^{m} (X_i - \mu_P)$, where $\mu_P = \mathbb{E}_P[X]$. Here, $V_P^{(m)}$ explicitly depends on $P$ because of the quantity $\mu_P$. In this example, $X_i$'s outside the expectation can have an arbitrary distribution (unless stated), *i.e.*, could be determined by $P$ or any other distribution. With this context, (i) $V_P^{(m)} = o_{\mathcal{P}}(1)$ means that for every $\varepsilon > 0$ we have $\sup_{P \in \mathcal{P}} \mathbb{P}_P(|V_P^{(m)}| > \varepsilon) \to 0$ as $m \to \infty$, (ii) $V_P^{(m)} = \mathcal{O}_{\mathcal{P}}(1)$ means that for every $\varepsilon > 0$ there exists a $M > 0$ such that $\sup_{m \in \mathbb{N}, P \in \mathcal{P}} \mathbb{P}_P(|V_P^{(m)}| > M) < \varepsilon$, (iii) $V_P^{(m)} = o_{\mathcal{P}}(a_m)$ means $V_P^{(m)}/a_m = o_{\mathcal{P}}(1)$, and (iv) $V_P^{(m)} = \mathcal{O}_{\mathcal{P}}(a_m)$ means $V_P^{(m)}/a_m = \mathcal{O}_{\mathcal{P}}(1)$.

**Related work.** There is a growing literature on the problem of conditional independence testing regarding both theoretical and methodological aspects[3]. From the methodological point of view, there is a great variety of tests with different natures. Perhaps, the most important groups of tests are (i) simulation-based tests [5, 3, 4, 32, 35, 20], (ii) regression-based tests [40, 24, 42, 38, 31, 7], (iii) kernel-based tests [10, 8, 34, 30], and (iv) information-theoretic based tests [29, 16, 39]. Due to the advance of supervised and generative models in recent years, regression and simulation-based tests have become particularly appealing, especially when $Z$ is not low-dimensional or discrete. A related but different line of research is constructing a lower confidence bound for conditional dependence of $X$ and $Y$ given $Z$ [41]. In that work, the authors propose a method that relies on computing the conditional expectation of a possibly misspecified regression model, which can be related to

---

[3] See, for example, Marx and Vreeken [21], Shah and Peters [31], Li and Fan [19], Neykov et al. [23], Watson and Wright [37], Kim et al. [15], Shi et al. [32], Scetbon et al. [30], Tansey et al. [35], Zhang et al. [39], Ai et al. [1]

our method presented in Section 4. Despite the relationship between methods, their motivations, assumptions, and contexts are different.

Simulation-based tests depend on the fact that we can, implicitly or explicitly, approximate the conditional distributions $P_{X|Z}$ or $P_{Y|Z}$. Two relevant simulation-based methods are the conditional randomization and conditional permutation tests (CRT/CPT) [5, 3, 4, 35]. For these tests, Berrett et al. [4] presents robustness results showing that we can *approximately* control Type I error even if our estimates for the conditional distributions are not perfect and we are under a finite-sample regime. However, it is also clear from their results that CRT and CPT might not control Type I error asymptotically when models for conditional distributions are misspecified. On the other hand, regression-based tests work under the assumption that we can accurately approximate the conditional expectations $\mathbb{E}[X \mid Z]$ and $\mathbb{E}[Y \mid Z]$ or other Bayes predictors, which is hardly true if the modeling and training inductive biases are misspecified. To the best of our knowledge, there are no published robustness results for regression-based CI tests like those presented by Berrett et al. [4]. We explore this literature gap.

# 3 Regression-based conditional independence tests under misspecified inductive biases

This section provides results for the Significance Test of Feature Relevance (STFR) [7]. Due to limited space, the results for the Generalized Covariance Measure (GCM) test [31] and the REgression with Subsequent Independence Test (RESIT) [42, 24, 12] are presented in Appendix A. From the results in Appendix A, one can easily derive a double robustness property for both GCM and RESIT, implying that not all models need to be correctly specified or trained with the correct inductive biases for Type-I error control.

## 3.1 Significance Test of Feature Relevance (STFR)

The STFR method studied by Dai et al. [7] offers a scalable approach for conducting conditional independence testing by comparing the performance of two predictors. To apply this method, we first train two predictors $\hat{g}_1^{(m)} : \mathcal{X} \times \mathcal{Z} \to \mathcal{Y}$ and $\hat{g}_2^{(m)} : \mathcal{Z} \to \mathcal{Y}$ on the training set $\mathcal{D}_{tr}^{(m)}$ to predict $Y$ given $(X, Z)$ and $Z$, respectively. We assume that candidates for $\hat{g}_2^{(m)}$ are models in the same class as $\hat{g}_1^{(m)}$ but replacing $X$ with null entries. Using samples from the test set $\mathcal{D}_{te}^{(n)}$, we conduct the test rejecting $H_0 : X \perp\!\!\!\perp Y \mid Z$ if the statistic $\Lambda^{(n,m)} \triangleq \sqrt{n}\bar{T}^{(n,m)}/\hat{\sigma}^{(n,m)}$ exceeds $\tau_\alpha \triangleq \Phi^{-1}(1-\alpha)$, depending on the significance level $\alpha \in (0,1)$. We define $\bar{T}^{(n,m)}$ and $\hat{\sigma}^{(n,m)}$ as

$$\bar{T}^{(n,m)} \triangleq \tfrac{1}{n}\sum_{i=1}^n T_i^{(m)} \text{ and } \hat{\sigma}^{(n,m)} \triangleq \left[\tfrac{1}{n}\sum_{i=1}^n (T_i^{(m)})^2 - \left(\tfrac{1}{n}\sum_{i=1}^n T_i^{(m)}\right)^2\right]^{1/2} \tag{3.1}$$

with $T_i^{(m)} \triangleq \ell(\hat{g}_2^{(m)}(Z_i), Y_i) - \ell(\hat{g}_1^{(m)}(X_i, Z_i), Y_i) + \varepsilon_i$. Here, $\ell$ is a loss function, typically used during the training phase, and $\{\varepsilon_i\}_{i=1}^n \overset{iid}{\sim} N(0, \rho^2)$ are small artificial random noises that do not let $\hat{\sigma}^{(n,m)}$ vanish with a growing training set, thus allowing the asymptotic distribution of $\Lambda^{(n,m)}$ to be standard normal under $H_0 : X \perp\!\!\!\perp Y \mid Z$. If the $p$-value is defined as $p(\mathcal{D}_{te}^{(n)}, \mathcal{D}_{tr}^{(m)}) = 1 - \Phi(\Lambda^{(n,m)})$, the test is equivalently given by

$$\varphi_\alpha^{\text{STFR}}(\mathcal{D}_{te}^{(n)}, \mathcal{D}_{tr}^{(m)}) \triangleq \begin{cases} 1, \text{if } p(\mathcal{D}_{te}^{(n)}, \mathcal{D}_{tr}^{(m)}) \leq \alpha \\ 0, \text{otherwise} \end{cases} \tag{3.2}$$

The rationale behind STFR is that if $H_0 : X \perp\!\!\!\perp Y \mid Z$ holds, then $\hat{g}_1^{(m)}$ and $\hat{g}_2^{(m)}$ should have similar performance in the test set. On the other hand, if $H_0$ does not hold, we expect $\hat{g}_1^{(m)}$ to have significantly better performance, and then we would reject the null hypothesis. Said that, to control STFR's Type-I error, it is necessary that the risk gap between $\hat{g}_1^{(m)}$ and $\hat{g}_2^{(m)}$, $\mathbb{E}_P[\ell(\hat{g}_2^{(m)}(Z), Y) \mid \mathcal{D}_{tr}^{(m)}] - \mathbb{E}_P[\ell(\hat{g}_1^{(m)}(X, Z), Y) \mid \mathcal{D}_{tr}^{(m)}]$, under $H_0$ vanishes as the training set size increases. Moreover, we need the risk gap to be positive for the test to have non-trivial power. These conditions can be met if the risk gap of $g_{1,P}^*$ and $g_{2,P}^*$, the limiting models of $\hat{g}_1^{(m)}$ and $\hat{g}_2^{(m)}$, is the same as the risk gap of the Bayes' predictors

$$f_{1,P}^* \triangleq \arg\min_{f_1} \mathbb{E}_P[\ell(f_1(X, Z), Y)] \text{ and } f_{2,P}^* \triangleq \arg\min_{f_2} \mathbb{E}_P[\ell(f_2(Z), Y)],$$

where the minimization is done over the set of all measurable functions[4]. However, the risk gap between $\hat{g}_1^{(m)}$ and $\hat{g}_2^{(m)}$ will typically not vanish if $g_{1,P}^*$ and $g_{2,P}^*$ are not the Bayes' predictors even under $H_0$. In general, we should expect $g_{1,P}^*$ to perform better than $g_{2,P}^*$ because the second predictor does not depend on $X$. Furthermore, their risk gap can be non-positive even if $f_{1,P}^*$ performs better than $f_{2,P}^*$. In Appendix A.2, we present two examples in which model misspecification plays an important role when conducting STFR. The examples show that Type-I error control and/or power can be compromised due to model misspecification.

To derive theoretical results, we adapt the assumptions from Dai et al. [7]:

**Assumption 3.1.** *There are functions $g_{1,P}^*$, $g_{2,P}^*$, and a constant $\gamma > 0$ such that*

$$\mathbb{E}_P\big[\ell(\hat{g}_2^{(m)}(Z), Y) \mid \mathcal{D}_{tr}^{(m)}\big] - \mathbb{E}_P\big[\ell(g_{2,P}^*(Z), Y)\big] - \Big(\mathbb{E}_P\big[\ell(\hat{g}_1^{(m)}(X, Z), Y) \mid \mathcal{D}_{tr}^{(m)}\big] - \mathbb{E}_P\big[\ell(g_{1,P}^*(X, Z), Y)\big]\Big)$$
$$= \mathcal{O}_\mathcal{P}(m^{-\gamma})$$

**Assumption 3.2.** *There exists a constant $k > 0$ such that*

$$\mathbb{E}_P\big[|T_1^{(m)}|^{2+k} \mid \mathcal{D}_{tr}^{(m)}\big] = \mathcal{O}_\mathcal{P}(1) \text{ as } m \to \infty$$

**Assumption 3.3.** *For every $P \in \mathcal{P}$, there exists a constant $\sigma_P^2 > 0$ such that*

$$\mathsf{Var}_P[T_1^{(m)} \mid \mathcal{D}_{tr}^{(m)}] - \sigma_P^2 = o_\mathcal{P}(1) \text{ as } m \to \infty \text{ and } \inf_{P \in \mathcal{P}} \sigma_P^2 > 0$$

Finally, we present the results for this section. We start with an extension of Theorem 2 presented by Dai et al. [7] in the case of misspecified inductive biases.

**Theorem 3.4.** *Suppose that Assumptions 3.1, 3.2, and 3.3 hold. If $n$ is a function of $m$ such that $n \to \infty$ and $n = o(m^{2\gamma})$ as $m \to \infty$, then*

$$\mathbb{E}_P[\varphi_\alpha^{\text{STFR}}(\mathcal{D}_{te}^{(n)}, \mathcal{D}_{tr}^{(m)})] = 1 - \Phi\Big(\tau_\alpha - \sqrt{\tfrac{n}{\sigma_P^2}}\Omega_P^{\text{STFR}}\Big) + o(1)$$

*where $o(1)$ denotes uniform convergence over all $P \in \mathcal{P}$ as $m \to \infty$ and*

$$\Omega_P^{\text{STFR}} \triangleq \mathbb{E}_P[\ell(g_{2,P}^*(Z), Y)] - \mathbb{E}_P[\ell(g_{1,P}^*(X, Z), Y)]$$

Theorem 3.4 demonstrates that the performance of STFR depends on the limiting models $g_{1,P}^*$ and $g_{2,P}^*$. Specifically, if $\Omega_P^{\text{STFR}} > 0$, then $\mathbb{E}_P[\varphi_\alpha^{\text{STFR}}(\mathcal{D}_{te}^{(n)}, \mathcal{D}_{tr}^{(m)})] \to 1$ even if $H_0 : X \perp\!\!\!\perp Y \mid Z$ holds. In practice, we should expect $\Omega_P^{\text{STFR}} > 0$ because of how we set the class for $\hat{g}_2^{(m)}$. In contrast, we could have $\Omega_P^{\text{STFR}} \leq 0$, and then $\mathbb{E}_P[\varphi_\alpha^{\text{STFR}}(\mathcal{D}_{te}^{(n)}, \mathcal{D}_{tr}^{(m)})] \leq \alpha + o(1)$, even if the gap between Bayes' predictors is positive. See examples in Appendix A.2 for both scenarios. Next, we provide Corollary 3.6 to clarify the relationship between testing and misspecification errors. This corollary formalizes the intuition that controlling Type-I error is directly related to misspecification of $g_{2,P}^*$, while minimizing Type-II error is directly related to misspecification of $g_{1,P}^*$.

**Definition 3.5.** *For a distribution $P$ and a loss function $\ell$, define the misspecification gaps:*

$$\Delta_{1,P} \triangleq \mathbb{E}_P[\ell(g_{1,P}^*(X, Z), Y)] - \mathbb{E}_P[\ell(f_{1,P}^*(X, Z), Y)] \text{ and } \Delta_{2,P} \triangleq \mathbb{E}_P[\ell(g_{2,P}^*(Z), Y)] - \mathbb{E}_P[\ell(f_{2,P}^*(Z), Y)]$$

The misspecification gaps defined in Definition 3.5 quantify the difference between the limiting predictors $g_{1,P}^*$ and $g_{2,P}^*$ and the Bayes predictors $f_{1,P}^*$ and $f_{2,P}^*$, *i.e.*, give a misspecification measure for $g_{1,P}^*$ and $g_{2,P}^*$. Corollary 3.6 implies that the STFR controls Type-I error asymptotically if $\Delta_{2,P} = 0$, and guarantees non-trivial power if the degree of misspecification of $g_{1,P}^*$ is not large compared to the performance difference of the Bayes predictors $\Delta_P$, that is, when $\Delta_P - \Delta_{1,P} > 0$.

**Corollary 3.6** (Bounding testing errors). *Suppose we are under the conditions of Theorem 3.4.*

*(Type-I error) If $H_0 : X \perp\!\!\!\perp Y \mid Z$ holds, then*

$$\mathbb{E}_P[\varphi_\alpha^{\text{STFR}}(\mathcal{D}_{te}^{(n)}, \mathcal{D}_{tr}^{(m)})] \leq 1 - \Phi\Big(\tau_\alpha - \sqrt{\tfrac{n}{\sigma_P^2}}\Delta_{2,P}\Big) + o(1)$$

*where $o(1)$ denotes uniform convergence over all $P \in \mathcal{P}_0$ as $m \to \infty$.*

*(Type-II error) In general, we have*

$$1 - \mathbb{E}_P[\varphi_\alpha^{\text{STFR}}(\mathcal{D}_{te}^{(n)}, \mathcal{D}_{tr}^{(m)})] \leq \Phi\Big(\tau_\alpha - \sqrt{\tfrac{n}{\sigma_P^2}}(\Delta_P - \Delta_{1,P})\Big) + o(1)$$

*where $o(1)$ denotes uniform convergence over all $P \in \mathcal{P}$ as $m \to \infty$ and $\Delta_P \triangleq \mathbb{E}_P[\ell(f_{2,P}^*(Z), Y)] - \mathbb{E}_P[\ell(f_{1,P}^*(X, Z), Y)]$.*

---

[4]We assume $f_{1,P}^*$ and $f_{2,P}^*$ to be well-defined and unique.

# 4   A robust regression-based conditional independence test

This section introduces the Rao-Blackwellized Predictor Test (RBPT), a misspecification robust conditional independence test based on ideas from both regression and simulation-based CI tests. The RBPT assumes that we can implicitly or explicitly approximate the conditional distribution of $X \mid Z$ and does not require inductive biases to be correctly specified. Because RBPT involves comparing the performance of two predictors and requires an approximation of the distribution of $X \mid Z$, we can directly compare it with the STFR [7] and the conditional randomization/permutation tests (CRT/CPT) [5, 4]. The RBPT can control Type-I error under relatively weaker assumptions compared to other tests, allowing some misspecified inductive biases.

The RBPT can be summarized as follows: (i) we train $\hat{g}^{(m)}$ that predicts $Y$ given $(X, Z)$ using $\mathcal{D}_{tr}^{(m)}$; (ii) we obtain the Rao-Blackwellized predictor $h^{(m)}$ by smoothing $\hat{g}^{(m)}$, *i.e.*,

$$h^{(m)}(z) \triangleq \int \hat{g}^{(m)}(x, z) \mathrm{d}P_{X \mid Z=z}(x),$$

then (iii) compare its performance with $\hat{g}^{(m)}$'s using the test set $\mathcal{D}_{te}^{(n)}$ and a convex loss[5] function $\ell$ (not necessarily used to train $\hat{g}^{(m)}$), and (iv) if the performance of $\hat{g}^{(m)}$ is statistically better than $h^{(m)}$'s, we reject $H_0 : X \perp\!\!\!\perp Y \mid Z$. The procedure described here bears a resemblance to the Rao-Blackwellization of estimators. In classical statistics, the Rao-Blackwell theorem [17] states that by taking the conditional expectation of an estimator with respect to a sufficient statistic, we can obtain a better estimator if the loss function is convex. In our case, the variable $Z$ can be seen as a "sufficient statistic" for $Y$ under the assumption of conditional independence $H_0 :$ $X \perp\!\!\!\perp Y \mid Z$. If $H_0$ holds and the loss $\ell(\hat{y}, y)$ is convex in its first argument, we can show using Jensen's inequality that the resulting model $h^{(m)}$ has a lower risk relative to the initial model $\hat{g}^{(m)}$, *i.e.*, $\mathbb{E}_P[\ell(h^{(m)}(Z), Y) \mid \mathcal{D}_{tr}^{(m)}] - \mathbb{E}_P[\ell(\hat{g}^{(m)}(X, Z), Y) \mid \mathcal{D}_{tr}^{(m)}] \le 0$. Then, the risk gap in RBPT is non-positive under $H_0$ in contrast with STFR's risk gap,

---

**Algorithm 1:** Obtaining $p$-value for the RBPT

**1 Input:** (i) Test set $\mathcal{D}_{te}^{(n)} = \{(X_i, Y_i, Z_i)\}_{i=1}^n$, (ii) initial predictor $\hat{g}^{(m)}$, (iii) conditional distribution estimate $\hat{Q}_{X \mid Z}^{(m)}$, (iv) convex loss function $\ell$;

**2 Output:** p-value $p$;

**3** For each $i \in [n]$, get
$$\hat{h}^{(m)}(Z_i) = \int \hat{g}^{(m)}(x, Z_i) \mathrm{d}\hat{Q}_{X \mid Z=Z_i}^{(m)}(x);$$

**4** Compute $\Xi^{(n,m)} \triangleq \sqrt{n}\bar{T}^{(n,m)}/\hat{\sigma}^{(n,m)}$ where $\bar{T}^{(n,m)} \triangleq \frac{1}{n}\sum_{i=1}^n T_i^{(m)}$ with

$$T_i^{(m)} \triangleq \ell(\hat{h}^{(m)}(Z_i), Y_i) - \ell(\hat{g}^{(m)}(X_i, Z_i), Y_i)$$

and $\hat{\sigma}^{(n,m)}$ being $\{T_i\}$'s sample std dev (Eq. 3.1).

**5 return** $p = 1 - \Phi(\Xi^{(n,m)})$.

---

which we should expect to be always non-negative given the definition of $\hat{g}_2^{(m)}$ in that case. That fact negatively biases the RBPT test statistic, enabling better Type-I error control.

In practice, we cannot compute $h^{(m)}$ exactly because $P_{X \mid Z}$ is usually unknown. Then, we use an approximation $\hat{Q}_{X \mid Z}^{(m)}$, which can be given explicitly, *e.g.*, using probabilistic classifiers or conditional density estimators [13], or implicitly, *e.g.*, using generative adversarial networks (GANs) [22, 3]. We assume that $\hat{Q}_{X \mid Z}^{(m)}$ is obtained using the training set. The approximated $h^{(m)}$ is

$$\hat{h}^{(m)}(z) \triangleq \int \hat{g}^{(m)}(x, z) \mathrm{d}\hat{Q}_{X \mid Z=z}^{(m)}(x)$$

where the integral can be solved numerically in case $\hat{Q}_{X \mid Z}^{(m)}$ has a known probability mass function or Lebesgue density (*e.g.*, via trapezoidal rule) or via Monte Carlo integration in case we can only sample from $\hat{Q}_{X \mid Z}^{(m)}$. Finally, for a fixed significance level $\alpha \in (0, 1)$, the test $\varphi_\alpha^{\mathrm{RBPT}}$ is given by Equation 3.2 where the $p$-value is obtained via Algorithm 1.

Before presenting RBPT results, we introduce some assumptions. Let $Q_{X \mid Z}^*$ represent the limiting model for $\hat{Q}_{X \mid Z}^{(m)}$. The conditional distribution $Q_{X \mid Z}^*$ depends on the underlying distribution $P$, but we omit additional subscripts for ease of notation. Assumption 4.1 defines the limiting models and fixes a convergence rate.

---

[5] The loss function $\ell(\hat{y}, y)$ needs to be convex with respect to its first entry ($\hat{y}$) for all $y$. Both the test set and training set sizes, and the loss function $\ell$ can be chosen using the heuristics introduced by Dai et al. [7].

**Assumption 4.1.** *There is a function $g_P^*$, a conditional distribution $Q_{X|Z}^*$, and a constant $\gamma > 0$ s.t.*

$$\mathbb{E}_P\left[\left\|\hat{g}^{(m)}(Z) - g_P^*(Z)\right\|_2^2 \,\Big|\, \mathcal{D}_{tr}^{(m)}\right] = \mathcal{O}_{\mathcal{P}}(m^{-\gamma}) \text{ and } \mathbb{E}_P\left[d_{TV}(\hat{Q}_{X|Z}^{(m)}, Q_{X|Z}^*) \mid \mathcal{D}_{tr}^{(m)}\right] = \mathcal{O}_{\mathcal{P}}(m^{-\gamma})$$

*where $d_{TV}$ denotes the total variation (TV) distance. Additionally, assume that both $\hat{Q}_{X|Z}^{(m)}$ and $Q_{X|Z}^*$ are dominated by a common $\sigma$-finite measure which does not depend on $Z$ or $m$.*

The common dominating measure in Assumption 4.1 could be, for example, the Lebesgue measure in $\mathbb{R}^{d_X}$. Next, Assumption 4.2 imposes additional constraints on the limiting model $Q_{X|Z}^*$. Under that assumption, the limiting misspecification level must be uniformly bounded over all $P \in \mathcal{P}$.

**Assumption 4.2.** *For all $P \in \mathcal{P}$, the chi-square divergence*

$$\chi^2\left(Q_{X|Z}^*\|P_{X|Z}\right) \triangleq \int \frac{\mathrm{d}Q_{X|Z}^*}{\mathrm{d}P_{X|Z}} \mathrm{d}Q_{X|Z}^* - 1$$

*is a well-defined integrable random variable and $\sup_{P \in \mathcal{P}} \mathbb{E}_P\left[\chi^2\left(Q_{X|Z}^*\|P_{X|Z}\right)\right] < \infty$.*

Now, assume $\hat{g}^{(m)}$ is chosen from a model class $\mathcal{G}^{(m)}$. Assumption 4.3 imposes constraints on the model classes $\{\mathcal{G}^{(m)}\}$ and loss function $\ell$.

**Assumption 4.3.** *Assume (i) $\sup_{g \in \mathcal{G}^{(m)}} \sup_{(x,z) \in \mathcal{X} \times \mathcal{Z}} \|g(x,z)\|_1 \leq M < \infty$, for some real and positive $M > 0$, uniformly for all $m$, and (ii) that $\ell$ is a $L$-Lipschitz loss function (with respect to its first argument) for a certain $L > 0$, i.e., for any $\hat{y}, \hat{y}', y \in \mathcal{Y}$, we have that $|\ell(\hat{y}, y) - \ell(\hat{y}', y)| \leq L \|\hat{y} - \hat{y}'\|_2$.*

Assumption 4.3 is valid by construction since we choose $\mathcal{G}^{(m)}$ and the loss function $\ell$. That assumption is satisfied when, for example, (a) models in $\cup_m \mathcal{G}^{(m)}$ are uniformly bounded, (b) $\ell(\hat{y}, y) = \|\hat{y} - y\|_p^p$ with $p \geq 1$, and (c) $\mathcal{Y}$ is a bounded subset of $\mathbb{R}^{d_Y}$, i.e., in classification problems and most of the practical regression problems. The loss $\ell(\hat{y}, y) = \|\hat{y} - y\|_p^p$, with $p \geq 1$, is also convex with respect to its first entry and then a suitable loss for RBPT. It is important to emphasize that $\ell$ does not need to be the same loss function used during the training phase. For example, we could use $\ell(\hat{y}, y) = \|\hat{y} - y\|_2^2$ in classification problems, where $y$ is a one-hot encoded class label and $\hat{y}$ is a vector of predicted probabilities given by a model trained using the cross-entropy loss.

**Theorem 4.4.** *Suppose that Assumptions 3.2, 3.3, 4.1, 4.2, and 4.3 hold. If $n$ is a function of $m$ such that $n \to \infty$ and $n = o(m^\gamma)$ as $m \to \infty$, then*

$$\mathbb{E}_P[\varphi_\alpha^{\text{RBPT}}(\mathcal{D}_{te}^{(n)}, \mathcal{D}_{tr}^{(m)})] = 1 - \Phi\left(\tau_\alpha - \sqrt{\frac{n}{\sigma_P^2}}\Omega_P^{\text{RBPT}}\right) + o(1)$$

*where $o(1)$ denotes uniform convergence over all $P \in \mathcal{P}$ as $m \to \infty$ and $\Omega_P^{\text{RBPT}} = \Omega_{P,1}^{\text{RBPT}} - \Omega_{P,2}^{\text{RBPT}}$ with*

$$\Omega_{P,1}^{\text{RBPT}} \triangleq \mathbb{E}_P\left[\ell\left(\int g_P^*(x, Z)\mathrm{d}Q_{X|Z}^*(x), Y\right)\right] - \mathbb{E}_P\left[\ell\left(\int g_P^*(x, Z)\mathrm{d}P_{X|Z}(x), Y\right)\right]$$

*and*

$$\underbrace{\Omega_{P,2}^{\text{RBPT}}}_{\text{Jensen's gap}} \triangleq \mathbb{E}_P\left[\ell(g_P^*(X, Z), Y)\right] - \mathbb{E}_P\left[\ell\left(\int g_P^*(x, Z)\mathrm{d}P_{X|Z}(x), Y\right)\right]$$

When $H_0 : X \perp\!\!\!\perp Y \mid Z$ holds and $\ell$ is a strictly convex loss function (w.r.t. its first entry), we have that $\Omega_{P,2}^{\text{RBPT}} > 0$, allowing[6] some room for the "incorrectness" of $Q_{X|Z}^*$. That is, from Theorem 4.4, as long as $\Omega_P^{\text{RBPT}} \leq 0$, i.e., if $Q_{X|Z}^*$'s incorrectness (measured by $\Omega_{P,1}^{\text{RBPT}}$) is not as big as Jensen's gap $\Omega_{P,2}$, RBPT has asymptotic Type-I error control. Uniform asymptotic Type-I error control is possible if $\sup_{P \in \mathcal{P}_0} \Omega_P^{\text{RBPT}} \leq 0$. This is a great improvement of previous work (*e.g.*, STFR, GCM, RESIT, CRT, CPT) since there is no need for any model to converge to the ground truth if $\Omega_{P,1}^{\text{RBPT}} \leq \Omega_{P,2}^{\text{RBPT}}$, which is a weaker condition. See however that a small $\Omega_{P,2}^{\text{RBPT}}$ reduces the room for $Q_{X|Z}^*$ incorrectness. In the extreme case, when $g_P^*$ is the Bayes predictor, and therefore does not depend on $X$ under $H_0$, we need[7] $Q_{X|Z}^* = P_{X|Z}$ almost surely. On the other hand, if $g_P^*$ is close to the Bayes predictor, RBPT has better power. That imposes an expected trade-off between Type-I error control and

---

[6] In practice, we do not need $\ell$ to be strictly convex for the Jensen's gap to be positive. Assuming that $g_P^*$ depends on $X$ under $H_0$ is necessary, though. That condition is usually true when $g_P^*$ is not the Bayes predictor.

[7] In this case, Assumption 3.3 is not true. We need to include artificial noises in the definition of $T_i$ as it was done in STFR by Dai et al. [7] in case we have high confidence that models converge to the ground truth.

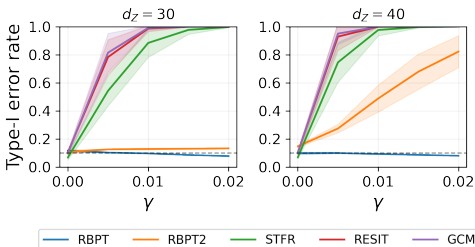 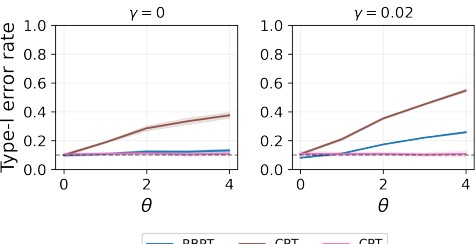

| RBPT | RBPT2 | STFR | RESIT | GCM |
| --- | --- | --- | --- | --- |

| RBPT | CRT | CPT |
| --- | --- | --- |

Figure 2: Type-I error rates ($c = 0$). In the first two plots, we set $\theta = 0$ for RBPT, permitting Type-I error control across different $d_Z$ values (Theorem 4.4), while RBPT2 allows Type-I error control for moderate $d_Z$. All the baselines fail to control Type-I errors regardless of $d_Z$. The last two plots illustrate that CRT emerges as the least robust test in this context, succeeded by RBPT and CPT.

power. To make a comparison with Berrett et al. [4] 's results in the case of CRT and CPT, we can express our remark in terms of the TV distance between $Q^*_{X|Z}$ and $P_{X|Z}$. It can be shown that if $\mathbb{E}_P[d_{\mathrm{TV}}(Q^*_{X|Z}, P_{X|Z})] \leq \Omega^{\mathrm{RBPT}}_{P,2}/(2ML)$, then Type-I error control is guaranteed (see Appendix A.5). This contrasts with Berrett et al. [4] 's results because $\mathbb{E}_P[d_{\mathrm{TV}}(Q^*_{X|Z}, P_{X|Z})] = 0$ is not needed.

We conclude this section with some relevant observations related to the RBPT.

**On RBPT's power.** Like STFR, non-trivial power is guaranteed if the predictor $g^*_P$ is *good enough*. Indeed, the second part of Corollary 3.6 can be applied for an upper bound on RBPT's Type-II error.

**Semi-supervised learning.** Let $Y$ denote a label variable. Situations in which unlabeled samples $(X_i, Z_i)$ are abundant while labeled samples $(X_i, Y_i, Z_i)$ are scarce happen in real applications of conditional independence testing [5, 4]. RBPT is well suited for those cases because the practitioner can use the abundant data to estimate $P_{X|Z}$ flexibly. The semi-supervised learning scenario also applies to RBPT2, which we describe next.

**Running RBPT when it is hard to estimate $P_{X|Z}$: the RBPT2.** There might be situations in which it is hard to estimate the full conditional distribution $P_{X|Z}$. An alternative approach would be estimating the Rao-Blackwellized predictor directly using a second regressor. After training $\hat{g}^{(m)}$, we could use the training set, organizing it in pairs $\{(Z_i, \hat{g}^{(m)}(Z_i, X_i))\}$, to train a second predictor $\hat{h}^{(m)}$ to predict $\hat{g}^{(m)}(Z, X)$ given $Z$. That predictor could be trained to minimize the mean-squared error. The model $\hat{h}^{(m)}$ should be more complex than $\hat{g}^{(m)}$, in the sense that we should hope that the first model performs better than the second under $H_0$ in predicting $Y$. Consequently, this approach is effective when unlabeled samples are abundant, and we can train $\hat{h}$ using both unlabeled data and the given training set. After obtaining $\hat{h}^{(m)}$, the test is conducted normally. We name this version of RBPT as "RBPT2". We include a note on how to adapt Theorem 4.4 for RBPT2 in Appendix A.6.

## 5 Experiments

We empirically[8] analyze RBPT/RBPT2 in the following experiments and compare them with relevant benchmarks, especially when the used models are misspecified. We assume $\alpha = 10\%$ and $\ell(\hat{y}, y) = (\hat{y} - y)^2$. The benchmarks encompass STFR [7], GCM [31], and RESIT [42], which represent regression-based CI tests. Furthermore, we examine the conditional randomization/permutation tests (CRT/CPT) [5, 4] that necessitate the estimation of $P_{X|Z}$.

**Artificial data experiments.** Our setup takes inspiration from Berrett et al. [4], and the data is generated as

$$Z \sim N\left(0, I_{d_Z}\right), \quad X \mid Z \sim N\left((b^\top Z)^2, 1\right), \quad Y \mid X, Z \sim N\left(cX + a^\top Z + \gamma(b^\top Z)^2, 1\right).$$

Here, $d_Z$ denotes the dimensionality of $Z$, $a$ and $b$ are sampled from $N\left(0, I_{d_Z}\right)$, the constant $c$ determines the conditional dependence of $X$ and $Y$ on $Z$, and the parameter $\gamma$ dictates the hardness of conditional independence testing: a non-zero $\gamma$ implies potential challenges in Type-I error control as there might be a pronounced marginal dependence between $X$ and $Y$ under $H_0$. Moreover, the training (resp. test) dataset consists of 800 (resp. 200) entries, and every predictor we employ operates on linear regression. RESIT employs Spearman's correlation between residuals as a test statistic while

---

[8]Code in `https://github.com/felipemaiapolo/cit`.

CRT and CPT[9] deploy STFR's test statistic, all of them with $p$-values determined by conditional sampling/permutations executed 100 times ($B = 100$), considering $\hat{Q}_{X|Z} = N\left((b^\top Z)^2 + \theta, 1\right)$. The value of $\theta$ gives the error level in approximating $P_{X|Z}$. To get $\hat{h}$ in the two variations of RBPT, we either use $\hat{Q}_{X|Z}$ (RBPT) or kernel ridge regression (KRR) equipped with a polynomial kernel to predict $\hat{g}_1(X, Z)$ from $Z$ (RBPT2). We sample generative parameters $(a, b)$ five times using different random seeds, and for each iteration, we conduct 480 Monte Carlo simulations to estimate Type-I error and power. The presented results are the average ($\pm$ standard deviation) estimated Type-I error/power across iterations.

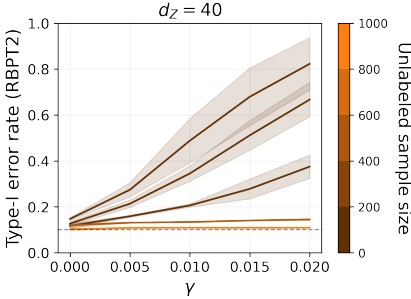

Figure 3: Making RBPT2 more robust using unlabeled data. With $d_Z = 40$, we gradually increase the unlabeled sample size from 0 to 1000 when fitting $\hat{h}$. The results show that a larger unlabeled sample size leads to effective Type-I error control. Even though we present this result for RBPT2, the same pattern is expected for RBPT in the presence of unlabeled data.

In Figure 2 (resp. 4) we compare our methods' Type-I error rates (with $c = 0$) (resp. power) against benchmarks. Regarding Figure 2, we focus on regression-based tests (STFR, GCM, and RESIT) in the first two plots and on simulation-based tests (CRT and CPT) in the last two plots. Regarding the first two plots, it is not straightforward to compare the level of misspecification between our methods and the benchmarks, so we use this as an opportunity to illustrate Theorem 4.4 and results from Section 3 and Appendix A. Fixing $\theta = 0$ for RBPT, the Rao-Blackwellized predictor $h$ is perfectly obtained, permitting Type-I error control regardless of the chosen $d_Z$. Using KRR for RBPT2 makes $\hat{h}$ close to $h$ when $d_Z$ is not big and permits Type-I error control. When $d_Z$ is big, more data is needed to fit $\hat{h}$, which can be accomplished using unlabeled data, as demonstrated in Figure 3 and commented in Section 4. On the other hand, Type-I error control is always violated for STFR, GCM, and RESIT when $\gamma$ grows. Regarding the final two plots, we can more readily assess the robustness of the methods when discrepancies arise between $\hat{Q}_{X|Z}$ and $P_{X|Z}$ as influenced by varying $\theta$. Figure 2 illustrates that CRT is the least robust test in this context, succeeded by RBPT and CPT. In Figure 4, we investigate how powerful RBPT and RBPT2 can be in practice when $d_Z = 30$. We compare our methods with CPT (when $\theta = 0$), which seems to have practical robustness against misspecified inductive biases. Figure 4 shows that RBPT2 and CPT have similar power while RBPT is slightly more conservative.

Some concluding remarks are needed. First, RBPT and RBPT2 have shown to be practical and robust alternatives to conditional independence testing, exhibiting reasonable Type-I error control, mainly when employed in conjunction with a large unlabeled dataset, and power. Second, while CPT demonstrates notable robustness and relatively good power, its practicality falls short compared to RBPT (or RBPT2). This is because CPT needs a known density functional form for $\hat{Q}_{X|Z}$ (plus the execution of MCMC chains) whereas RBPT (resp. RBPT2) can rely on conventional Monte Carlo integration using samples from $\hat{Q}_{X|Z}$ (resp. supervised learning).

**Real data experiments.** For our subsequent experiments, we employ the car insurance dataset examined by Angwin et al. [2]. This dataset encompasses four US states (California, Illinois, Missouri, and Texas) and includes information from numerous insurance providers compiled at the ZIP code granularity. The data offers a risk metric and the insurance price levied on a hypothetical customer with consistent attributes from every ZIP

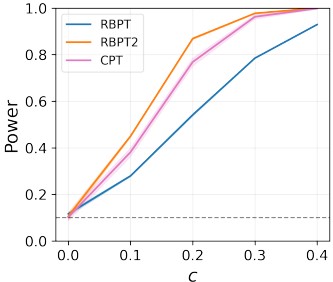

Figure 4: Power curves for different methods. We compare our methods with CPT (when $\theta = 0$), which seems to have practical robustness against misspecified inductive biases. RBPT2 and CPT have similar power, while RBPT is slightly more conservative.

code. ZIP codes are categorized as either minority or non-minority, contingent on the percentage of non-white residents. The variables in consideration are $Z$, denoting the driving risk; $X$, an indicator for minority ZIP codes; and $Y$, signifying the insurance price. A pertinent question revolves around

---

[9]For CPT execution, the Python script at http://www.stat.uchicago.edu/~rina/cpt.html was used, operating a single MCMC chain and preserving all other parameters as defined by the original authors.

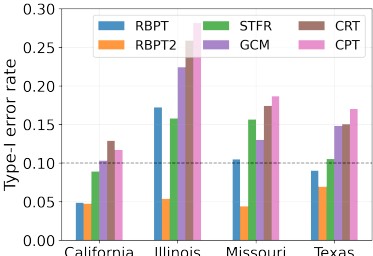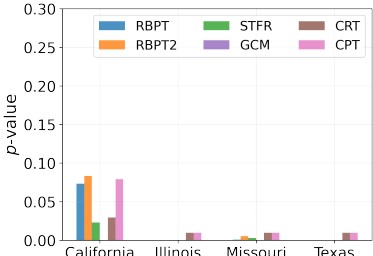

Figure 5: Type-I error control and power analysis using car insurance data [2]. The first plot shows that RBPT and RBPT2 have better control over Type-I errors compared to all other methods, including CPT. The second plot reveals that all methods give the same qualitative result, corroborating the findings of Angwin et al. [2], suggesting that RBPT and RBPT2 can have good power while being more robust to Type-I errors.

the validity of the null hypothesis $H_0 : X \perp\!\!\!\perp Y \mid Z$, essentially questioning if demographic biases influence pricing.

We split our experiments into two parts. In the initial part, our primary goal is to compare the Type-I error rate across various tests. To ensure that $H_0$ is valid, we discretize $Z$ into twenty distinct values and shuffle the $Y$ values corresponding to each discrete $Z$ value. If a test maintains Type-I error control, we expect it to reject $H_0$ for at most $\alpha = 10\%$ of the companies in each state. In the second part, we focus on assessing the power of our methods. Given our lack of ground truth, we qualitatively compare RBPT and RBPT2 findings with those obtained by baseline methods and delineated by Angwin et al. [2], utilizing a detailed and multifaceted approach. In this last experiment, we aggregate the analysis for each state without conditioning on the firm. We resort to logistic regression for estimating the distribution of $X \mid Z$ used by RBPT, GCM, CRT, and CPT. For RBPT2, we use a CatBoost regressor [26] to yield the Rao-Blackwellized predictor. We omit RESIT in this experiment as the additive model assumption is inappropriate. Both CRT and CPT methods utilize the same test metrics as STFR. The first plot[10] of Figure 5 shows that RBPT and RBPT2 methods have better control over Type-I errors compared to all other methods, including CPT. The second plot reveals that all methods give the same qualitative result that discrimination against minorities in ZIP codes is most evident in Illinois, followed by Texas, Missouri, and California. These findings corroborate with those of Angwin et al. [2], indicating that our methodology has satisfactory power while maintaining a robust Type-I error control.

# 6 Conclusion

In this work, we theoretically and empirically showed that widely used regression-based conditional independence tests are sensitive to the specification of inductive biases. Furthermore, we introduced the Rao-Blackwellized Predictor Test (RBPT), a misspecification-robust conditional independence test. RBPT is theoretically grounded and has been shown to perform well in practical situations compared to benchmarks.

**Limitations and future work.** Two limitations of RBPT are that (i) the robustness of RBPT can lead to a more conservative test, as we have seen in the simulations; moreover, (ii) it requires the estimation of the conditional distribution $P_{X|Z}$, which can be challenging. To overcome the second problem, we introduced a variation of RBPT, named RBPT2, in which the Rao-Blackwellized predictor is obtained in a supervised fashion by fitting a second model $\hat{h} : \mathcal{Z} \to \mathcal{Y}$ that predicts the outputs of the first model $\hat{g} : \mathcal{X} \times \mathcal{Z} \to \mathcal{Y}$. However, this solution only works if $\hat{h}$ is better than $\hat{g}$ in predicting $Y$ under $H_0$, which ultimately depends on the model class for $\hat{h}$ and how that model is trained. Future research directions may include (i) theoretically studying the power of RBPT in more detail and (ii) better understanding RBPT2 from a theoretical or methodological point of view, *e.g.*, answering questions on how to choose and train the second model.

# 7 Acknowledgements

This paper is based upon work supported by the National Science Foundation (NSF) under grants no. 1916271, 2027737, 2113373, and 2113364.

---

[10]We run the experiment for 48 different random seeds and report the average Type-I error rate.

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

## A    Extra content

### A.1    Misspecified inductive biases in modern statistics and machine learning

We present a toy experiment to empirically demonstrate how the training algorithm can prevent us from accurately estimating the Bayes predictor even when the model class is correctly specified, leading to invalid significance tests. We work in the context of a high-dimensional (overparameterized) regression with a training set of 250 observations and $\geq 300$ covariates. We use the Significance Test of Feature Relevance test[11] (STFR) [7] to conduct the CI test. The data are generated as

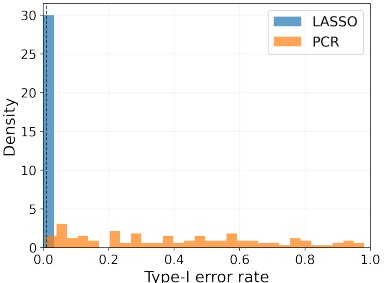

$$Z \sim N(0, I_{300}), \ X \mid Z \sim N(\beta^\top Z, 1), \ Y \mid X, Z \sim N(\beta^\top Z, 1)$$

where the first thirty entries of $\beta$ are set to 1, and the remaining entries are zero. See that $X \perp\!\!\!\perp Y \mid Z$ and that $Y$ is linearly related to $Z$ and $(X, Z)$, and then the class of linear predictors is correctly specified when predicting $Y$ from $Z$ or $(X, Z)$. To perform the STFR test, we use LASSO ($\|\cdot\|_1$ penalization term added to empirical squared error) and principal components regression (PCR) to train the linear predictors. Since $\beta$

Figure 6: Type-I error rate is contingent on the training algorithm and not solely on the model classes. Unlike PCR, the LASSO fit provides the correct inductive bias in high-dimensional regression, controlling Type-I error.

is sparse, the LASSO fit provides the correct inductive bias while PCR leads to *misspecification*. We set the significance level to $\alpha = 1\%$ and estimate the Type-I error rate for 100 different training sets. Figure 6 provides the Type-I error rate empirical distribution and illustrates that, despite using the same model class for both fitting methods, the training algorithm induces the wrong model in the PCR case, implying an invalid test most of the time.

### A.2    Examples on when STFR fails

Examples A.1 and A.2 show simple situations in which Type-I error control is compromised or the conditional independence test has no power due to model misspecification. As we see in the next examples, Type-I error control is directly related to $\mathcal{G}_2$ misspecification, while Type-II error minimization directly relates to $\mathcal{G}_1$ misspecification.

**Example A.1** (No Type-I error control). *Suppose $Y = Z + Z^2 + \varepsilon_y$ and $X = Z^2 + \varepsilon_x$, where $\varepsilon_y, \varepsilon_x \sim N(0, 1)$ are independent noise variables and $Z$ has finite variance. Consequently, $X \perp\!\!\!\perp Y \mid Z$. Let $\mathcal{G}_1$ and $\mathcal{G}_2$ be the classes of linear regressors with no intercept, i.e.,*

$$\mathcal{G}_1 = \{g_1(x, z) = \beta_x x + \beta_z z : \beta_x, \beta_z \in \mathbb{R}\} \text{ and } \mathcal{G}_2 = \{g_2(z) = \beta_z z : \beta_z \in \mathbb{R}\}.$$

*If $\ell$ denotes the mean squared error, we have that $\mathbb{E}_P[\ell(g_{2,P}^*(Z), Y)] - \mathbb{E}_P[\ell(g_{1,P}^*(X, Z), Y)] > 0$ because the model class $\mathcal{G}_2$ is misspecified, that is, it does not contain the Bayes predictor given by the conditional expectation $\mathbb{E}[Y \mid Z]$.*

**Example A.2** (Powerless test). *Suppose $Y = Z + \sin(X) + \varepsilon_y$, where $Z, X, \varepsilon_y \overset{iid}{\sim} N(0, 1)$. Consequently, $X \not\perp\!\!\!\perp Y \mid Z$. Define $\mathcal{G}_1$ and $\mathcal{G}_2$ as in Example A.1. If $\ell$ denotes the mean squared error, we have that[12] $\mathbb{E}_P[\ell(g_{2,P}^*(Z), Y)] - \mathbb{E}_P[\ell(g_{1,P}^*(X, Z), Y)] = 0$ even though the different Bayes' predictors have a difference in performance. This happens because the model $g_{1,P}^*$ is not close enough to the Bayes predictor $\mathbb{E}_P[Y \mid X, Z]$.*

### A.3    Generalized Covariance Measure (GCM) test

In the GCM test proposed by Shah and Peters [31], the expected value of the conditional covariance between $X$ and $Y$ given $Z$ is estimated and then tested to determine if it equals zero. To simplify the exposition, we consider $X$ and $Y$ univariate and work in a setup similar to the STFR's. If $(X, Y, Z) \sim P$, the GCM test relies on the observation that we can always write

$$X = f_{1,P}^*(Z) + \epsilon \ \text{ and } \ Y = f_{2,P}^*(Z) + \eta,$$

---

[11]See Section 3.1 for more details

[12]Because $\mathbb{E}_P[XY] = \mathbb{E}_P[X\mathbb{E}_P[Y|X]] = \mathbb{E}_P[X \cdot 0] = 0$.

where $f_{1,P}^*(Z) = \mathbb{E}_P[X \mid Z]$ and $f_{2,P}^*(Z) = \mathbb{E}_P[Y \mid Z]$ while the error terms $\{\epsilon, \eta\}$ have zero mean when conditioned on $Z$. Consequently, we can write $\mathbb{E}_P[\text{Cov}_P(X, Y \mid Z)] = \mathbb{E}_P[\epsilon\eta]$. To estimate $\mathbb{E}_P[\text{Cov}_P(X, Y \mid Z)]$, we can first fit two models $\hat{g}_1^{(m)} : \mathcal{Z} \to \mathcal{X}$ and $\hat{g}_2^{(m)} : \mathcal{Z} \to \mathcal{Y}$, that approximate $f_{1,P}^*$ and $f_{2,P}^*$, using the training set $\mathcal{D}_{tr}^{(m)}$, and then compute an empirical version of $\mathbb{E}_P[\epsilon\eta] = \mathbb{E}_P[(X - f_{1,P}^*(Z))(Y - f_{2,P}^*(Z))]$ using $\hat{g}_1^{(m)}$, $\hat{g}_2^{(m)}$, and $\mathcal{D}_{te}^{(n)}$.

In the GCM test, we reject $H_0 : X \perp\!\!\!\perp Y \mid Z$ if the statistic $\Gamma^{(n,m)} \triangleq |\sqrt{n}\bar{T}^{(n,m)}/\hat{\sigma}^{(n,m)}|$ exceeds $\tau_{\alpha/2} \triangleq \Phi^{-1}(1 - \alpha/2)$, depending on the test significance level $\alpha \in (0, 1)$. Here, $\bar{T}^{(n,m)}$ and $\hat{\sigma}^{(n,m)}$ are defined as in 3.1 with $T_i^{(m)} \triangleq (X_i - \hat{g}_1^{(m)}(Z_i))(Y_i - \hat{g}_2^{(m)}(Z_i))$. If the $p$-value is defined as $p(\mathcal{D}_{te}^{(n)}, \mathcal{D}_{tr}^{(m)}) = 2(1 - \Phi(\Gamma^{(n,m)}))$, the test $\varphi_\alpha^{\text{GCM}}(\mathcal{D}_{te}^{(n)}, \mathcal{D}_{tr}^{(m)})$ is analogously given by Equation 3.2. Like the STFR, the GCM test depends on the models' classes and implicitly on the training algorithm. If the limiting models $g_{1,P}^*$ and $g_{2,P}^*$ are not $f_{1,P}^*$ and $f_{2,P}^*$, then Type-I error control is not guaranteed.

We introduce definitions and assumptions. Assumption A.3 gives a rate of convergence for the models $\hat{g}_j^{(m)}$ in the mean squared error sense. Definition A.4 gives a definition for the misspecification gaps.

**Assumption A.3.** *There are functions $g_{1,P}^*$, $g_{2,P}^*$, and a constant $\gamma > 0$ such that*

$$\mathbb{E}_P\left[(\hat{g}_j^{(m)}(Z) - g_{j,P}^*(Z))^2 \mid \mathcal{D}_{tr}^{(m)}\right] = \mathcal{O}_\mathcal{P}(m^{-\gamma}), \text{ for } j = 1, 2$$

**Definition A.4.** *For each $j \in \{1, 2\}$, define the misspecification gap as $\delta_{j,P} \triangleq g_{j,P}^* - f_{j,P}^*$.*

In the next result, we approximate GCM test Type-I error rate and power using the gaps in Definition A.4 and Assumptions A.3, 3.2, and 3.3 applied to this context.

**Theorem A.5.** *Suppose that Assumptions 3.2, 3.3, and A.3 hold. If $n$ is a function of $m$ such that $n \to \infty$ and $n = o(m^\gamma)$ as $m \to \infty$, then*

$$\mathbb{E}_P[\varphi_\alpha^{\text{GCM}}(\mathcal{D}_{te}^{(n)}, \mathcal{D}_{tr}^{(m)})] = 1 - \Phi\left(\tau_{\alpha/2} - \sqrt{\frac{n}{\sigma_P^2}}\Omega_P^{\text{GCM}}\right) + \Phi\left(-\tau_{\alpha/2} - \sqrt{\frac{n}{\sigma_P^2}}\Omega_P^{\text{GCM}}\right) + o(1)$$

*where $o(1)$ denotes uniform convergence over all $P \in \mathcal{P}$ as $m \to \infty$ and*

$$\Omega_P^{\text{GCM}} \triangleq \mathbb{E}_P[\text{Cov}_P(X, Y \mid Z)] + \mathbb{E}_P[\delta_{1,P}(Z)\delta_{2,P}(Z)]$$

From Theorem A.5, it is possible to verify that if $\delta_{j,P}(Z)$ is zero for at least one $j \in \{1, 2\}$, *i.e.*, if at least one model converges to the conditional expectation, the GCM test asymptotically controls Type-I error. This can be seen as a *double-robustness* property of the GCM, which is not present[13] in Shah and Peters [31]. If $\mathbb{E}_P[\delta_{1,P}(Z)\delta_{2,P}(Z)] \neq 0$, then $\mathbb{E}_P[\varphi_\alpha^{\text{GCM}}(\mathcal{D}_{te}^{(n)}, \mathcal{D}_{tr}^{(m)})] \to 1$ as $m \to \infty$ even when $H_0 : X \perp\!\!\!\perp Y \mid Z$. Under the alternative, if $\Omega_P^{\text{GCM}} \neq 0$, Type-II error approaches 0 asymptotically.

### A.4 REgression with Subsequent Independence Test (RESIT)

As revisited by Zhang et al. [42], the idea behind RESIT is to first residualize $Y$ and $X$ given $Z$ and then test dependence between the residuals. It is similar to GCM, but requires the error terms and $Z$ to be independent. When that assumption is reasonable, one advantage of RESIT over GCM is that it has power against a broader set of alternatives. In this section, we use a permutation test [18, Example 15.2.3] to assess the independence of residuals. We analyse RESIT's Type-I error control.

If $(X, Y, Z) \sim P$ and $(X, Y)$ can be modeled as an additive noise model (ANM), that is, we can write

$$X = f_{1,P}^*(Z) + \epsilon \quad \text{and} \quad Y = f_{2,P}^*(Z) + \eta, \tag{A.1}$$

where $f_{1,P}^*(Z) = \mathbb{E}_P[X \mid Z]$, $f_{2,P}^*(Z) = \mathbb{E}_P[Y \mid Z]$, and the error terms $(\epsilon, \eta)$ are independent of $Z$, it is possible to show that $X \perp\!\!\!\perp Y \mid Z \Leftrightarrow \epsilon \perp\!\!\!\perp \eta$. To facilitate our analysis[14], we consider first fitting two models $\hat{g}_1^{(m)}$ and $\hat{g}_2^{(m)}$ that approximate $f_{1,P}^*$ and $f_{2,P}^*$ using the training set $\mathcal{D}_{tr}^{(m)}$ and then test the independence of the residuals[15] $\hat{\epsilon}_i = X_i - \hat{g}_1^{(m)}(Z_i)$ and $\hat{\eta}_i = Y_i - \hat{g}_2^{(m)}(Z_i)$ using the test set $\mathcal{D}_{te}^{(n)}$. Define (i) $(\hat{\boldsymbol{\epsilon}}, \hat{\boldsymbol{\eta}}) \triangleq \{(\hat{\epsilon}_i, \hat{\eta}_i)\}_{i=1}^n$ (test set residuals vertically stacked in matrix form) and (ii) $(\hat{\boldsymbol{\epsilon}}, \hat{\boldsymbol{\eta}})^{(b)}$

---

[13]This property is clear in our result because we consider data splitting.

[14]In practice, data splitting is not necessary. However, this procedure helps when theoretically analyzing the method.

[15]We omit the residuals superscript to ease notation.

as one of the $B$ permutations, *i.e.*, consider that we fix $\hat{\epsilon}$ and permute $\hat{\eta}$ row-wise. Let $\Psi$ be a test statistic and $\Psi((\hat{\epsilon}, \hat{\eta}))$ and $\Psi((\hat{\epsilon}, \hat{\eta})^{(b)})$ its evaluation on the original residuals and the $b$-the permuted set. If the permutation $p$-value is given by

$$p(\mathcal{D}_{te}^{(n)}, \mathcal{D}_{tr}^{(m)}) = \frac{1 + \sum_{b=1}^{B} \mathbb{1}[\Psi((\hat{\epsilon}, \hat{\eta})^{(b)}) \geq \Psi((\hat{\epsilon}, \hat{\eta}))]}{1 + B} \tag{A.2}$$

a test $\varphi_\alpha^{\text{RESIT}}$ aiming level $\alpha \in (0, 1)$ is given by Equation 3.2.

Similarly to STFR and GCM, we consider $g_{1,P}^*$ and $g_{2,P}^*$ to be the limiting models for $\hat{g}_1^{(m)}$ and $\hat{g}_2^{(m)}$. Different from GCM, both models $g_{1,P}^*$ and $g_{2,P}^*$ are multi-output since $X$ and $Y$ are not necessarily univariate.

Now, we introduce some assumptions before we present our result for RESIT. Assumption A.6 gives a rate of convergence for the models $\hat{g}_j^{(m)}$ in the mean squared error sense.

**Assumption A.6.** *There are models $g_{1,P}^*$, $g_{2,P}^*$, and a constant $\gamma > 0$ such that*

$$\mathbb{E}_P\left[\left\|\hat{g}_j^{(m)}(Z) - g_{j,P}^*(Z)\right\|_2^2 \,\middle|\, \mathcal{D}_{tr}^{(m)}\right] = \mathcal{O}_{\mathcal{P}_0}(m^{-\gamma}), \quad j = 1, 2$$

Assumption A.7 puts more structure on the distributions of the error terms $(\epsilon, \eta)$ and is a mild assumption.

**Assumption A.7.** *Assume that for all $P \in \mathcal{P}_0$, the distribution of $(\epsilon, \eta)$, $P_{\epsilon,\eta}$, is absolutely continuous with respect to the Lebesgue measure in $\mathbb{R}^{d_X} \times \mathbb{R}^{d_Y}$ with $L$-Lipschitz density $p_{\epsilon,\eta}$ for a certain $L > 0$. That is, for any $e_1, e_2 \in \mathbb{R}^{d_X}$ and $h_1, h_2 \in \mathbb{R}^{d_Y}$, we have*

$$|p_{\epsilon,\eta}(e_1, h_1) - p_{\epsilon,\eta}(e_2, h_2)| \leq L \|(e_1, h_1) - (e_2, h_2)\|_2$$

*We assume that $L$ does not depend on $P$.*

Assumption A.8 states that some of the variables we work with are uniformly almost surely bounded over all $P \in \mathcal{P}_0$. This assumption is realistic in most practical cases.

**Assumption A.8.** *There is bounded Borel set $A \in \mathcal{B}(\mathbb{R}^{d_X \times d_Y})$ such that*

$$\inf_{P \in \mathcal{P}_0} \mathbb{P}_P\left((X, Y) \in A\right) = 1$$

*and*

$$\inf_{P \in \mathcal{P}_0} \inf_{g_1, g_2} \mathbb{P}_P\left((g_1(Z), g_2(Z)) \in A\right) = 1$$

Here, $\inf_{g_1, g_2}$ is taken over the model classes we consider (if the model classes vary with $m$, consider the union of model classes). In the following, we present the result for RESIT. For that result, let: (i) $\epsilon^* \triangleq \epsilon - \delta_{1,P}(Z_i)$ and $\eta^* \triangleq \eta - \delta_{2,P}(Z_i)$, where the misspecification gaps are given as in Definition A.4; (ii) $d_{\text{TV}}$ represent the total variation (TV) distance between two probability distributions [36]; and (iii) the superscript $n$, *e.g.*, in $P_{\epsilon^*,\eta^*}^n$, represent a product measure.

**Theorem A.9.** *Under Assumptions A.6, A.7, and A.8, if $H_0 : X \perp\!\!\!\perp Y \mid Z$ holds and $n$ is a function of $m$ such that $n \to \infty$ and $n = o(m^{\frac{\gamma}{2}})$ as $m \to \infty$, then*

$$\mathbb{E}_P[\varphi_\alpha^{\text{RESIT}}(\mathcal{D}_{te}^{(n)}, \mathcal{D}_{tr}^{(m)})] \leq \alpha + \min\{d_{\text{TV}}(P_{\epsilon^*,\eta^*}^n, P_{\epsilon,\eta^*}^n), d_{\text{TV}}(P_{\epsilon^*,\eta^*}^n, P_{\epsilon^*,\eta}^n)\} + o(1)$$

*where $o(1)$ denotes uniform convergence over all $P \in \mathcal{P}_0$ as $m \to \infty$.*

From Theorem A.9, we can see that if at least one of the misspecification gaps $\delta_{1,P}$ or $\delta_{2,P}$ is null, then $\mathbb{E}_P[\varphi_\alpha^{\text{RESIT}}(\mathcal{D}_{te}^{(n)}, \mathcal{D}_{tr}^{(m)})] \leq \alpha$ under $H_0 : X \perp\!\!\!\perp Y \mid Z$. This can be seen as a *double-robustness* property of the RESIT. If none of the misspecification gaps are null, we do not have any guarantees on RESIT's Type-I error control. It can be shown that the proposed upper bound converges to 1.

## A.5 RBPT extra derivation

Let $\mu_X$ be a dominating measure of $Q^*_{X|Z}$ and $P_{X|Z}$ that does not depend on $Z$. Let $q^*_{X|Z}$ (resp. $p_{X|Z}$) be $Q^*_{X|Z}$ (resp. $P_{X|Z}$) density with respect to $\mu_X$.

$$
\begin{aligned}
\Omega^{\text{RBPT}}_{P,1} &= \mathbb{E}_P\left[\ell\left(\int g^*_P(x,Z)dQ^*_{X|Z}(x),Y\right) - \ell\left(\int g^*_P(x,Z)dP_{X|Z}(x),Y\right)\right] \\
&\leq L \cdot \mathbb{E}_P\left\|\int g^*_P(x,Z)dQ^*_{X|Z}(x) - \int g^*_P(x,Z)dP_{X|Z}(x)\right\|_2 \\
&\leq L \cdot \mathbb{E}_P\left\|\int g^*_P(x,Z)dQ^*_{X|Z}(x) - \int g^*_P(x,Z)dP_{X|Z}(x)\right\|_1 \\
&= L \cdot \mathbb{E}_P\left\|\int g^*_P(x,Z)(q^*_{X|Z}(x|Z) - p_{X|Z}(x|Z))d\mu_X(x)\right\|_1 \\
&\leq L \cdot \mathbb{E}_P\int \|g^*_P(x,Z)\|_1 |q^*_{X|Z}(x|Z) - p_{X|Z}(x|Z)|d\mu_X(x) \\
&\leq ML \cdot \mathbb{E}_P\int |q^*_{X|Z}(x|Z) - p_{X|Z}(x|Z)|d\mu_X(x) \\
&= 2ML \cdot \mathbb{E}_P[d_{\text{TV}}(Q^*_{X|Z}, P_{X|Z})]
\end{aligned}
$$

If

$$
\mathbb{E}_P[d_{\text{TV}}(Q^*_{X|Z}, P_{X|Z})] \leq \Omega^{\text{RBPT}}_{P,2}/(2ML)
$$

then

$$
\Omega^{\text{RBPT}}_P \leq 0
$$

## A.6 How to obtain a result for RBPT2?

We informally give some ideas on extending Theorem 4.4. Theorem 4.4 states that

$$
\mathbb{E}_P[\varphi^{\text{RBPT}}_\alpha(\mathcal{D}^{(n)}_{te}, \mathcal{D}^{(m)}_{tr})] = 1 - \Phi\left(\tau_\alpha - \sqrt{\frac{n}{\sigma^2_P}}\Omega^{\text{RBPT}}_P\right) + o(1)
$$

where $\Omega^{\text{RBPT}}_P = \Omega^{\text{RBPT}}_{P,1} - \Omega^{\text{RBPT}}_{P,2}$ with

$$
\Omega^{\text{RBPT}}_{P,1} \triangleq \mathbb{E}_P\left[\ell\left(\int g^*_P(x,Z)dQ^*_{X|Z}(x),Y\right)\right] - \mathbb{E}_P\left[\ell\left(\int g^*_P(x,Z)dP_{X|Z}(x),Y\right)\right]
$$

and

$$
\Omega^{\text{RBPT}}_{P,2} \triangleq \mathbb{E}_P\left[\ell(g^*_P(X,Z),Y)\right] - \mathbb{E}_P\left[\ell\left(\int g^*_P(x,Z)dP_{X|Z}(x),Y\right)\right]
$$

If we wanted to adapt that result for RBPT2, we would have to redefine $\Omega^{\text{RBPT}}_{P,1}$. The analogue of $\Omega^{\text{RBPT}}_{P,1}$ for RBPT2 would be

$$
\Omega^{\text{RBPT2}}_{P,1} \triangleq \mathbb{E}_P\left[\ell\left(\tilde{\mathbb{E}}_P\left[g^*_P(X,Z) \mid Z\right],Y\right)\right] - \mathbb{E}_P\left[\ell\left(\mathbb{E}_P\left[g^*_P(X,Z) \mid Z\right],Y\right)\right]
$$

where $\tilde{\mathbb{E}}_P\left[g^*_P(X,Z) \mid Z = z\right]$ denotes the limiting regression model to predict $g^*_P(X,Z)$ given $Z$. If we assume the existence of a big unlabeled dataset, deriving a result might be easier as we can avoid the asymptotic details on the convergence of the second regression model by assuming that the limiting Rao-Blackwellization model, for a fixed initial predictor, is known. The only challenge is proving the convergence of $\tilde{\mathbb{E}}_P\left[\hat{g}(X,Z) \mid Z\right]$ to $\tilde{\mathbb{E}}_P\left[g^*_P(X,Z) \mid Z\right]$.

## B Technical proofs

### B.1 STFR

**Lemma B.1.** *Assume we are under the conditions of Theorem 3.4. Then:*

$$(\hat{\sigma}^{(n,m)})^2 - \mathsf{Var}_P[T_1^{(m)} \mid \mathcal{D}_{tr}^{(m)}] = o_{\mathcal{P}}(1) \text{ as } m \to \infty$$

*Proof.* First, see that for an arbitrary $\varepsilon > 0$, there must be[16] a sequence of probability measures in $\mathcal{P}$, $(P^{(m)})_{m \in \mathbb{N}}$, such that

$$\sup_{P \in \mathcal{P}} \mathbb{P}_P \left[ \left| (\hat{\sigma}^{(n,m)})^2 - \mathsf{Var}_P[T_1^{(m)} \mid \mathcal{D}_{tr}^{(m)}] \right| > \varepsilon \right] \leq \mathbb{P}_{P^{(m)}} \left[ \left| (\hat{\sigma}^{(n,m)})^2 - \mathsf{Var}_{P^{(m)}}[T_1^{(m)} \mid \mathcal{D}_{tr}^{(m)}] \right| > \varepsilon \right] + \frac{1}{m}$$

Pick one of such sequences. Then, to prove that $(\hat{\sigma}^{(n,m)})^2 - \mathsf{Var}_P[T_1^{(m)} \mid \mathcal{D}_{tr}^{(m)}] = o_{\mathcal{P}}(1)$ as $m \to \infty$, it suffices to show that

$$\mathbb{P}_{P^{(m)}} \left[ \left| (\hat{\sigma}^{(n,m)})^2 - \mathsf{Var}_{P^{(m)}}[T_1^{(m)} \mid \mathcal{D}_{tr}^{(m)}] \right| > \varepsilon \right] \to 0 \text{ as } m \to \infty$$

Now, expanding $(\hat{\sigma}^{(n,m)})^2$ we get

$$
\begin{aligned}
(\hat{\sigma}^{(n,m)})^2 &= \frac{1}{n} \sum_{i=1}^n (T_i^{(m)})^2 - \left( \frac{1}{n} \sum_{i=1}^n T_i^{(m)} \right)^2 \\
&= \frac{1}{n} \sum_{i=1}^n \left[ (T_i^{(m)})^2 - \mathbb{E}_{P^{(m)}}[(T_1^{(m)})^2 \mid \mathcal{D}_{tr}^{(m)}] \right] + \mathbb{E}_{P^{(m)}}[(T_1^{(m)})^2 \mid \mathcal{D}_{tr}^{(m)}] - \left( \frac{1}{n} \sum_{i=1}^n T_i^{(m)} \right)^2 \\
&= \frac{1}{n} \sum_{i=1}^n \left[ (T_i^{(m)})^2 - \mathbb{E}_{P^{(m)}}[(T_1^{(m)})^2 \mid \mathcal{D}_{tr}^{(m)}] \right] - \left( \frac{1}{n} \sum_{i=1}^n T_i^{(m)} - \mathbb{E}_{P^{(m)}}[T_1^{(m)} \mid \mathcal{D}_{tr}^{(m)}] \right)^2 \\
&\quad - 2 \left( \frac{1}{n} \sum_{i=1}^n T_i^{(m)} \mathbb{E}_{P^{(m)}}[T_1^{(m)} \mid \mathcal{D}_{tr}^{(m)}] - (\mathbb{E}_{P^{(m)}}[T_1^{(m)} \mid \mathcal{D}_{tr}^{(m)}])^2 \right) + \mathsf{Var}_{P^{(m)}}[T_1^{(m)} \mid \mathcal{D}_{tr}^{(m)}] \\
&= \frac{1}{n} \sum_{i=1}^n \left[ (T_i^{(m)})^2 - \mathbb{E}_{P^{(m)}}[(T_1^{(m)})^2 \mid \mathcal{D}_{tr}^{(m)}] \right] - \left( \frac{1}{n} \sum_{i=1}^n T_i^{(m)} - \mathbb{E}_{P^{(m)}}[T_1^{(m)} \mid \mathcal{D}_{tr}^{(m)}] \right)^2 \\
&\quad - 2\mathbb{E}_{P^{(m)}}[T_1^{(m)} \mid \mathcal{D}_{tr}^{(m)}] \left( \frac{1}{n} \sum_{i=1}^n T_i^{(m)} - \mathbb{E}_{P^{(m)}}[T_1^{(m)} \mid \mathcal{D}_{tr}^{(m)}] \right) + \mathsf{Var}_{P^{(m)}}[T_1^{(m)} \mid \mathcal{D}_{tr}^{(m)}]
\end{aligned}
$$

Then,

$$
\begin{aligned}
(\hat{\sigma}^{(n,m)})^2 &- \mathsf{Var}_{P^{(m)}}[T_1^{(m)} \mid \mathcal{D}_{tr}^{(m)}] = \\
&= \frac{1}{n} \sum_{i=1}^n \left[ (T_i^{(m)})^2 - \mathbb{E}_{P^{(m)}}[(T_1^{(m)})^2 \mid \mathcal{D}_{tr}^{(m)}] \right] - \left( \frac{1}{n} \sum_{i=1}^n T_i^{(m)} - \mathbb{E}_{P^{(m)}}[T_1^{(m)} \mid \mathcal{D}_{tr}^{(m)}] \right)^2 \\
&\quad - 2\mathbb{E}_{P^{(m)}}[T_1^{(m)} \mid \mathcal{D}_{tr}^{(m)}] \left( \frac{1}{n} \sum_{i=1}^n T_i^{(m)} - \mathbb{E}_{P^{(m)}}[T_1^{(m)} \mid \mathcal{D}_{tr}^{(m)}] \right)
\end{aligned}
$$

Using a law of large numbers for triangular arrays [6, Corollary 9.5.6] (we comment on needed conditions to use this result below) and the continuous mapping theorem, we have that

- $\frac{1}{n} \sum_{i=1}^n \left[ (T_i^{(m)})^2 - \mathbb{E}_{P^{(m)}}[(T_1^{(m)})^2 \mid \mathcal{D}_{tr}^{(m)}] \right] = o_p(1)$

- $\left( \frac{1}{n} \sum_{i=1}^n T_i^{(m)} - \mathbb{E}_{P^{(m)}}[T_1^{(m)} \mid \mathcal{D}_{tr}^{(m)}] \right)^2 = o_p(1)$

- $\underbrace{2\mathbb{E}_{P^{(m)}}[T_1^{(m)} \mid \mathcal{D}_{tr}^{(m)}]}_{\mathcal{O}_p(1) \text{ (Assumption 3.2)}} \left( \frac{1}{n} \sum_{i=1}^n T_i^{(m)} - \mathbb{E}_{P^{(m)}}[T_1^{(m)} \mid \mathcal{D}_{tr}^{(m)}] \right) = o_p(1)$

---

[16]Because of the definition of sup.

and then

$$\sup_{P \in \mathcal{P}} \mathbb{P}_P \left[ \left| (\hat{\sigma}^{(n,m)})^2 - \mathsf{Var}_P[T_1^{(m)} \mid \mathcal{D}_{tr}^{(m)}] \right| > \varepsilon \right] \leq \mathbb{P}_{P^{(m)}} \left[ \left| (\hat{\sigma}^{(n,m)})^2 - \mathsf{Var}_{P^{(m)}}[T_1^{(m)} \mid \mathcal{D}_{tr}^{(m)}] \right| > \varepsilon \right] + \frac{1}{m} \to 0$$

as $m \to \infty$, i.e.,

$$(\hat{\sigma}^{(n,m)})^2 - \mathsf{Var}_P[T_1^{(m)} \mid \mathcal{D}_{tr}^{(m)}] = o_{\mathcal{P}}(1) \text{ as } m \to \infty$$

*Conditions to use the law of large numbers.* Let $(P^{(m)})_{m \in \mathbb{N}}$ be an arbitrary sequence of probability measures in $\mathcal{P}$. Define our triangular arrays as $\left\{ V_{i,1}^{(m)} \right\}_{1 \leq i \leq n}$ and $\left\{ V_{i,2}^{(m)} \right\}_{1 \leq i \leq n}$, where $V_{i,1}^{(m)} \triangleq T_i^{(m)} - \mathbb{E}_{P^{(m)}}[T_1^{(m)} \mid \mathcal{D}_{tr}^{(m)}]$ and $V_{i,2}^{(m)} \triangleq (T_i^{(m)})^2 - \mathbb{E}_{P^{(m)}}[(T_1^{(m)})^2 \mid \mathcal{D}_{tr}^{(m)}]$. Now, we comment on the conditions for the law of large numbers [6, Corollary 9.5.6]:

1. This condition naturally applies by definition and because of Assumption 3.2.

2. From Assumption 3.2 and Resnick [27, Example 6.5.2],

$$\mathbb{E}_{P^{(m)}} \left[ \left| V_{i,1}^{(m)} \right| \mid \mathcal{D}_{tr}^{(m)} \right] \leq \mathbb{E}_{P^{(m)}} \left[ \left| T_i^{(m)} \right| \mid \mathcal{D}_{tr}^{(m)} \right] + \left| \mathbb{E}_{P^{(m)}}[T_1^{(m)} \mid \mathcal{D}_{tr}^{(m)}] \right| \leq 2\mathbb{E}_{P^{(m)}} \left[ \left| T_i^{(m)} \right| \mid \mathcal{D}_{tr}^{(m)} \right] = \mathcal{O}_p(1)$$

and

$$\mathbb{E}_{P^{(m)}} \left[ \left| V_{i,2}^{(m)} \right| \mid \mathcal{D}_{tr}^{(m)} \right] \leq 2\mathbb{E}_{P^{(m)}} \left[ (T_i^{(m)})^2 \mid \mathcal{D}_{tr}^{(m)} \right] = \mathcal{O}_p(1)$$

3. Fix any $\epsilon > 0$ and let $k$ be defined as in Assumption 3.2. See that

$$
\begin{aligned}
\mathbb{E}_{P^{(m)}} \left[ \left| V_{i,1}^{(m)} \right| \mathbb{1}[|V_{i,1}^{(m)}| \geq \epsilon n] \mid \mathcal{D}_{tr}^{(m)} \right] &= \mathbb{E}_{P^{(m)}} \left[ \left| V_{i,1}^{(m)} \right| \mathbb{1}[(|V_{i,1}^{(m)}|/(\epsilon n))^k \geq 1] \mid \mathcal{D}_{tr}^{(m)} \right] \\
&\leq \frac{1}{(n\epsilon)^k} \mathbb{E}_{P^{(m)}} \left[ |V_{i,1}^{(m)}|^{1+k} \mid \mathcal{D}_{tr}^{(m)} \right] \\
&= \frac{1}{(n\epsilon)^k} \mathbb{E}_{P^{(m)}} \left[ |T_i^{(m)} - \mathbb{E}_{P^{(m)}}[T_1^{(m)} \mid \mathcal{D}_{tr}^{(m)}]|^{1+k} \mid \mathcal{D}_{tr}^{(m)} \right] \\
&\leq \frac{1}{(n\epsilon)^k} \left\{ \mathbb{E}_{P^{(m)}} \left[ |T_i^{(m)}|^{1+k} \mid \mathcal{D}_{tr}^{(m)} \right]^{\frac{1}{1+k}} + \mathbb{E}_{P^{(m)}} \left[ |T_i^{(m)}| \mid \mathcal{D}_{tr}^{(m)} \right] \right\}^{1+k} \\
&= \frac{1}{(n\epsilon)^k} \mathcal{O}_p(1) \\
&= o_p(1)
\end{aligned}
$$

where the third inequality is obtained via Minkowski Inequality [27] and the fifth step is an application of Assumption 3.2 and Resnick [27, Example 6.5.2]. Analogously, define $k' = k/2$ and see that

$$
\begin{aligned}
\mathbb{E}_{P^{(m)}} \left[ \left| V_{i,2}^{(m)} \right| \mathbb{1}[|V_{i,2}^{(m)}| \geq \epsilon n] \mid \mathcal{D}_{tr}^{(m)} \right] &= \mathbb{E}_{P^{(m)}} \left[ \left| V_{i,2}^{(m)} \right| \mathbb{1}[(|V_{i,2}^{(m)}|/(\epsilon n))^{k'} \geq 1] \mid \mathcal{D}_{tr}^{(m)} \right] \\
&\leq \frac{1}{(n\epsilon)^{k'}} \mathbb{E}_{P^{(m)}} \left[ |V_{i,2}^{(m)}|^{1+k'} \mid \mathcal{D}_{tr}^{(m)} \right] \\
&= \frac{1}{(n\epsilon)^{k'}} \mathbb{E}_{P^{(m)}} \left[ |(T_i^{(m)})^2 - \mathbb{E}_{P^{(m)}}[(T_1^{(m)})^2 \mid \mathcal{D}_{tr}^{(m)}]|^{1+k'} \mid \mathcal{D}_{tr}^{(m)} \right] \\
&\leq \frac{1}{(n\epsilon)^{k'}} \left\{ \mathbb{E}_{P^{(m)}} \left[ (T_i^{(m)})^{2(1+k')} \mid \mathcal{D}_{tr}^{(m)} \right]^{\frac{1}{1+k'}} + \mathbb{E}_{P^{(m)}} \left[ (T_i^{(m)})^2 \mid \mathcal{D}_{tr}^{(m)} \right] \right\}^{1+k'} \\
&= \frac{1}{(n\epsilon)^{k'}} \left\{ \mathbb{E}_{P^{(m)}} \left[ (T_i^{(m)})^{2+k} \mid \mathcal{D}_{tr}^{(m)} \right]^{\frac{1}{1+k'}} + \mathbb{E}_{P^{(m)}} \left[ (T_i^{(m)})^2 \mid \mathcal{D}_{tr}^{(m)} \right] \right\}^{1+k'} \\
&= \frac{1}{(n\epsilon)^{k'}} \mathcal{O}_p(1) \\
&= o_p(1)
\end{aligned}
$$

$\square$

**Theorem 3.4.** Suppose that Assumptions 3.1, 3.2, and 3.3 hold. If $n$ is a function of $m$ such that $n \to \infty$ and $n = o(m^{2\gamma})$ as $m \to \infty$, then

$$\mathbb{E}_P[\varphi_\alpha^{\text{STFR}}(\mathcal{D}_{te}^{(n)}, \mathcal{D}_{tr}^{(m)})] = 1 - \Phi\left(\tau_\alpha - \sqrt{\frac{n}{\sigma_P^2}}\Omega_P^{\text{STFR}}\right) + o(1)$$

where $o(1)$ denotes uniform convergence over all $P \in \mathcal{P}$ as $m \to \infty$ and

$$\Omega_P^{\text{STFR}} \triangleq \mathbb{E}_P[\ell(g_{2,P}^*(Z), Y)] - \mathbb{E}_P[\ell(g_{1,P}^*(X, Z), Y)]$$

*Proof.* First, note that there must be[17] a sequence of probability measures in $\mathcal{P}$, $(P^{(m)})_{m \in \mathbb{N}}$, such that

$$\sup_{P \in \mathcal{P}} \left| \mathbb{E}_P[\varphi_\alpha^{\text{STFR}}(\mathcal{D}_{te}^{(n)}, \mathcal{D}_{tr}^{(m)})] - 1 + \Phi\left(\tau_\alpha - \sqrt{\frac{n}{\sigma_P^2}}\Omega_P^{\text{STFR}}\right) \right| \leq \left| \mathbb{E}_{P^{(m)}}[\varphi_\alpha^{\text{STFR}}(\mathcal{D}_{te}^{(n)}, \mathcal{D}_{tr}^{(m)})] - 1 + \Phi\left(\tau_\alpha - \sqrt{\frac{n}{\sigma_{P^{(m)}}^2}}\Omega_{P^{(m)}}^{\text{STFR}}\right) \right| + \frac{1}{m}$$

Then, it suffices to show that the RHS vanishes when we consider such a sequence $(P^{(m)})_{m \in \mathbb{N}}$.

Now, let us first decompose the test statistic $\Lambda^{(n,m)}$ in the following way:

$$\Lambda^{(n,m)} \triangleq$$

$$\triangleq \frac{\sqrt{n}\bar{T}^{(n,m)}}{\hat{\sigma}^{(n,m)}}$$

$$= \frac{\sqrt{n}\left(\bar{T}^{(n,m)} - \mathbb{E}_{P^{(m)}}[T_1^{(m)} \mid \mathcal{D}_{tr}^{(m)}]\right)}{\hat{\sigma}^{(n,m)}} + \frac{\sqrt{n}\mathbb{E}_{P^{(m)}}[T_1^{(m)} \mid \mathcal{D}_{tr}^{(m)}]}{\hat{\sigma}^{(n,m)}}$$

$$= \frac{\sqrt{n}\left(\bar{T}^{(n,m)} - \mathbb{E}_{P^{(m)}}[T_1^{(m)} \mid \mathcal{D}_{tr}^{(m)}]\right)}{\hat{\sigma}^{(n,m)}} +$$

$$+ \frac{\sqrt{n}\left(\mathbb{E}_{P^{(m)}}[\ell(\hat{g}_2^{(m)}(Z_1), Y_1) - \ell(g_{2,P^{(m)}}^*(Z_1), Y_1) \mid \mathcal{D}_{tr}^{(m)}] - \left(\mathbb{E}_{P^{(m)}}[\ell(\hat{g}_1^{(m)}(X_1, Z_1), Y_1) - \ell(g_{1,P^{(m)}}^*(X_1, Z_1), Y_1) \mid \mathcal{D}_{tr}^{(m)}]\right)\right)}{\hat{\sigma}^{(n,m)}}$$

$$+ \frac{\sqrt{n}\left(\mathbb{E}_{P^{(m)}}[\ell(g_{2,P^{(m)}}^*(Z_1), Y_1)] - \mathbb{E}_{P^{(m)}}[\ell(g_{1,P^{(m)}}^*(X_1, Z_1), Y_1)]\right)}{\hat{\sigma}^{(n,m)}}$$

$$= \frac{\sqrt{n}W_{1,P^{(m)}}^{(m)}}{\hat{\sigma}^{(n,m)}} + \frac{\sqrt{n}W_{2,P^{(m)}}^{(m)}}{\hat{\sigma}^{(n,m)}} + \frac{\sqrt{n}\Omega_{P^{(m)}}^{\text{STFR}}}{\hat{\sigma}^{(n,m)}}$$

Given that $n$ is a function of $m$, we omit it when writing the $W_{j,P^{(m)}}^{(m)}$'s. Define $\sigma_{P^{(m)}}^{(m)} \triangleq \sqrt{\text{Var}_{P^{(m)}}[T_1^{(m)} \mid \mathcal{D}_{tr}^{(m)}]}$ and see that

$$\mathbb{E}_{P^{(m)}}[\varphi_\alpha^{\text{STFR}}(\mathcal{D}_{te}^{(n)}, \mathcal{D}_{tr}^{(m)})] = \mathbb{P}_{P^{(m)}}[p(\mathcal{D}_{te}^{(n)}, \mathcal{D}_{tr}^{(m)}) \leq \alpha]$$

$$= \mathbb{P}_{P^{(m)}}\left[1 - \Phi\left(\Lambda^{(n,m)}\right) \leq \alpha\right]$$

$$= \mathbb{P}_{P^{(m)}}\left[\Lambda^{(n,m)} \geq \tau_\alpha\right]$$

$$= \mathbb{P}_{P^{(m)}}\left[\frac{\sqrt{n}W_{1,P^{(m)}}^{(m)}}{\hat{\sigma}^{(n,m)}} + \frac{\sqrt{n}W_{2,P^{(m)}}^{(m)}}{\hat{\sigma}^{(n,m)}} + \frac{\sqrt{n}\Omega_{P^{(m)}}^{\text{STFR}}}{\hat{\sigma}^{(n,m)}} \geq \tau_\alpha\right]$$

$$= \mathbb{P}_{P^{(m)}}\left[\frac{\sqrt{n}W_{1,P^{(m)}}^{(m)}}{\sigma_{P^{(m)}}} + \frac{\sqrt{n}W_{2,P^{(m)}}^{(m)}}{\sigma_{P^{(m)}}} + \frac{\sqrt{n}\Omega_{P^{(m)}}^{\text{STFR}}}{\sigma_{P^{(m)}}} + \tau_\alpha - \tau_\alpha \frac{\hat{\sigma}^{(n,m)}}{\sigma_{P^{(m)}}} \geq \tau_\alpha\right]$$

$$= \mathbb{P}_{P^{(m)}}\left[\frac{\sqrt{n}W_{1,P^{(m)}}^{(m)}}{\sigma_{P^{(m)}}^{(m)}} \frac{\sigma_{P^{(m)}}^{(m)}}{\sigma_{P^{(m)}}} + \frac{\sqrt{n}W_{2,P^{(m)}}^{(m)}}{\sigma_{P^{(m)}}} + \tau_\alpha - \tau_\alpha \frac{\hat{\sigma}^{(n,m)}}{\sigma_{P^{(m)}}^{(m)}} \frac{\sigma_{P^{(m)}}^{(m)}}{\sigma_{P^{(m)}}} \geq \tau_\alpha - \frac{\sqrt{n}\Omega_{P^{(m)}}^{\text{STFR}}}{\sigma_{P^{(m)}}}\right]$$

$$= 1 - \Phi\left(\tau_\alpha - \sqrt{\frac{n}{\sigma_{P^{(m)}}^2}}\Omega_{P^{(m)}}^{\text{STFR}}\right) + o(1) \qquad (\text{B.1})$$

---

[17]Because of the definition of sup.

Implying that

$$\sup_{P \in \mathcal{P}} \left| \mathbb{E}_P[\varphi_\alpha^{\text{STFR}}(\mathcal{D}_{te}^{(n)}, \mathcal{D}_{tr}^{(m)})] - 1 + \Phi\left(\tau_\alpha - \sqrt{\frac{n}{\sigma_P^2}} \Omega_P^{\text{STFR}}\right) \right| = o(1) \text{ as } m \to \infty$$

*Justifying step B.1.* First, from a central limit theorem for triangular arrays [6, Corollary 9.5.11], we have that

$$\sqrt{n}\left(\frac{W_{1,P^{(m)}}^{(m)}}{\sigma_{P^{(m)}}^{(m)}}\right) = \sqrt{n}\left(\frac{\bar{T}^{(n,m)} - \mathbb{E}_{P^{(m)}}[T_1^{(m)} \mid \mathcal{D}_{tr}^{(m)}]}{\sigma_{P^{(m)}}^{(m)}}\right) = \frac{1}{\sqrt{n}} \sum_{i=1}^n \left(\frac{T_i^{(m)} - \mathbb{E}_{P^{(m)}}[T_1^{(m)} \mid \mathcal{D}_{tr}^{(m)}]}{\sigma_{P^{(m)}}^{(m)}}\right) \Rightarrow N(0,1)$$

we comment on the conditions to use this theorem below.

Second, we have that

$$\frac{\sqrt{n}W_{1,P^{(m)}}^{(m)}}{\sigma_{P^{(m)}}^{(m)}} - \left(\frac{\sqrt{n}W_{1,P^{(m)}}^{(m)}}{\sigma_{P^{(m)}}^{(m)}} \frac{\sigma_{P^{(m)}}^{(m)}}{\sigma_{P^{(m)}}} + \frac{\sqrt{n}W_{2,P^{(m)}}^{(m)}}{\sigma_{P^{(m)}}} + \tau_\alpha - \tau_\alpha \frac{\hat{\sigma}^{(n,m)}}{\sigma_{P^{(m)}}^{(m)}} \frac{\sigma_{P^{(m)}}^{(m)}}{\sigma_{P^{(m)}}}\right) =$$

$$= \frac{\sqrt{n}W_{1,P^{(m)}}^{(m)}}{\sigma_{P^{(m)}}^{(m)}} \left(1 - \frac{\sigma_{P^{(m)}}^{(m)}}{\sigma_{P^{(m)}}}\right) - \frac{\sqrt{n}W_{2,P^{(m)}}^{(m)}}{\sigma_{P^{(m)}}} + \tau_\alpha\left(\frac{\hat{\sigma}^{(n,m)}}{\sigma_{P^{(m)}}^{(m)}} - 1\right)\left(\frac{\sigma_{P^{(m)}}^{(m)}}{\sigma_{P^{(m)}}} - 1\right) + \tau_\alpha\left(\frac{\sigma_{P^{(m)}}^{(m)}}{\sigma_{P^{(m)}}} - 1\right) + \tau_\alpha\left(\frac{\hat{\sigma}^{(n,m)}}{\sigma_{P^{(m)}}^{(m)}} - 1\right)$$

$$= o_p(1) \text{ as } m \to \infty$$

To see why the random quantity above converges to zero in probability, see that because of Assumption 3.3, Lemma B.1, and continuous mapping theorem, we have that

$$\frac{\hat{\sigma}^{(n,m)}}{\sigma_{P^{(m)}}^{(m)}} - 1 = o_p(1) \text{ and } \frac{\sigma_{P^{(m)}}^{(m)}}{\sigma_{P^{(m)}}} - 1 = o_p(1) \text{ as } m \to \infty$$

Additionally, because of Assumptions 3.1, 3.3 and condition $n = o(m^{2\gamma})$, we have that

$$\left|\frac{\sqrt{n}W_{2,P^{(m)}}^{(m)}}{\sigma_{P^{(m)}}}\right| = \left|\frac{o(m^\gamma)\mathcal{O}_p(m^{-\gamma})}{\sigma_{P^{(m)}}}\right| \leq \left|\frac{o(m^\gamma)\mathcal{O}_p(m^{-\gamma})}{\inf_{P \in \mathcal{P}} \sigma_P}\right| = o_p(1)$$

Finally,

$$\frac{\sqrt{n}W_{1,P^{(m)}}^{(m)}}{\sigma_{P^{(m)}}^{(m)}} \frac{\sigma_{P^{(m)}}^{(m)}}{\sigma_{P^{(m)}}} + \frac{\sqrt{n}W_{2,P^{(m)}}^{(m)}}{\sigma_{P^{(m)}}} + \tau_\alpha - \tau_\alpha \frac{\hat{\sigma}^{(n,m)}}{\sigma_{P^{(m)}}^{(m)}} \frac{\sigma_{P^{(m)}}^{(m)}}{\sigma_{P^{(m)}}} =$$

$$= \frac{\sqrt{n}W_{1,P^{(m)}}^{(m)}}{\sigma_{P^{(m)}}^{(m)}} - \left[\frac{\sqrt{n}W_{1,P^{(m)}}^{(m)}}{\sigma_{P^{(m)}}^{(m)}} - \left(\frac{\sqrt{n}W_{1,P^{(m)}}^{(m)}}{\sigma_{P^{(m)}}^{(m)}} \frac{\sigma_{P^{(m)}}^{(m)}}{\sigma_{P^{(m)}}} + \frac{\sqrt{n}W_{2,P^{(m)}}^{(m)}}{\sigma_{P^{(m)}}} + \tau_\alpha - \tau_\alpha \frac{\hat{\sigma}^{(n,m)}}{\sigma_{P^{(m)}}^{(m)}} \frac{\sigma_{P^{(m)}}^{(m)}}{\sigma_{P^{(m)}}}\right)\right]$$

$$= \frac{\sqrt{n}W_{1,P^{(m)}}^{(m)}}{\sigma_{P^{(m)}}^{(m)}} + o_p(1) \Rightarrow N(0,1)$$

by Slutsky's theorem. Because $N(0,1)$ is a continuous distribution, we have uniform convergence of the distribution function [27][Chapter 8, Exercise 5] and we do not have to worry about the fact that $\tau_\alpha - \frac{\sqrt{n}\Omega_{P^{(m)}}^{\text{STFR}}}{\sigma_{P^{(m)}}}$ depends on $m$.

*Conditions to apply the central limit theorem.* Now, we comment on the conditions for the central limit theorem [6, Corollary 9.5.11]. Define our triangular array as $\left\{V_i^{(m)}\right\}_{1 \leq i \leq n}$, where $V_i^{(m)} \triangleq \frac{T_i^{(m)} - \mathbb{E}_{P^{(m)}}[T_i^{(m)} \mid \mathcal{D}_{tr}^{(m)}]}{\sigma_{P^{(m)}}^{(m)}}$.

1. This condition naturally applies by definition and because of Assumption 3.2.

2. See that, for any $m$, we have that

$$\mathbb{E}_{P^{(m)}}[(V_i^{(m)})^2 \mid \mathcal{D}_{tr}^{(m)}] = 1$$

3. Fix any $\epsilon > 0$ and let $k$ be defined as in Assumption 3.2. See that

$$\mathbb{E}_{P^{(m)}}\left[(V_i^{(m)})^2 \mathbb{1}[|V_i^{(m)}| \geq \epsilon n] \mid \mathcal{D}_{tr}^{(m)}\right] =$$

$$= \mathbb{E}_{P^{(m)}}\left[(V_i^{(m)})^2 \mathbb{1}[(|V_i^{(m)}|/(\epsilon n))^k \geq 1] \mid \mathcal{D}_{tr}^{(m)}\right]$$

$$\leq \frac{1}{(n\epsilon)^k} \mathbb{E}_{P^{(m)}}\left[|V_i^{(m)}|^{2+k} \mid \mathcal{D}_{tr}^{(m)}\right]$$

$$= \frac{1}{(n\epsilon)^k} \mathbb{E}_{P^{(m)}}\left[\left|\frac{T_i^{(m)} - \mathbb{E}_{P^{(m)}}[T_i^{(m)} \mid \mathcal{D}_{tr}^{(m)}]}{\sigma_{P^{(m)}}^{(m)}}\right|^{2+k} \mid \mathcal{D}_{tr}^{(m)}\right]$$

$$\leq \frac{1}{(n\epsilon)^k} \frac{1}{(\sigma_{P^{(m)}}^{(m)})^{2+k}} \left\{\mathbb{E}_{P^{(m)}}\left[|T_i^{(m)}|^{2+k} \mid \mathcal{D}_{tr}^{(m)}\right]^{\frac{1}{2+k}} + \mathbb{E}_{P^{(m)}}\left[|T_i^{(m)}| \mid \mathcal{D}_{tr}^{(m)}\right]\right\}^{2+k}$$

$$= \frac{1}{(n\epsilon)^k} \left[\frac{1}{(\sigma_{P^{(m)}})^{2+k}} + o_p(1)\right] \left\{\mathbb{E}_{P^{(m)}}\left[|T_i^{(m)}|^{2+k} \mid \mathcal{D}_{tr}^{(m)}\right]^{\frac{1}{2+k}} + \mathbb{E}_{P^{(m)}}\left[|T_i^{(m)}| \mid \mathcal{D}_{tr}^{(m)}\right]\right\}^{2+k}$$

$$\leq \frac{1}{(n\epsilon)^k} \left[\frac{1}{(\inf_{P \in \mathcal{P}} \sigma_P)^{2+k}} + o_p(1)\right] \mathcal{O}_p(1)$$

$$= \frac{1}{(n\epsilon)^k} \mathcal{O}_p(1)$$

$$= o_p(1)$$

where the third inequality is obtained via Minkowski Inequality [27] and the last inequality is an application of Assumption 3.2 and Resnick [27, Example 6.5.2].

$\square$

**Corollary 3.6.** Suppose we are under the conditions of Theorem 3.4.

• (Type-I error) If $H_0 : X \perp\!\!\!\perp Y \mid Z$ holds, then

$$\mathbb{E}_P[\varphi_\alpha^{\text{STFR}}(\mathcal{D}_{te}^{(n)}, \mathcal{D}_{tr}^{(m)})] \leq 1 - \Phi\left(\tau_\alpha - \sqrt{\frac{n}{\sigma_P^2}}\Delta_{2,P}\right) + o(1)$$

where $o(1)$ denotes uniform convergence over all $P \in \mathcal{P}_0$ as $m \to \infty$.

• (Type-II error) In general, we have

$$1 - \mathbb{E}_P[\varphi_\alpha^{\text{STFR}}(\mathcal{D}_{te}^{(n)}, \mathcal{D}_{tr}^{(m)})] \leq \Phi\left(\tau_\alpha - \sqrt{\frac{n}{\sigma_P^2}}(\Delta_P - \Delta_{1,P})\right) + o(1)$$

where $o(1)$ denotes uniform convergence over all $P \in \mathcal{P}$ as $m \to \infty$ and $\Delta_P \triangleq \mathbb{E}_P[\ell(f_{2,P}^*(Z), Y)] - \mathbb{E}_P[\ell(f_{1,P}^*(X, Z), Y)]$.

*Proof.* Under the conditions of Theorem 3.4, we start proving that

1. $\Delta_P - \Delta_{1,P} \leq \Omega_P^{\text{STFR}}$ holds;

2. Under $H_0$, $\Omega_P^{\text{STFR}} \leq \Delta_{2,P}$ holds.

For (1), see that,

$$\begin{aligned}
\Omega_P^{\text{STFR}} &= \mathbb{E}_P[\ell(g_{2,P}^*(Z), Y)] - \mathbb{E}_P[\ell(g_{1,P}^*(X, Z), Y)] \\
&= \mathbb{E}_P[\ell(g_{2,P}^*(Z), Y)] - \mathbb{E}_P[\ell(f_{2,P}^*(Z), Y)] \quad\quad (\geq 0) \\
&\quad + \mathbb{E}_P[\ell(f_{1,P}^*(X, Z), Y)] - \mathbb{E}_P[\ell(g_{1,P}^*(X, Z), Y)] \\
&\quad + \mathbb{E}_P[\ell(f_{2,P}^*(Z), Y)] - \mathbb{E}_P[\ell(f_{1,P}^*(X, Z), Y)] \\
&\geq \mathbb{E}_P[\ell(f_{1,P}^*(X, Z), Y)] - \mathbb{E}_P[\ell(g_{1,P}^*(X, Z), Y)] + \Delta_P \\
&= \Delta_P - \Delta_{1,P}
\end{aligned}$$

Now, for (2):

$$\Omega_P^{\text{STFR}} = \mathbb{E}_P[\ell(g_{2,P}^*(Z), Y)] - \mathbb{E}_P[\ell(g_{1,P}^*(X, Z), Y)]$$

$$= \mathbb{E}_P[\ell(g_{2,P}^*(Z), Y)] - \mathbb{E}_P[\ell(f_{2,P}^*(Z), Y)]$$

$$+ \mathbb{E}_P[\ell(f_{1,P}^*(X, Z), Y)] - \mathbb{E}_P[\ell(g_{1,P}^*(X, Z), Y)] \qquad (\leq 0)$$

$$+ \mathbb{E}_P[\ell(f_{2,P}^*(Z), Y)] - \mathbb{E}_P[\ell(f_{1,P}^*(X, Z), Y)] \qquad (= 0, \text{ because } H_0 \text{ holds})$$

$$\leq \mathbb{E}_P[\ell(g_{2,P}^*(Z), Y)] - \mathbb{E}_P[\ell(f_{2,P}^*(Z), Y)]$$

$$= \Delta_{2,P}$$

Finally, see that

$$1 - \Phi\left(\tau_\alpha - \sqrt{\frac{n}{\sigma_P^2}}\Omega_P^{\text{STFR}}\right) \leq 1 - \Phi\left(\tau_\alpha - \sqrt{\frac{n}{\sigma_P^2}}\Delta_{2,P}\right)$$

and

$$\Phi\left(\tau_\alpha - \sqrt{\frac{n}{\sigma_P^2}}\Omega_P^{\text{STFR}}\right) \leq \Phi\left(\tau_\alpha - \sqrt{\frac{n}{\sigma_P^2}}(\Delta_P - \Delta_{1,P})\right)$$

Combining these observations with Theorem 3.4, we get the result. $\qquad\square$

## B.2 GCM

**Theorem A.5.** Suppose that Assumptions A.3, 3.2, and 3.3 hold. If $n$ is a function of $m$ such that $n \to \infty$ and $n = o(m^\gamma)$ as $m \to \infty$, then

$$\mathbb{E}_P[\varphi_\alpha^{\text{GCM}}(\mathcal{D}_{te}^{(n)}, \mathcal{D}_{tr}^{(m)})] = 1 - \Phi\left(\tau_{\alpha/2} - \sqrt{\frac{n}{\sigma_P^2}}\Omega_P^{\text{GCM}}\right) + \Phi\left(-\tau_{\alpha/2} - \sqrt{\frac{n}{\sigma_P^2}}\Omega_P^{\text{GCM}}\right) + o(1)$$

where $o(1)$ denotes uniform convergence over all $P \in \mathcal{P}$ as $m \to \infty$ and

$$\Omega_P^{\text{GCM}} \triangleq \mathbb{E}_P[\text{Cov}_P(X, Y \mid Z)] + \mathbb{E}_P[\delta_{1,P}(Z)\delta_{2,P}(Z)]$$

*Proof.* First, note that there must be[18] a sequence of probability measures in $\mathcal{P}$, $(P^{(m)})_{m\in\mathbb{N}}$, such that

$$\sup_{P\in\mathcal{P}}\left|\mathbb{E}_P[\varphi_\alpha^{\text{GCM}}(\mathcal{D}_{te}^{(n)}, \mathcal{D}_{tr}^{(m)})] - 1 + \Phi\left(\tau_{\alpha/2} - \sqrt{\frac{n}{\sigma_P^2}}\Omega_P^{\text{GCM}}\right) - \Phi\left(-\tau_{\alpha/2} - \sqrt{\frac{n}{\sigma_P^2}}\Omega_P^{\text{GCM}}\right)\right| \leq$$

$$\leq \left|\mathbb{E}_{P^{(m)}}[\varphi_\alpha^{\text{GCM}}(\mathcal{D}_{te}^{(n)}, \mathcal{D}_{tr}^{(m)})] - 1 + \Phi\left(\tau_{\alpha/2} - \sqrt{\frac{n}{\sigma_{P^{(m)}}^2}}\Omega_{P^{(m)}}^{\text{GCM}}\right) - \Phi\left(-\tau_{\alpha/2} - \sqrt{\frac{n}{\sigma_{P^{(m)}}^2}}\Omega_{P^{(m)}}^{\text{GCM}}\right)\right| + \frac{1}{m}$$

Then, it suffices to show that the RHS vanishes when we consider such a sequence $(P^{(m)})_{m\in\mathbb{N}}$.

Now, let us first decompose the test statistic $\Gamma^{(n,m)}$ in the following way:

$$\Gamma^{(n,m)} \triangleq \left|\frac{\sqrt{n}\bar{T}^{(n,m)}}{\hat{\sigma}^{(n,m)}}\right| =$$

$$= \left|\frac{\sqrt{n}\left(\bar{T}^{(n,m)} - \mathbb{E}_{P^{(m)}}[T_1^{(m)} \mid \mathcal{D}_{tr}^{(m)}]\right)}{\hat{\sigma}^{(n,m)}} + \frac{\sqrt{n}\mathbb{E}_{P^{(m)}}[T_1^{(m)} \mid \mathcal{D}_{tr}^{(m)}]}{\hat{\sigma}^{(n,m)}}\right|$$

$$= \left|\frac{\sqrt{n}\left(\bar{T}^{(n,m)} - \mathbb{E}_{P^{(m)}}[T_1^{(m)} \mid \mathcal{D}_{tr}^{(m)}]\right)}{\hat{\sigma}^{(n,m)}} + \frac{\sqrt{n}\mathbb{E}_{P^{(m)}}[(\hat{g}_1^{(m)}(Z_1) - g_{1,P^{(m)}}^*(Z_1))(\hat{g}_2^{(m)}(Z_1) - g_{2,P^{(m)}}^*(Z_1)) \mid \mathcal{D}_{tr}^{(m)}]}{\hat{\sigma}^{(n,m)}}\right.$$

$$+ \frac{\sqrt{n}\mathbb{E}_{P^{(m)}}[(\hat{g}_1^{(m)}(Z_1) - g_{1,P^{(m)}}^*(Z_1))\delta_{2,P^{(m)}}(Z_1) \mid \mathcal{D}_{tr}^{(m)}]}{\hat{\sigma}^{(n,m)}} + \frac{\sqrt{n}\mathbb{E}_{P^{(m)}}[\delta_{1,P^{(m)}}(Z_1)(\hat{g}_2^{(m)}(Z_1) - g_{2,P^{(m)}}^*(Z_1)) \mid \mathcal{D}_{tr}^{(m)}]}{\hat{\sigma}^{(n,m)}}$$

$$\left. + \frac{\sqrt{n}\Omega_{P^{(m)}}^{\text{GCM}}}{\hat{\sigma}^{(n,m)}}\right|$$

$$= \left|\frac{\sqrt{n}W_{1,P^{(m)}}^{(m)}}{\hat{\sigma}^{(n,m)}} + \frac{\sqrt{n}W_{2,P^{(m)}}^{(m)}}{\hat{\sigma}^{(n,m)}} + \frac{\sqrt{n}W_{3,P^{(m)}}^{(m)}}{\hat{\sigma}^{(n,m)}} + \frac{\sqrt{n}W_{4,P^{(m)}}^{(m)}}{\hat{\sigma}^{(n,m)}} + \frac{\sqrt{n}\Omega_{P^{(m)}}^{\text{GCM}}}{\hat{\sigma}^{(n,m)}}\right|$$

---

[18]Because of the definition of sup.

The terms involving one of the $\epsilon$ and $\eta$ were all zero and were omitted. Given that $n$ is a function of $m$, we omit it when writing the $W_{j,P(m)}^{(m)}$'s. Define $\sigma_{P(m)}^{(m)} \triangleq \sqrt{\mathsf{Var}_{P(m)}[T_1^{(m)} \mid \mathcal{D}_{tr}^{(m)}]}$ and see that

$$
\begin{aligned}
&\mathbb{E}_P[\varphi_\alpha^{\mathrm{GCM}}(\mathcal{D}_{te}^{(n)}, \mathcal{D}_{tr}^{(m)})] = \\
&= \mathbb{P}_P[p(\mathcal{D}_{te}^{(n)}, \mathcal{D}_{tr}^{(m)}) \le \alpha] \\
&= \mathbb{P}_P\left[1 - \Phi\left(\Gamma^{(n,m)}\right) \le \alpha/2\right] \\
&= \mathbb{P}_P\left[\Gamma^{(n,m)} \ge \tau_{\alpha/2}\right] \\
&= \mathbb{P}_P\left[\frac{\sqrt{n}W_{1,P(m)}^{(m)}}{\hat{\sigma}^{(n,m)}} + \frac{\sqrt{n}W_{2,P(m)}^{(m)}}{\hat{\sigma}^{(n,m)}} + \frac{\sqrt{n}W_{3,P(m)}^{(m)}}{\hat{\sigma}^{(n,m)}} + \frac{\sqrt{n}W_{4,P(m)}^{(m)}}{\hat{\sigma}^{(n,m)}} + \frac{\sqrt{n}\Omega_{P(m)}^{\mathrm{GCM}}}{\hat{\sigma}^{(n,m)}} \le -\tau_{\alpha/2}\right] + \\
&\quad + \mathbb{P}_P\left[\frac{\sqrt{n}W_{1,P(m)}^{(m)}}{\hat{\sigma}^{(n,m)}} + \frac{\sqrt{n}W_{2,P(m)}^{(m)}}{\hat{\sigma}^{(n,m)}} + \frac{\sqrt{n}W_{3,P(m)}^{(m)}}{\hat{\sigma}^{(n,m)}} + \frac{\sqrt{n}W_{4,P(m)}^{(m)}}{\hat{\sigma}^{(n,m)}} + \frac{\sqrt{n}\Omega_{P(m)}^{\mathrm{GCM}}}{\hat{\sigma}^{(n,m)}} \ge \tau_{\alpha/2}\right] \\
&= \mathbb{P}_P\left[\frac{\sqrt{n}W_{1,P(m)}^{(m)}}{\sigma_{P(m)}^{(m)}}\frac{\sigma_{P(m)}^{(m)}}{\sigma_{P(m)}} + \frac{\sqrt{n}W_{2,P(m)}^{(m)}}{\sigma_{P(m)}} + \frac{\sqrt{n}W_{3,P(m)}^{(m)}}{\sigma_{P(m)}} + \frac{\sqrt{n}W_{4,P(m)}^{(m)}}{\sigma_{P(m)}} + \frac{\hat{\sigma}^{(n,m)}}{\sigma_{P(m)}^{(m)}}\frac{\sigma_{P(m)}^{(m)}}{\sigma_{P(m)}}\tau_{\alpha/2} - \tau_{\alpha/2} \le -\tau_{\alpha/2} - \frac{\sqrt{n}\Omega_{P(m)}^{\mathrm{GCM}}}{\sigma_{P(m)}}\right] + \\
&\quad + \mathbb{P}_P\left[\frac{\sqrt{n}W_{1,P(m)}^{(m)}}{\sigma_{P(m)}^{(m)}}\frac{\sigma_{P(m)}^{(m)}}{\sigma_{P(m)}} + \frac{\sqrt{n}W_{2,P(m)}^{(m)}}{\sigma_{P(m)}} + \frac{\sqrt{n}W_{3,P(m)}^{(m)}}{\sigma_{P(m)}} + \frac{\sqrt{n}W_{4,P(m)}^{(m)}}{\sigma_{P(m)}} - \frac{\hat{\sigma}^{(n,m)}}{\sigma_{P(m)}^{(m)}}\frac{\sigma_{P(m)}^{(m)}}{\sigma_{P(m)}}\tau_{\alpha/2} + \tau_{\alpha/2} \ge \tau_{\alpha/2} - \frac{\sqrt{n}\Omega_{P(m)}^{\mathrm{GCM}}}{\sigma_{P(m)}}\right] \\
&= 1 - \Phi\left(\tau_{\alpha/2} - \sqrt{\frac{n}{\sigma_{P(m)}^2}}\Omega_{P(m)}^{\mathrm{GCM}}\right) + \Phi\left(-\tau_{\alpha/2} - \sqrt{\frac{n}{\sigma_{P(m)}^2}}\Omega_{P(m)}^{\mathrm{GCM}}\right) + o(1) \quad\quad\quad\quad (\mathrm{B.2})
\end{aligned}
$$

Implying that

$$
\sup_{P\in\mathcal{P}}\left|\mathbb{E}_P[\varphi_\alpha^{\mathrm{GCM}}(\mathcal{D}_{te}^{(n)}, \mathcal{D}_{tr}^{(m)})] - 1 + \Phi\left(\tau_{\alpha/2} - \sqrt{\frac{n}{\sigma_P^2}}\Omega_P^{\mathrm{GCM}}\right) - \Phi\left(-\tau_{\alpha/2} - \sqrt{\frac{n}{\sigma_P^2}}\Omega_P^{\mathrm{GCM}}\right)\right| = o(1) \text{ as } m \to \infty
$$

*Justifying step B.2.* First, from a central limit theorem for triangular arrays [6, Corollary 9.5.11], we have that

$$
\sqrt{n}\left(\frac{W_{1,P(m)}^{(m)}}{\sigma_{P(m)}^{(m)}}\right) = \sqrt{n}\left(\frac{\bar{T}^{(n,m)} - \mathbb{E}_{P(m)}[T_1^{(m)} \mid \mathcal{D}_{tr}^{(m)}]}{\sigma_{P(m)}^{(m)}}\right) = \frac{1}{\sqrt{n}}\sum_{i=1}^{n}\left(\frac{T_i^{(m)} - \mathbb{E}_{P(m)}[T_1^{(m)} \mid \mathcal{D}_{tr}^{(m)}]}{\sigma_{P(m)}^{(m)}}\right) \Rightarrow N(0,1)
$$

The conditions for the central limit theorem [6, Corollary 9.5.11] can be proven to hold like in Theorem 3.4's proof.

Second, we have that

$$
\begin{aligned}
&\frac{\sqrt{n}W_{1,P(m)}^{(m)}}{\sigma_{P(m)}^{(m)}} - \left(\frac{\sqrt{n}W_{1,P(m)}^{(m)}}{\sigma_{P(m)}^{(m)}}\frac{\sigma_{P(m)}^{(m)}}{\sigma_{P(m)}} + \frac{\sqrt{n}W_{2,P(m)}^{(m)}}{\sigma_{P(m)}} + \frac{\sqrt{n}W_{3,P(m)}^{(m)}}{\sigma_{P(m)}} + \frac{\sqrt{n}W_{4,P(m)}^{(m)}}{\sigma_{P(m)}} - \tau_{\alpha/2} + \tau_{\alpha/2}\frac{\hat{\sigma}^{(n,m)}}{\sigma_{P(m)}^{(m)}}\frac{\sigma_{P(m)}^{(m)}}{\sigma_{P(m)}}\right) = \\
&= \frac{\sqrt{n}W_{1,P(m)}^{(m)}}{\sigma_{P(m)}^{(m)}}\left(1 - \frac{\sigma_{P(m)}^{(m)}}{\sigma_{P(m)}}\right) - \frac{\sqrt{n}W_{2,P(m)}^{(m)}}{\sigma_{P(m)}} - \frac{\sqrt{n}W_{3,P(m)}^{(m)}}{\sigma_{P(m)}} - \frac{\sqrt{n}W_{4,P(m)}^{(m)}}{\sigma_{P(m)}} + \\
&\quad + \tau_{\alpha/2}\left(1 - \frac{\hat{\sigma}^{(n,m)}}{\sigma_{P(m)}^{(m)}}\right)\left(\frac{\sigma_{P(m)}^{(m)}}{\sigma_{P(m)}} - 1\right) + \tau_{\alpha/2}\left(1 - \frac{\sigma_{P(m)}^{(m)}}{\sigma_{P(m)}}\right) + \tau_{\alpha/2}\left(1 - \frac{\hat{\sigma}^{(n,m)}}{\sigma_{P(m)}^{(m)}}\right) \\
&= o_p(1) \text{ as } m \to \infty
\end{aligned}
$$

and

$$\frac{\sqrt{n}W_{1,P^{(m)}}^{(m)}}{\sigma_{P^{(m)}}^{(m)}} - \left( \frac{\sqrt{n}W_{1,P^{(m)}}^{(m)}}{\sigma_{P^{(m)}}^{(m)}} \frac{\sigma_{P^{(m)}}^{(m)}}{\sigma_{P^{(m)}}} + \frac{\sqrt{n}W_{2,P^{(m)}}^{(m)}}{\sigma_{P^{(m)}}} + \frac{\sqrt{n}W_{3,P^{(m)}}^{(m)}}{\sigma_{P^{(m)}}} + \frac{\sqrt{n}W_{4,P^{(m)}}^{(m)}}{\sigma_{P^{(m)}}} + \tau_{\alpha/2} - \tau_{\alpha/2} \frac{\hat{\sigma}^{(n,m)}}{\sigma_{P^{(m)}}^{(m)}} \frac{\sigma_{P^{(m)}}^{(m)}}{\sigma_{P^{(m)}}} \right) =$$

$$= \frac{\sqrt{n}W_{1,P^{(m)}}^{(m)}}{\sigma_{P^{(m)}}^{(m)}} \left( 1 - \frac{\sigma_{P^{(m)}}^{(m)}}{\sigma_{P^{(m)}}} \right) - \frac{\sqrt{n}W_{2,P^{(m)}}^{(m)}}{\sigma_{P^{(m)}}} - \frac{\sqrt{n}W_{3,P^{(m)}}^{(m)}}{\sigma_{P^{(m)}}} - \frac{\sqrt{n}W_{4,P^{(m)}}^{(m)}}{\sigma_{P^{(m)}}} +$$

$$+ \tau_{\alpha/2} \left( \frac{\hat{\sigma}^{(n,m)}}{\sigma_{P^{(m)}}^{(m)}} - 1 \right) \left( \frac{\sigma_{P^{(m)}}^{(m)}}{\sigma_{P^{(m)}}} - 1 \right) + \tau_{\alpha/2} \left( \frac{\sigma_{P^{(m)}}^{(m)}}{\sigma_{P^{(m)}}} - 1 \right) + \tau_{\alpha/2} \left( \frac{\hat{\sigma}^{(n,m)}}{\sigma_{P^{(m)}}^{(m)}} - 1 \right)$$

$$= o_p(1) \text{ as } m \to \infty$$

To see why the random quantities above converge to zero in probability, see that because of Assumption 3.3, Lemma[19] B.1, and continuous mapping theorem, we have that

$$\frac{\hat{\sigma}^{(n,m)}}{\sigma_{P^{(m)}}^{(m)}} - 1 = o_p(1) \text{ and } \frac{\sigma_{P^{(m)}}^{(m)}}{\sigma_{P^{(m)}}} - 1 = o_p(1) \text{ as } m \to \infty$$

Additionally, because of Assumptions A.3, 3.3, Cauchy-Schwarz inequality, and condition $n = o(m^\gamma)$, we have that

$$\left| \frac{\sqrt{n}W_{2,P^{(m)}}^{(m)}}{\sigma_{P^{(m)}}} \right| = \left| \frac{\sqrt{n}\mathbb{E}_{P^{(m)}}[(\hat{g}_1^{(m)}(Z_1) - g_{1,P^{(m)}}^*(Z_1))(\hat{g}_2^{(m)}(Z_1) - g_{2,P^{(m)}}^*(Z_1)) \mid \mathcal{D}_{tr}^{(m)}]}{\sigma_{P^{(m)}}} \right|$$

$$\leq \frac{\sqrt{n\mathbb{E}_{P^{(m)}}[(\hat{g}_1^{(m)}(Z_1) - g_{1,P^{(m)}}^*(Z_1))^2 \mid \mathcal{D}_{tr}^{(m)}]\mathbb{E}_{P^{(m)}}[(\hat{g}_2^{(m)}(Z_1) - g_{2,P^{(m)}}^*(Z_1))^2 \mid \mathcal{D}_{tr}^{(m)}]}}{\inf_{P \in \mathcal{P}} \sigma_P}$$

$$= \frac{\sqrt{o(m^\gamma)\mathcal{O}_p(m^{-2\gamma})}}{\inf_{P \in \mathcal{P}} \sigma_P}$$

$$= o_p(1)$$

$$\left| \frac{\sqrt{n}W_{3,P^{(m)}}^{(m)}}{\sigma_{P^{(m)}}} \right| = \left| \frac{\sqrt{n}\mathbb{E}_{P^{(m)}}[(\hat{g}_1^{(m)}(Z_1) - g_{1,P^{(m)}}^*(Z_1))\delta_{2,P^{(m)}}(Z_1) \mid \mathcal{D}_{tr}^{(m)}]}{\sigma_{P^{(m)}}} \right|$$

$$\leq \frac{\sqrt{n\mathbb{E}_{P^{(m)}}[(\hat{g}_1^{(m)}(Z_1) - g_{1,P^{(m)}}^*(Z_1))^2 \mid \mathcal{D}_{tr}^{(m)}]\mathbb{E}_{P^{(m)}}[(\delta_{2,P^{(m)}}(Z_1))^2]}}{\inf_{P \in \mathcal{P}} \sigma_P}$$

$$= \frac{\sqrt{o(m^\gamma)\mathcal{O}_p(m^{-\gamma})}}{\inf_{P \in \mathcal{P}} \sigma_P}$$

$$= o_p(1)$$

$$\left| \frac{\sqrt{n}W_{4,P^{(m)}}^{(m)}}{\sigma_{P^{(m)}}} \right| = \left| \frac{\sqrt{n}\mathbb{E}_{P^{(m)}}[(\hat{g}_2^{(m)}(Z_1) - g_{2,P^{(m)}}^*(Z_1))\delta_{1,P^{(m)}}(Z_1) \mid \mathcal{D}_{tr}^{(m)}]}{\sigma_{P^{(m)}}} \right|$$

$$\leq \frac{\sqrt{n\mathbb{E}_{P^{(m)}}[(\hat{g}_2^{(m)}(Z_1) - g_{2,P^{(m)}}^*(Z_1))^2 \mid \mathcal{D}_{tr}^{(m)}]\mathbb{E}_{P^{(m)}}[(\delta_{1,P^{(m)}}(Z_1))^2]}}{\inf_{P \in \mathcal{P}} \sigma_P}$$

$$= \frac{\sqrt{o(m^\gamma)\mathcal{O}_p(m^{-\gamma})}}{\inf_{P \in \mathcal{P}} \sigma_P}$$

$$= o_p(1)$$

---

[19]We can apply this STFR's lemma because it still holds when we consider GCM's test statistic.

Finally,

$$\left(\frac{\sqrt{n}W_{1,P^{(m)}}^{(m)}}{\sigma_{P^{(m)}}^{(m)}}\frac{\sigma_{P^{(m)}}^{(m)}}{\sigma_{P^{(m)}}^{(m)}}+\frac{\sqrt{n}W_{2,P^{(m)}}^{(m)}}{\sigma_{P^{(m)}}^{(m)}}+\frac{\sqrt{n}W_{3,P^{(m)}}^{(m)}}{\sigma_{P^{(m)}}^{(m)}}+\frac{\sqrt{n}W_{4,P^{(m)}}^{(m)}}{\sigma_{P^{(m)}}^{(m)}}-\tau_{\alpha/2}+\tau_{\alpha/2}\frac{\hat{\sigma}^{(n,m)}}{\sigma_{P^{(m)}}^{(m)}}\frac{\sigma_{P^{(m)}}^{(m)}}{\sigma_{P^{(m)}}^{(m)}}\right)=$$

$$=\frac{\sqrt{n}W_{1,P^{(m)}}^{(m)}}{\sigma_{P^{(m)}}^{(m)}}-\left[\frac{\sqrt{n}W_{1,P^{(m)}}^{(m)}}{\sigma_{P^{(m)}}^{(m)}}-\left(\frac{\sqrt{n}W_{1,P^{(m)}}^{(m)}}{\sigma_{P^{(m)}}^{(m)}}\frac{\sigma_{P^{(m)}}^{(m)}}{\sigma_{P^{(m)}}^{(m)}}+\frac{\sqrt{n}W_{2,P^{(m)}}^{(m)}}{\sigma_{P^{(m)}}^{(m)}}+\frac{\sqrt{n}W_{3,P^{(m)}}^{(m)}}{\sigma_{P^{(m)}}^{(m)}}+\frac{\sqrt{n}W_{4,P^{(m)}}^{(m)}}{\sigma_{P^{(m)}}^{(m)}}-\tau_{\alpha/2}+\tau_{\alpha/2}\frac{\hat{\sigma}^{(n,m)}}{\sigma_{P^{(m)}}}\right)\right]$$

$$=\frac{\sqrt{n}W_{1,P^{(m)}}^{(m)}}{\sigma_{P^{(m)}}^{(m)}}+o_p(1)\Rightarrow N(0,1)$$

and

$$\left(\frac{\sqrt{n}W_{1,P^{(m)}}^{(m)}}{\sigma_{P^{(m)}}^{(m)}}\frac{\sigma_{P^{(m)}}^{(m)}}{\sigma_{P^{(m)}}^{(m)}}+\frac{\sqrt{n}W_{2,P^{(m)}}^{(m)}}{\sigma_{P^{(m)}}^{(m)}}+\frac{\sqrt{n}W_{3,P^{(m)}}^{(m)}}{\sigma_{P^{(m)}}^{(m)}}+\frac{\sqrt{n}W_{4,P^{(m)}}^{(m)}}{\sigma_{P^{(m)}}^{(m)}}+\tau_{\alpha/2}-\tau_{\alpha/2}\frac{\hat{\sigma}^{(n,m)}}{\sigma_{P^{(m)}}^{(m)}}\frac{\sigma_{P^{(m)}}^{(m)}}{\sigma_{P^{(m)}}^{(m)}}\right)=$$

$$=\frac{\sqrt{n}W_{1,P^{(m)}}^{(m)}}{\sigma_{P^{(m)}}^{(m)}}-\left[\frac{\sqrt{n}W_{1,P^{(m)}}^{(m)}}{\sigma_{P^{(m)}}^{(m)}}-\left(\frac{\sqrt{n}W_{1,P^{(m)}}^{(m)}}{\sigma_{P^{(m)}}^{(m)}}\frac{\sigma_{P^{(m)}}^{(m)}}{\sigma_{P^{(m)}}^{(m)}}+\frac{\sqrt{n}W_{2,P^{(m)}}^{(m)}}{\sigma_{P^{(m)}}^{(m)}}+\frac{\sqrt{n}W_{3,P^{(m)}}^{(m)}}{\sigma_{P^{(m)}}^{(m)}}+\frac{\sqrt{n}W_{4,P^{(m)}}^{(m)}}{\sigma_{P^{(m)}}^{(m)}}+\tau_{\alpha/2}-\tau_{\alpha/2}\frac{\hat{\sigma}^{(n,m)}}{\sigma_{P^{(m)}}}\right)\right]$$

$$=\frac{\sqrt{n}W_{1,P^{(m)}}^{(m)}}{\sigma_{P^{(m)}}^{(m)}}+o_p(1)\Rightarrow N(0,1)$$

by Slutsky's theorem. Because $N(0,1)$ is a continuous distribution, we have uniform convergence of the distribution function [27][Chapter 8, Exercise 5] and we do not have to worry about the fact that $\tau_{\alpha/2}-\sqrt{\frac{n}{\sigma_{P^{(m)}}^2}}\Omega_{P^{(m)}}^{\text{GCM}}$ or $-\tau_{\alpha/2}-\sqrt{\frac{n}{\sigma_{P^{(m)}}^2}}\Omega_{P^{(m)}}^{\text{GCM}}$ depends on $m$.

$\square$

## B.3 RESIT

**Lemma B.2.** *Let $P_{U,V}$ and $P'_{U,V}$ be two distributions on $(\mathcal{U}\times\mathcal{V},\mathcal{B}(\mathcal{U}\times\mathcal{V})),\ \mathcal{U}\times\mathcal{V}\subseteq\mathbb{R}^{d_U\times d_V}$, with $d_U,d_V\geq1$. Assume $P_U$ and $P'_U$ are dominated by a common $\sigma$-finite measure $\mu$ and that $P_{V|U=u}=P'_{V|U=u}$ is dominated by a $\sigma$-finite measure $\nu_u$, for all $u\in\mathbb{R}^{d_U}$. Then,*

$$d_{\text{TV}}(P_{U,V},P'_{U,V})=d_{\text{TV}}(P_U,P'_U)$$

*where $d_{TV}$ denotes the total variation distance between two probability measures.*

*Proof.* Let $p_U$ and $p'_U$ denote the densities of $P_U$ and $P'_U$ w.r.t. $\mu$, and let $p_{V|U}(\cdot\mid u)$ denote the density of $P_{V|U=u}=P'_{V|U=u}$ w.r.t. $\nu_u$. From Scheffe's theorem [36][Lemma 2.1], we have that:

$$d_{\text{TV}}(P_{U,V},P'_{U,V})=\frac{1}{2}\int\int|p_U(u)p_{V|U}(v\mid u)-p'_U(u)p_{V|U}(v\mid u)|d\nu_u(v)d\mu(u)$$

$$=\frac{1}{2}\int\left(\int|p_{V|U}(v\mid u)|d\nu_u(v)\right)|p_U(u)-p'_U(u)|d\mu(u)$$

$$=\frac{1}{2}\int\left(\int p_{V|U}(v\mid u)d\nu_u(v)\right)|p_U(u)-p'_U(u)|d\mu(u)$$

$$=\frac{1}{2}\int|p_U(u)-p'_U(u)|d\mu(u)$$

$$=d_{\text{TV}}(P_U,P'_U)$$

$\square$

**Lemma B.3.** *For all $i\in[n]$, consider*

$$(\hat{\epsilon}_i,\hat{\eta}_i)\mid\mathcal{D}_{tr}^{(m)}\sim P_{\hat{\epsilon},\hat{\eta}|\mathcal{D}_{tr}^{(m)}},\ \ (\epsilon_i,\hat{\eta}_i)\mid\mathcal{D}_{tr}^{(m)}\sim P_{\epsilon,\hat{\eta}|\mathcal{D}_{tr}^{(m)}},\ \ (\hat{\epsilon}_i,\eta_i)\mid\mathcal{D}_{tr}^{(m)}\sim P_{\hat{\epsilon},\eta|\mathcal{D}_{tr}^{(m)}},$$

*where $\hat{\epsilon}_i=\epsilon_i-\delta_{1,P}(Z_i)-\left(\hat{g}_1^{(m)}(Z_i)-g_{1,P}^*(Z_i)\right)$ and $\hat{\eta}_i=\eta_i-\delta_{2,P}(Z_i)-\left(\hat{g}_2^{(m)}(Z_i)-g_{2,P}^*(Z_i)\right)$.*

*Under Assumption A.7 and $H_0 : X \perp\!\!\!\perp Y \mid Z$, we have that*

$$\mathbb{E}_P[\varphi_\alpha^{\mathrm{RESIT}}(\mathcal{D}_{te}^{(n)}, \mathcal{D}_{tr}^{(m)}) \mid \mathcal{D}_{tr}^{(m)}] \leq \alpha + \min\left\{ d_{\mathrm{TV}}\left(P_{\hat{\epsilon},\hat{\eta}|\mathcal{D}_{tr}^{(m)}}^n, P_{\epsilon,\hat{\eta}|\mathcal{D}_{tr}^{(m)}}^n\right), d_{\mathrm{TV}}\left(P_{\hat{\epsilon},\hat{\eta}|\mathcal{D}_{tr}^{(m)}}^n, P_{\hat{\epsilon},\eta|\mathcal{D}_{tr}^{(m)}}^n\right) \right\}$$

*where $d_{\mathrm{TV}}$ denotes the total variation distance between two probability measures.*

*Proof.* Let us represent the stacked residuals (in matrix form) as $(\hat{\boldsymbol{\epsilon}}, \hat{\boldsymbol{\eta}}) = \{(\hat{\epsilon}_i, \hat{\eta}_i)\}_{i=1}^n$. See that

$$\left((\hat{\boldsymbol{\epsilon}}, \hat{\boldsymbol{\eta}})^{(1)}, \cdots, (\hat{\boldsymbol{\epsilon}}, \hat{\boldsymbol{\eta}})^{(B)} \mid (\hat{\boldsymbol{\epsilon}}, \hat{\boldsymbol{\eta}}) = (\bar{\boldsymbol{\epsilon}}, \bar{\boldsymbol{\eta}}), \mathcal{D}_{tr}^{(m)}\right) \stackrel{d}{=} \left((\boldsymbol{\epsilon}, \hat{\boldsymbol{\eta}})^{(1)}, \cdots, (\boldsymbol{\epsilon}, \hat{\boldsymbol{\eta}})^{(B)} \mid (\boldsymbol{\epsilon}, \hat{\boldsymbol{\eta}}) = (\bar{\boldsymbol{\epsilon}}, \bar{\boldsymbol{\eta}}), \mathcal{D}_{tr}^{(m)}\right)$$

Then, because the random quantities above are conditionally discrete, their distribution is dominated by a counting measure depending on $(\bar{\boldsymbol{\epsilon}}, \bar{\boldsymbol{\eta}})$. Because the distribution of $(\epsilon, \eta)$ is absolutely continuous with respect to the Lebesgue measure, $(\hat{\epsilon}, \hat{\eta}) \mid \mathcal{D}_{tr}^{(m)}$ and $(\epsilon, \hat{\eta}) \mid \mathcal{D}_{tr}^{(m)}$ are also absolutely continuous[20] for every training set configuration, and then we can apply Lemma B.2 to get that

$$d_{\mathrm{TV}}\left(((\hat{\boldsymbol{\epsilon}}, \hat{\boldsymbol{\eta}}), (\hat{\boldsymbol{\epsilon}}, \hat{\boldsymbol{\eta}})^{(1)}, \cdots, (\hat{\boldsymbol{\epsilon}}, \hat{\boldsymbol{\eta}})^{(B)}) \mid \mathcal{D}_{tr}^{(m)}, ((\boldsymbol{\epsilon}, \hat{\boldsymbol{\eta}}), (\boldsymbol{\epsilon}, \hat{\boldsymbol{\eta}})^{(1)}, \cdots, (\boldsymbol{\epsilon}, \hat{\boldsymbol{\eta}})^{(B)}) \mid \mathcal{D}_{tr}^{(m)}\right) = d_{\mathrm{TV}}\left((\hat{\boldsymbol{\epsilon}}, \hat{\boldsymbol{\eta}}) \mid \mathcal{D}_{tr}^{(m)}, (\boldsymbol{\epsilon}, \hat{\boldsymbol{\eta}}) \mid \mathcal{D}_{tr}^{(m)}\right).$$

In the last step, we abuse TV distance notation: by the TV distance of two random variables, we mean the TV distance of their distributions.

Now, define the event

$$A_\alpha \triangleq \left\{((\boldsymbol{e}, \boldsymbol{h}), (\boldsymbol{e}, \boldsymbol{h})^{(1)}, \cdots, (\boldsymbol{e}, \boldsymbol{h})^{(B)}) : \frac{1 + \sum_{b=1}^B \mathbb{1}[\Psi((\boldsymbol{e}, \boldsymbol{h})^{(b)}) \geq \Psi((\boldsymbol{e}, \boldsymbol{h}))]}{1 + B} \leq \alpha\right\}$$

By the definition of the TV distance, we have that (under $H_0$):

$$\mathbb{E}_P[\varphi_\alpha^{\mathrm{RESIT}}(\mathcal{D}_{te}^{(n)}, \mathcal{D}_{tr}^{(m)}) \mid \mathcal{D}_{tr}^{(m)}] = \mathbb{P}_P\left(((\hat{\boldsymbol{\epsilon}}, \hat{\boldsymbol{\eta}}), (\hat{\boldsymbol{\epsilon}}, \hat{\boldsymbol{\eta}})^{(1)}, \cdots, (\hat{\boldsymbol{\epsilon}}, \hat{\boldsymbol{\eta}})^{(B)}) \in A_\alpha \mid \mathcal{D}_{tr}^{(m)}\right)$$

$$\leq \mathbb{P}_P\left(((\boldsymbol{\epsilon}, \hat{\boldsymbol{\eta}}), (\boldsymbol{\epsilon}, \hat{\boldsymbol{\eta}})^{(1)}, \cdots, (\boldsymbol{\epsilon}, \hat{\boldsymbol{\eta}})^{(B)}) \in A_\alpha \mid \mathcal{D}_{tr}^{(m)}\right) + d_{\mathrm{TV}}\left((\hat{\boldsymbol{\epsilon}}, \hat{\boldsymbol{\eta}}) \mid \mathcal{D}_{tr}^{(m)}, (\boldsymbol{\epsilon}, \hat{\boldsymbol{\eta}}) \mid \mathcal{D}_{tr}^{(m)}\right)$$

$$\leq \alpha + d_{\mathrm{TV}}\left((\hat{\boldsymbol{\epsilon}}, \hat{\boldsymbol{\eta}}) \mid \mathcal{D}_{tr}^{(m)}, (\boldsymbol{\epsilon}, \hat{\boldsymbol{\eta}}) \mid \mathcal{D}_{tr}^{(m)}\right)$$

$$= \alpha + d_{\mathrm{TV}}\left(P_{\hat{\epsilon},\hat{\eta}|\mathcal{D}_{tr}^{(m)}}^n, P_{\epsilon,\hat{\eta}|\mathcal{D}_{tr}^{(m)}}^n\right)$$

where the last equality holds from the fact that, given the training set, rows of $(\hat{\boldsymbol{\epsilon}}, \hat{\boldsymbol{\eta}})$ and $(\boldsymbol{\epsilon}, \hat{\boldsymbol{\eta}})$ are i.i.d.. See that $\mathbb{P}_P\left(((\boldsymbol{\epsilon}, \hat{\boldsymbol{\eta}}), (\boldsymbol{\epsilon}, \hat{\boldsymbol{\eta}})^{(1)}, \cdots, (\boldsymbol{\epsilon}, \hat{\boldsymbol{\eta}})^{(B)}) \in A_\alpha \mid \mathcal{D}_{tr}^{(m)}\right) \leq \alpha$ because $H_0$ holds and therefore $\epsilon_i \perp\!\!\!\perp \hat{\eta}_i \mid \mathcal{D}_{tr}^{(m)}$ (making the permuted samples exchangeable).

Using symmetry, we have that

$$\mathbb{E}_P[\varphi_\alpha^{\mathrm{RESIT}}(\mathcal{D}_{te}^{(n)}, \mathcal{D}_{tr}^{(m)}) \mid \mathcal{D}_{tr}^{(m)}] \leq \alpha + \min\left\{ d_{\mathrm{TV}}\left(P_{\hat{\epsilon},\hat{\eta}|\mathcal{D}_{tr}^{(m)}}^n, P_{\epsilon,\hat{\eta}|\mathcal{D}_{tr}^{(m)}}^n\right), d_{\mathrm{TV}}\left(P_{\hat{\epsilon},\hat{\eta}|\mathcal{D}_{tr}^{(m)}}^n, P_{\hat{\epsilon},\eta|\mathcal{D}_{tr}^{(m)}}^n\right) \right\}$$

$\square$

**Lemma B.4.** *For any $i \in [n]$, consider that*

$$(\hat{\epsilon}_i, \hat{\eta}_i) \mid \mathcal{D}_{tr}^{(m)} \sim P_{\hat{\epsilon},\hat{\eta}|\mathcal{D}_{tr}^{(m)}}, \quad (\epsilon_i, \hat{\eta}_i) \mid \mathcal{D}_{tr}^{(m)} \sim P_{\epsilon,\hat{\eta}|\mathcal{D}_{tr}^{(m)}}, \quad (\hat{\epsilon}_i, \eta_i) \mid \mathcal{D}_{tr}^{(m)} \sim P_{\hat{\epsilon},\eta|\mathcal{D}_{tr}^{(m)}}$$

$$(\epsilon_i^*, \eta_i^*) \sim P_{\epsilon^*,\eta^*}, \quad (\epsilon_i, \eta_i^*) \sim P_{\epsilon,\eta^*}, \quad (\epsilon_i^*, \eta_i) \sim P_{\epsilon^*,\eta}$$

*where $\hat{\epsilon}_i = \epsilon_i - \delta_{1,P}(Z_i) - \left(\hat{g}_1^{(m)}(Z_i) - g_{1,P}^*(Z_i)\right)$, $\epsilon_i^* = \epsilon_i - \delta_{1,P}(Z_i)$, $\hat{\eta}_i = \eta_i - \delta_{2,P}(Z_i) - \left(\hat{g}_2^{(m)}(Z_i) - g_{2,P}^*(Z_i)\right)$, and $\eta_i^* = \eta_i - \delta_{2,P}(Z_i)$. Then, under $H_0 : X \perp\!\!\!\perp Y \mid Z$ and Assumptions*

---

[20]Given any training set configuration, the vectors $(\hat{\epsilon}_i, \hat{\eta}_i)$, $(\epsilon_i, \hat{\eta}_i)$, $(\hat{\epsilon}_i, \eta_i)$ are given by the sum of two independent random vectors where at least one of them is continuous because of Assumption A.7 and therefore the sum must be continuous, *e.g.*, $(\hat{\epsilon}_i, \hat{\eta}_i) = (\epsilon_i, \eta_i) + (g_{1,P}^*(Z_i) - \hat{g}_1^{(m)}(Z_i) - \delta_{1,P}(Z_i), g_{2,P}^*(Z_i) - \hat{g}_2^{(m)}(Z_i) - \delta_{2,P}(Z_i))$. See Lemma B.4 for a proof.

 all six distributions are absolutely continuous with respect to the Lebesgue measure and their densities are given by

$$p_{\hat{\epsilon},\hat{\eta}|\mathcal{D}_{tr}^{(m)}}(e, h \mid \mathcal{D}_{tr}^{(m)}) = \mathbb{E}_P \left[ p_\epsilon \left( e + \delta_{1,P}(Z) + \hat{g}_1^{(m)}(Z) - g_{1,P}^*(Z) \right) p_\eta \left( h + \delta_{2,P}(Z) + \hat{g}_2^{(m)}(Z) - g_{2,P}^*(Z) \right) \mid \mathcal{D}_{tr}^{(m)} \right]$$

$$p_{\epsilon,\hat{\eta}|\mathcal{D}_{tr}^{(m)}}(e, h \mid \mathcal{D}_{tr}^{(m)}) = p_\epsilon(e) \cdot \mathbb{E}_P \left[ p_\eta \left( h + \delta_{2,P}(Z) + \hat{g}_2^{(m)}(Z) - g_{2,P}^*(Z) \right) \mid \mathcal{D}_{tr}^{(m)} \right]$$

$$p_{\hat{\epsilon},\eta|\mathcal{D}_{tr}^{(m)}}(e, h \mid \mathcal{D}_{tr}^{(m)}) = \mathbb{E}_P \left[ p_\epsilon \left( e + \delta_{1,P}(Z) + \hat{g}_1^{(m)}(Z) - g_{1,P}^*(Z) \right) \mid \mathcal{D}_{tr}^{(m)} \right] \cdot p_\eta(h)$$

$$p_{\epsilon^*,\eta^*}(e, h) = \mathbb{E}_P \left[ p_\epsilon \left( e + \delta_{1,P}(Z) \right) p_\eta \left( h + \delta_{2,P}(Z) \right) \right]$$

$$p_{\epsilon,\eta^*}(e, h) = p_\epsilon(e) \cdot \mathbb{E}_P \left[ p_\eta \left( h + \delta_{2,P}(Z) \right) \right]$$

$$p_{\epsilon^*,\eta}(e, h) = \mathbb{E}_P \left[ p_\epsilon \left( e + \delta_{1,P}(Z) \right) \right] \cdot p_\eta(h)$$

*Additionally, we have that*

$$\sup_{(e,h)\in\mathbb{R}^{d_X \times d_Y}} \left( p_{\hat{\epsilon},\hat{\eta}|\mathcal{D}_{tr}^{(m)}}(e, h \mid \mathcal{D}_{tr}^{(m)}) - p_{\epsilon^*,\eta^*}(e, h) \right)^2 = \mathcal{O}_{\mathcal{P}_0}(m^{-\gamma})$$

$$\sup_{(e,h)\in\mathbb{R}^{d_X \times d_Y}} \left( p_{\epsilon,\hat{\eta}|\mathcal{D}_{tr}^{(m)}}(e, h \mid \mathcal{D}_{tr}^{(m)}) - p_{\epsilon,\eta^*}(e, h) \right)^2 = \mathcal{O}_{\mathcal{P}_0}(m^{-\gamma})$$

$$\sup_{(e,h)\in\mathbb{R}^{d_X \times d_Y}} \left( p_{\hat{\epsilon},\eta|\mathcal{D}_{tr}^{(m)}}(e, h \mid \mathcal{D}_{tr}^{(m)}) - p_{\epsilon^*,\eta}(e, h) \right)^2 = \mathcal{O}_{\mathcal{P}_0}(m^{-\gamma})$$

*Proof.* Assume we are under $H_0$. In order to show that $P_{\hat{\epsilon},\hat{\eta}|\mathcal{D}_{tr}^{(m)}}$ is absolutely continuous w.r.t. Lebesgue measure (for each training set configuration) and that its density is given by

$$p_{\hat{\epsilon},\hat{\eta}|\mathcal{D}_{tr}^{(m)}}(e, h \mid \mathcal{D}_{tr}^{(m)}) = \mathbb{E}_P \left[ p_\epsilon \left( e + \delta_{1,P}(Z) + \hat{g}_1^{(m)}(Z) - g_{1,P}^*(Z) \right) p_\eta \left( h + \delta_{2,P}(Z) + \hat{g}_2^{(m)}(Z) - g_{2,P}^*(Z) \right) \mid \mathcal{D}_{tr}^{(m)} \right],$$

it suffices to show that

$$\mathbb{P}_P((\hat{\epsilon}, \hat{\eta}) \in A \mid \mathcal{D}_{tr}^{(m)}) =$$
$$= \int \mathbb{1}_A(u, v) \mathbb{E}_P \left[ p_\epsilon \left( u + \delta_{1,P}(Z) + \hat{g}_1^{(m)}(Z) - g_{1,P}^*(Z) \right) p_\eta \left( v + \delta_{2,P}(Z) + \hat{g}_2^{(m)}(Z) - g_{2,P}^*(Z) \right) \mid \mathcal{D}_{tr}^{(m)} \right] d(u, v)$$

for any measurable set $A$.

Using Fubini's theorem, we get

$$\mathbb{P}_P((\hat{\epsilon}, \hat{\eta}) \in A \mid \mathcal{D}_{tr}^{(m)}) =$$
$$= \mathbb{E}_P \left[ \mathbb{1}_A(\epsilon - \delta_{1,P}(Z) - \hat{g}_1^{(m)}(Z) + g_{1,P}^*(Z), \eta - \delta_{2,P}(Z) - \hat{g}_2^{(m)}(Z) + g_{2,P}^*(Z)) \mid \mathcal{D}_{tr}^{(m)} \right]$$
$$= \mathbb{E}_P \left[ \int \mathbb{1}_A(e - \delta_{1,P}(Z) - \hat{g}_1^{(m)}(Z) + g_{1,P}^*(Z), h - \delta_{2,P}(Z) - \hat{g}_2^{(m)}(Z) + g_{2,P}^*(Z)) p_{\epsilon,\eta}(e, h) d(e, h) \mid \mathcal{D}_{tr}^{(m)} \right]$$
$$= \mathbb{E}_P \left[ \int \mathbb{1}_A(e - \delta_{1,P}(Z) - \hat{g}_1^{(m)}(Z) + g_{1,P}^*(Z), h - \delta_{2,P}(Z) - \hat{g}_2^{(m)}(Z) + g_{2,P}^*(Z)) p_\epsilon(e) p_\eta(h) d(e, h) \mid \mathcal{D}_{tr}^{(m)} \right]$$
$$= \mathbb{E}_P \left[ \int \mathbb{1}_A(u, v) p_\epsilon \left( u + \delta_{1,P}(Z) + \hat{g}_1^{(m)}(Z) - g_{1,P}^*(Z) \right) p_\eta \left( v + \delta_{2,P}(Z) + \hat{g}_2^{(m)}(Z) - g_{2,P}^*(Z) \right) d(u, v) \mid \mathcal{D}_{tr}^{(m)} \right]$$
$$= \int \mathbb{1}_A(u, v) \mathbb{E}_P \left[ p_\epsilon \left( u + \delta_{1,P}(Z) + \hat{g}_1^{(m)}(Z) - g_{1,P}^*(Z) \right) p_\eta \left( v + \delta_{2,P}(Z) + \hat{g}_2^{(m)}(Z) - g_{2,P}^*(Z) \right) \mid \mathcal{D}_{tr}^{(m)} \right] d(u, v)$$

The proof is analogous to the other distributions.

Now, we proceed with the second part of the lemma. Using Assumptions A.6 and A.7, we get

$$\left(p_{\hat{\epsilon},\hat{\eta}|\mathcal{D}_{tr}^{(m)}}(e, h \mid \mathcal{D}_{tr}^{(m)}) - p_{\epsilon^*,\eta^*}(e, h)\right)^2 =$$

$$= \left(\mathbb{E}_P\left[p_\epsilon\left(e + \delta_{1,P}(Z) + \hat{g}_1^{(m)}(Z) - g_{1,P}^*(Z)\right)p_\eta\left(h + \delta_{2,P}(Z) + \hat{g}_2^{(m)}(Z) - g_{2,P}^*(Z)\right)\right.$$

$$\left. - p_\epsilon\left(e + \delta_{1,P}(Z)\right)p_\eta\left(h + \delta_{2,P}(Z)\right) \mid \mathcal{D}_{tr}^{(m)}\right]\right)^2$$

$$\leq L^2\left(\mathbb{E}_P\left[\left\|\left(\hat{g}_1^{(m)}(Z) - g_1^*(Z), \hat{g}_2^{(m)}(Z) - g_2^*(Z)\right)\right\|_2 \mid \mathcal{D}_{tr}^{(m)}\right]\right)^2$$

$$\leq L^2\left(\mathbb{E}_P\left[\left\|\left(\hat{g}_1^{(m)}(Z) - g_1^*(Z), \hat{g}_2^{(m)}(Z) - g_2^*(Z)\right)\right\|_2^2 \mid \mathcal{D}_{tr}^{(m)}\right]\right)$$

$$= L^2\mathbb{E}_P\left[\left\|\hat{g}_1^{(m)}(Z) - g_{1,P}^*(Z)\right\|_2^2 \mid \mathcal{D}_{tr}^{(m)}\right] + L^2\mathbb{E}_P\left[\left\|\hat{g}_2^{(m)}(Z) - g_{2,P}^*(Z)\right\|_2^2 \mid \mathcal{D}_{tr}^{(m)}\right]$$

$$= \mathcal{O}_{\mathcal{P}_0}(m^{-\gamma})$$

where the last inequality is obtained via Jensen's inequality. Because the upper bound obtained through Lipschitzness does not depend on $(e, h)$, we get

$$\sup_{(e,h)\in\mathbb{R}^{d_X\times d_Y}}\left(p_{\hat{\epsilon},\hat{\eta}|\mathcal{D}_{tr}^{(m)}}(e, h \mid \mathcal{D}_{tr}^{(m)}) - p_{\epsilon^*,\eta^*}(e, h)\right)^2 = \mathcal{O}_{\mathcal{P}_0}(m^{-\gamma})$$

The results for the other converging quantities are obtained in the same manner. □

**Theorem A.9.** Under Assumptions A.6, A.7, and A.8, if $H_0 : X \perp\!\!\!\perp Y \mid Z$ holds and $n$ is a function of $m$ such that $n \to \infty$ and $n = o(m^{\frac{\gamma}{2}})$ as $m \to \infty$, then

$$\mathbb{E}_P[\varphi_\alpha^{\text{RESIT}}(\mathcal{D}_{te}^{(n)}, \mathcal{D}_{tr}^{(m)})] \leq \alpha + \min\{d_{\text{TV}}(P_{\epsilon^*,\eta^*}^n, P_{\epsilon,\eta^*}^n), d_{\text{TV}}(P_{\epsilon^*,\eta^*}^n, P_{\epsilon^*,\eta}^n)\} + o(1)$$

where $o(1)$ denotes uniform convergence over all $P \in \mathcal{P}_0$ as $m \to \infty$.

*Proof.* We are trying to prove that

$$\sup_{P\in\mathcal{P}_0}\left[\mathbb{E}_P[\varphi_\alpha^{\text{RESIT}}(\mathcal{D}_{te}^{(n)}, \mathcal{D}_{tr}^{(m)})] - \alpha - \min\{d_{\text{TV}}(P_{\epsilon^*,\eta^*}^n, P_{\epsilon,\eta^*}^n), d_{\text{TV}}(P_{\epsilon^*,\eta^*}^n, P_{\epsilon^*,\eta}^n)\}\right] \leq o(1) \text{ as } m \to \infty$$

First, see that using Lemma B.3 we get

$$\mathbb{E}_P[\varphi_\alpha^{\text{RESIT}}(\mathcal{D}_{te}^{(n)}, \mathcal{D}_{tr}^{(m)})] =$$

$$= \mathbb{E}_P[\mathbb{E}_P[\varphi_\alpha^{\text{RESIT}}(\mathcal{D}_{te}^{(n)}, \mathcal{D}_{tr}^{(m)}) \mid \mathcal{D}_{tr}^{(m)}]]$$

$$\leq \alpha + \mathbb{E}_P\left[\min\left\{d_{\text{TV}}\left(P_{\hat{\epsilon},\hat{\eta}|\mathcal{D}_{tr}^{(m)}}^n, P_{\epsilon,\hat{\eta}|\mathcal{D}_{tr}^{(m)}}^n\right), d_{\text{TV}}\left(P_{\hat{\epsilon},\hat{\eta}|\mathcal{D}_{tr}^{(m)}}^n, P_{\hat{\epsilon},\eta|\mathcal{D}_{tr}^{(m)}}^n\right)\right\}\right]$$

$$\leq \alpha + \min\left\{\mathbb{E}_P\left[d_{\text{TV}}\left(P_{\hat{\epsilon},\hat{\eta}|\mathcal{D}_{tr}^{(m)}}^n, P_{\epsilon,\hat{\eta}|\mathcal{D}_{tr}^{(m)}}^n\right)\right], \mathbb{E}_P\left[d_{\text{TV}}\left(P_{\hat{\epsilon},\hat{\eta}|\mathcal{D}_{tr}^{(m)}}^n, P_{\hat{\epsilon},\eta|\mathcal{D}_{tr}^{(m)}}^n\right)\right]\right\}$$

$$\leq \alpha + \min\left\{\mathbb{E}_P\left[d_{\text{TV}}\left(P_{\hat{\epsilon},\hat{\eta}|\mathcal{D}_{tr}^{(m)}}^n, P_{\epsilon,\hat{\eta}|\mathcal{D}_{tr}^{(m)}}^n\right)\right] - d_{\text{TV}}(P_{\epsilon^*,\eta^*}^n, P_{\epsilon,\eta^*}^n) + d_{\text{TV}}(P_{\epsilon^*,\eta^*}^n, P_{\epsilon,\eta^*}^n),\right.$$

$$\left.\mathbb{E}_P\left[d_{\text{TV}}\left(P_{\hat{\epsilon},\hat{\eta}|\mathcal{D}_{tr}^{(m)}}^n, P_{\hat{\epsilon},\eta|\mathcal{D}_{tr}^{(m)}}^n\right)\right] - d_{\text{TV}}(P_{\epsilon^*,\eta^*}^n, P_{\epsilon^*,\eta}^n) + d_{\text{TV}}(P_{\epsilon^*,\eta^*}^n, P_{\epsilon^*,\eta}^n)\right\}$$

$$\leq \alpha + \min\left\{\left|\mathbb{E}_P\left[d_{\text{TV}}\left(P_{\hat{\epsilon},\hat{\eta}|\mathcal{D}_{tr}^{(m)}}^n, P_{\epsilon,\hat{\eta}|\mathcal{D}_{tr}^{(m)}}^n\right)\right] - d_{\text{TV}}(P_{\epsilon^*,\eta^*}^n, P_{\epsilon,\eta^*}^n)\right| + d_{\text{TV}}(P_{\epsilon^*,\eta^*}^n, P_{\epsilon,\eta^*}^n),\right.$$

$$\left.\left|\mathbb{E}_P\left[d_{\text{TV}}\left(P_{\hat{\epsilon},\hat{\eta}|\mathcal{D}_{tr}^{(m)}}^n, P_{\hat{\epsilon},\eta|\mathcal{D}_{tr}^{(m)}}^n\right)\right] - d_{\text{TV}}(P_{\epsilon^*,\eta^*}^n, P_{\epsilon^*,\eta}^n)\right| + d_{\text{TV}}(P_{\epsilon^*,\eta^*}^n, P_{\epsilon^*,\eta}^n)\right\}$$

and

$$\sup_{P\in\mathcal{P}_0}\left[\mathbb{E}_P[\varphi_\alpha^{\text{RESIT}}(\mathcal{D}_{te}^{(n)}, \mathcal{D}_{tr}^{(m)})] - \alpha - \min\{d_{\text{TV}}(P_{\epsilon^*,\eta^*}^n, P_{\epsilon,\eta^*}^n), d_{\text{TV}}(P_{\epsilon^*,\eta^*}^n, P_{\epsilon^*,\eta}^n)\}\right] \leq \sup_{P\in\mathcal{P}_0}\Delta_P^{(m)}$$

(B.3)

where

$$\Delta_P^{(m)} \triangleq \min \left\{ \left| \mathbb{E}_P \left[ d_{\mathrm{TV}} \left( P^n_{\hat{\epsilon},\hat{\eta}|\mathcal{D}_{tr}^{(m)}}, P^n_{\epsilon,\hat{\eta}|\mathcal{D}_{tr}^{(m)}} \right) \right] - d_{\mathrm{TV}}(P^n_{\epsilon^*,\eta^*}, P^n_{\epsilon,\eta^*}) \right| + d_{\mathrm{TV}}(P^n_{\epsilon^*,\eta^*}, P^n_{\epsilon,\eta^*}), \right.$$

$$\left. \left| \mathbb{E}_P \left[ d_{\mathrm{TV}} \left( P^n_{\hat{\epsilon},\hat{\eta}|\mathcal{D}_{tr}^{(m)}}, P^n_{\epsilon,\eta|\mathcal{D}_{tr}^{(m)}} \right) \right] - d_{\mathrm{TV}}(P^n_{\epsilon^*,\eta^*}, P^n_{\epsilon^*,\eta}) \right| + d_{\mathrm{TV}}(P^n_{\epsilon^*,\eta^*}, P^n_{\epsilon^*,\eta}) \right\}$$

$$- \min\{d_{\mathrm{TV}}(P^n_{\epsilon^*,\eta^*}, P^n_{\epsilon,\eta^*}), d_{\mathrm{TV}}(P^n_{\epsilon^*,\eta^*}, P^n_{\epsilon^*,\eta})\}$$

It suffices to show that $\sup_{P \in \mathcal{P}_0} \Delta_P^{(m)} = o(1)$ as $m \to \infty$. Next step is to show that

$$\sup_{P \in \mathcal{P}_0} \left| \mathbb{E}_P \left[ d_{\mathrm{TV}} \left( P^n_{\hat{\epsilon},\hat{\eta}|\mathcal{D}_{tr}^{(m)}}, P^n_{\epsilon,\hat{\eta}|\mathcal{D}_{tr}^{(m)}} \right) \right] - d_{\mathrm{TV}}(P^n_{\epsilon^*,\eta^*}, P^n_{\epsilon,\eta^*}) \right| = o(1)$$

and

$$\sup_{P \in \mathcal{P}_0} \left| \mathbb{E}_P \left[ d_{\mathrm{TV}} \left( P^n_{\hat{\epsilon},\hat{\eta}|\mathcal{D}_{tr}^{(m)}}, P^n_{\epsilon,\eta|\mathcal{D}_{tr}^{(m)}} \right) \right] - d_{\mathrm{TV}}(P^n_{\epsilon^*,\eta^*}, P^n_{\epsilon^*,\eta}) \right| = o(1)$$

as $m \to \infty$. Given the symmetry, we focus on the first problem.

By the triangle inequality, we obtain

$$\mathbb{E}_P \left[ d_{\mathrm{TV}} \left( P^n_{\hat{\epsilon},\hat{\eta}|\mathcal{D}_{tr}^{(m)}}, P^n_{\epsilon,\hat{\eta}|\mathcal{D}_{tr}^{(m)}} \right) \right] \leq \mathbb{E}_P \left[ d_{\mathrm{TV}} \left( P^n_{\hat{\epsilon},\hat{\eta}|\mathcal{D}_{tr}^{(m)}}, P^n_{\epsilon^*,\eta^*} \right) \right] + d_{\mathrm{TV}} \left( P^n_{\epsilon^*,\eta^*}, P^n_{\epsilon,\eta^*} \right) + \mathbb{E}_P \left[ d_{\mathrm{TV}} \left( P^n_{\epsilon,\eta^*}, P^n_{\epsilon,\hat{\eta}|\mathcal{D}_{tr}^{(m)}} \right) \right]$$

and consequently

$$\sup_{P \in \mathcal{P}_0} \left| \mathbb{E}_P \left[ d_{\mathrm{TV}} \left( P^n_{\hat{\epsilon},\hat{\eta}|\mathcal{D}_{tr}^{(m)}}, P^n_{\epsilon,\hat{\eta}|\mathcal{D}_{tr}^{(m)}} \right) \right] - d_{\mathrm{TV}} \left( P^n_{\epsilon^*,\eta^*}, P^n_{\epsilon,\eta^*} \right) \right|$$

$$\leq \underbrace{\sup_{P \in \mathcal{P}_0} \mathbb{E}_P \left[ d_{\mathrm{TV}} \left( P^n_{\hat{\epsilon},\hat{\eta}|\mathcal{D}_{tr}^{(m)}}, P^n_{\epsilon^*,\eta^*} \right) \right]}_{\mathrm{I}} + \underbrace{\sup_{P \in \mathcal{P}_0} \mathbb{E}_P \left[ d_{\mathrm{TV}} \left( P^n_{\epsilon,\hat{\eta}|\mathcal{D}_{tr}^{(m)}}, P^n_{\epsilon,\eta^*} \right) \right]}_{\mathrm{II}}$$

We treat these terms separately.

(I) Consider a sequence of probability distributions $(P^{(m)})_{m \in \mathbb{N}}$ in $\mathcal{P}_0$ such that

$$\sup_{P \in \mathcal{P}_0} \mathbb{E}_P \left[ d_{\mathrm{TV}} \left( P^n_{\hat{\epsilon},\hat{\eta}|\mathcal{D}_{tr}^{(m)}}, P^n_{\epsilon^*,\eta^*} \right) \right] \leq \mathbb{E}_{P^{(m)}} \left[ d_{\mathrm{TV}} \left( P^{(m)}{}^n_{\hat{\epsilon},\hat{\eta}|\mathcal{D}_{tr}^{(m)}}, P^{(m)}{}^n_{\epsilon^*,\eta^*} \right) \right] + \frac{1}{m}$$

Here, the distributions $P$ and $P^{(m)}$ determine not only the distribution associated with $\mathcal{D}_{tr}^{(m)}$ but also the distribution of $(\hat{\epsilon}, \hat{\eta}, \epsilon^*, \eta^*)$. Because we have that

$$d_{\mathrm{TV}} \left( P^n_{\hat{\epsilon},\hat{\eta}|\mathcal{D}_{tr}^{(m)}}, P^n_{\epsilon^*,\eta^*} \right) \leq$$

$$\leq n \cdot d_{\mathrm{TV}} \left( P_{\hat{\epsilon},\hat{\eta}|\mathcal{D}_{tr}^{(m)}}, P_{\epsilon^*,\eta^*} \right) \qquad \text{(Subadditivity of TV distance)}$$

$$= \frac{n}{2} \int \left| p_{\hat{\epsilon},\hat{\eta}|\mathcal{D}_{tr}^{(m)}}(e, h \mid \mathcal{D}_{tr}^{(m)}) - p_{\epsilon^*,\eta^*}(e, h) \right| d(u, v) \qquad \text{(Scheffe's theorem [36][Lemma 2.1])}$$

$$\leq \frac{nV}{2} \sup_{(u,v) \in \mathbb{R}^{d_X \times d_Y}} \left| p_{\hat{\epsilon},\hat{\eta}|\mathcal{D}_{tr}^{(m)}}(e, h \mid \mathcal{D}_{tr}^{(m)}) - p_{\epsilon^*,\eta^*}(e, h) \right| \qquad \text{(Assumption A.8)}$$

$$= \left( \frac{n^2 V^2}{4} \sup_{(u,v) \in \mathbb{R}^{d_X \times d_Y}} \left| p_{\hat{\epsilon},\hat{\eta}|\mathcal{D}_{tr}^{(m)}}(e, h \mid \mathcal{D}_{tr}^{(m)}) - p_{\epsilon^*,\eta^*}(e, h) \right|^2 \right)^{1/2}$$

$$= \left( o(m^\gamma) \mathcal{O}_{\mathcal{P}_0}(m^{-\gamma}) \right)^{1/2} \qquad \text{(Lemma B.4)}$$

$$= o_{\mathcal{P}_0}(1)$$

where $V$ is the volume of a ball containing the support of the RVs (existence of that ball is due to Assumption A.8), we also have that $d_{\mathrm{TV}} \left( P^{(m)}{}^n_{\hat{\epsilon},\hat{\eta}|\mathcal{D}_{tr}^{(m)}}, P^{(m)}{}^n_{\epsilon^*,\eta^*} \right) = o_p(1)$, when $\mathcal{D}_{tr}^{(m)}$ samples come from the sequence $(P^{(m)})_{m \in \mathbb{N}}$. By the Dominated Convergence Theorem (DCT) [27, Corollary 6.3.2], we have

$$\sup_{P \in \mathcal{P}_0} \mathbb{E}_P \left[ d_{\mathrm{TV}} \left( P^n_{\hat{\epsilon},\hat{\eta}|\mathcal{D}_{tr}^{(m)}}, P^n_{\epsilon^*,\eta^*} \right) \right] \leq \mathbb{E}_{P^{(m)}} \left[ d_{\mathrm{TV}} \left( P^{(m)}{}^n_{\hat{\epsilon},\hat{\eta}|\mathcal{D}_{tr}^{(m)}}, P^{(m)}{}^n_{\epsilon^*,\eta^*} \right) \right] + \frac{1}{m} = o(1)$$

To see why we can use the DCT here, realize that when samples come from $P^{(m)}$, $W^{(m)} = d_{\text{TV}}\left(P^{(m)}{}^n_{\hat{\epsilon},\hat{\eta}|\mathcal{D}_{tr}^{(m)}}, P^{(m)}{}^n_{\epsilon^*,\eta^*}\right)$ can be seen as a measurable function going from an original probability space to some other space. Different distributions $\{P^{(m)}\}$ are due to different random variables while the original probability measure is fixed. Because of that,

$$\mathbb{E}_{P^{(m)}}\left[d_{\text{TV}}\left(P^{(m)}{}^n_{\hat{\epsilon},\hat{\eta}|\mathcal{D}_{tr}^{(m)}}, P^{(m)}{}^n_{\epsilon^*,\eta^*}\right)\right] = \mathbb{E}\left[W^{(m)}\right]$$

for the bounded random variable $W^{(m)}$, where the last expectation is taken in the original probability space. We apply the DCT in $\mathbb{E}\left[W^{(m)}\right]$.

(II) Following the same steps as in part (I), we obtain

$$\sup_{P \in \mathcal{P}_0} \mathbb{E}_P\left[d_{\text{TV}}\left(P^n_{\epsilon,\hat{\eta}|\mathcal{D}_{tr}^{(m)}}, P^n_{\epsilon,\eta^*}\right)\right] = o(1)$$

Going back to step B.3, consider another sequence of probability distributions $(Q^{(m)})_{m\in\mathbb{N}}$ in $\mathcal{P}_0$ such that

$$\sup_{P \in \mathcal{P}_0} \Delta_P^{(m)} \leq \Delta_{Q^{(m)}}^{(m)} + \frac{1}{m}$$

where

$$\Delta_{Q^{(m)}}^{(m)} = \min\left\{ \left| \mathbb{E}_{Q^{(m)}}\left[d_{\text{TV}}\left(Q^{(m)}{}^n_{\hat{\epsilon},\hat{\eta}|\mathcal{D}_{tr}^{(m)}}, Q^{(m)}{}^n_{\epsilon,\hat{\eta}|\mathcal{D}_{tr}^{(m)}}\right)\right] - d_{\text{TV}}(Q^{(m)}{}^n_{\epsilon^*,\eta^*}, Q^{(m)}{}^n_{\epsilon,\eta^*}) \right| + d_{\text{TV}}(Q^{(m)}{}^n_{\epsilon^*,\eta^*}, Q^{(m)}{}^n_{\epsilon,\eta^*}), \right.$$

$$\left. \left| \mathbb{E}_{Q^{(m)}}\left[d_{\text{TV}}\left(Q^{(m)}{}^n_{\hat{\epsilon},\hat{\eta}|\mathcal{D}_{tr}^{(m)}}, Q^{(m)}{}^n_{\hat{\epsilon},\eta|\mathcal{D}_{tr}^{(m)}}\right)\right] - d_{\text{TV}}(Q^{(m)}{}^n_{\epsilon^*,\eta^*}, Q^{(m)}{}^n_{\epsilon^*,\eta}) \right| + d_{\text{TV}}(Q^{(m)}{}^n_{\epsilon^*,\eta^*}, Q^{(m)}{}^n_{\epsilon^*,\eta}) \right\}$$

$$- \min\{d_{\text{TV}}(Q^{(m)}{}^n_{\epsilon^*,\eta^*}, Q^{(m)}{}^n_{\epsilon,\eta^*}), d_{\text{TV}}(Q^{(m)}{}^n_{\epsilon^*,\eta^*}, Q^{(m)}{}^n_{\epsilon^*,\eta})\}$$

Because of continuity of $\min$ and

$$\sup_{P \in \mathcal{P}_0} \left| \mathbb{E}_P\left[d_{\text{TV}}\left(P^n_{\hat{\epsilon},\hat{\eta}|\mathcal{D}_{tr}^{(m)}}, P^n_{\epsilon,\hat{\eta}|\mathcal{D}_{tr}^{(m)}}\right)\right] - d_{\text{TV}}(P^n_{\epsilon^*,\eta^*}, P^n_{\epsilon,\eta^*}) \right| = o(1)$$

and

$$\sup_{P \in \mathcal{P}_0} \left| \mathbb{E}_P\left[d_{\text{TV}}\left(P^n_{\hat{\epsilon},\hat{\eta}|\mathcal{D}_{tr}^{(m)}}, P^n_{\hat{\epsilon},\eta|\mathcal{D}_{tr}^{(m)}}\right)\right] - d_{\text{TV}}(P^n_{\epsilon^*,\eta^*}, P^n_{\epsilon^*,\eta}) \right| = o(1)$$

we have that

$$\Delta_{Q^{(m)}}^{(m)} =$$
$$= \min\{d_{\text{TV}}(Q^{(m)}{}^n_{\epsilon^*,\eta^*}, Q^{(m)}{}^n_{\epsilon,\eta^*}) + o(1), d_{\text{TV}}(Q^{(m)}{}^n_{\epsilon^*,\eta^*}, Q^{(m)}{}^n_{\epsilon^*,\eta}) + o(1)\}$$
$$\quad - \min\{d_{\text{TV}}(Q^{(m)}{}^n_{\epsilon^*,\eta^*}, Q^{(m)}{}^n_{\epsilon,\eta^*}), d_{\text{TV}}(Q^{(m)}{}^n_{\epsilon^*,\eta^*}, Q^{(m)}{}^n_{\epsilon^*,\eta})\}$$
$$= \min\{d_{\text{TV}}(Q^{(m)}{}^n_{\epsilon^*,\eta^*}, Q^{(m)}{}^n_{\epsilon,\eta^*}), d_{\text{TV}}(Q^{(m)}{}^n_{\epsilon^*,\eta^*}, Q^{(m)}{}^n_{\epsilon^*,\eta}))\}$$
$$\quad - \min\{d_{\text{TV}}(Q^{(m)}{}^n_{\epsilon^*,\eta^*}, Q^{(m)}{}^n_{\epsilon,\eta^*}), d_{\text{TV}}(Q^{(m)}{}^n_{\epsilon^*,\eta^*}, Q^{(m)}{}^n_{\epsilon^*,\eta})\} + o(1)$$
$$= o(1)$$

Finally implying, from step B.3,

$$\sup_{P \in \mathcal{P}_0} \left[\mathbb{E}_P[\varphi_\alpha^{\text{RESIT}}(\mathcal{D}_{te}^{(n)}, \mathcal{D}_{tr}^{(m)})] - \alpha - \min\{d_{\text{TV}}(P^n_{\epsilon^*,\eta^*}, P^n_{\epsilon,\eta^*}), d_{\text{TV}}(P^n_{\epsilon^*,\eta^*}, P^n_{\epsilon^*,\eta})\}\right] \leq o(1) \text{ as } m \to \infty$$

$\square$

## B.4  RBPT

**Theorem 4.4.** Suppose that Assumptions 4.1, 4.2, 4.3, 3.2, and 3.3 hold. If $n$ is a function of $m$ such that $n \to \infty$ and $n = o(m^\gamma)$ as $m \to \infty$, then

$$\mathbb{E}_P[\varphi_\alpha^{\text{RBPT}}(\mathcal{D}_{te}^{(n)}, \mathcal{D}_{tr}^{(m)})] = 1 - \Phi\left(\tau_\alpha - \sqrt{\frac{n}{\sigma_P^2}}\Omega_P^{\text{RBPT}}\right) + o(1)$$

where $o(1)$ denotes uniform convergence over all $P \in \mathcal{P}$ as $m \to \infty$ and

$$\Omega_P^{\text{RBPT}} = \Omega_{P,1}^{\text{RBPT}} - \Omega_{P,2}^{\text{RBPT}}$$

with

$$\Omega_{P,1}^{\text{RBPT}} \triangleq \mathbb{E}_P \left[ \ell \left( \int g_P^*(x, Z) dQ_{X|Z}^*(x), Y \right) - \ell \left( \int g_P^*(x, Z) dP_{X|Z}(x), Y \right) \right]$$

and

$$\underbrace{\Omega_{P,2}^{\text{RBPT}}}_{\text{Jensen's gap}} \triangleq \mathbb{E}_P \left[ \ell(g_P^*(X, Z), Y) - \ell \left( \int g_P^*(x, Z) dP_{X|Z}(x), Y \right) \right]$$

*Proof.* First, note that there must be[21] a sequence of probability measures in $\mathcal{P}$, $(P^{(m)})_{m \in \mathbb{N}}$, such that

$$\sup_{P \in \mathcal{P}} \left| \mathbb{E}_P[\varphi_\alpha^{\text{RBPT}}(\mathcal{D}_{te}^{(n)}, \mathcal{D}_{tr}^{(m)})] - 1 + \Phi\left(\tau_\alpha - \sqrt{\frac{n}{\sigma_P^2}} \Omega_P^{\text{RBPT}}\right) \right| \leq \left| \mathbb{E}_{P^{(m)}}[\varphi_\alpha^{\text{RBPT}}(\mathcal{D}_{te}^{(n)}, \mathcal{D}_{tr}^{(m)})] - 1 + \Phi\left(\tau_\alpha - \sqrt{\frac{n}{\sigma_{P^{(m)}}^2}} \Omega_{P^{(m)}}^{\text{RBPT}}\right) \right| + \frac{1}{m}$$

Then, it suffices to show that the RHS vanishes when we consider such a sequence $(P^{(m)})_{m \in \mathbb{N}}$.

Now, let us first decompose the test statistic $\Xi^{(n,m)}$ in the following way:

$$\Xi^{(n,m)} \triangleq \frac{\sqrt{n} \bar{T}^{(n,m)}}{\hat{\sigma}^{(n,m)}} =$$

$$= \frac{\sqrt{n}\left(\bar{T}^{(n,m)} - \mathbb{E}_{P^{(m)}}[T_1^{(m)} \mid \mathcal{D}_{tr}^{(m)}]\right)}{\hat{\sigma}^{(n,m)}} + \frac{\sqrt{n}\mathbb{E}_{P^{(m)}}[T_1^{(m)} \mid \mathcal{D}_{tr}^{(m)}]}{\hat{\sigma}^{(n,m)}}$$

$$= \frac{\sqrt{n}\left(\bar{T}^{(n,m)} - \mathbb{E}_{P^{(m)}}[T_1^{(m)} \mid \mathcal{D}_{tr}^{(m)}]\right)}{\hat{\sigma}^{(n,m)}} + \frac{\sqrt{n}\mathbb{E}_{P^{(m)}}[\ell(\hat{h}^{(m)}(Z_1), Y_1) - \ell(\hat{g}^{(m)}(X_1, Z_1), Y_1) \mid \mathcal{D}_{tr}^{(m)}]}{\hat{\sigma}^{(n,m)}}$$

$$= \frac{\sqrt{n}\left(\bar{T}^{(n,m)} - \mathbb{E}_{P^{(m)}}[T_1^{(m)} \mid \mathcal{D}_{tr}^{(m)}]\right)}{\hat{\sigma}^{(n,m)}} + \frac{\sqrt{n}\mathbb{E}_{P^{(m)}}[\ell(\int \hat{g}^{(m)}(x, Z_1) d\hat{Q}_{X|Z}^{(m)}(x), Y_1) - \ell(\hat{g}^{(m)}(X_1, Z_1), Y_1) \mid \mathcal{D}_{tr}^{(m)}]}{\hat{\sigma}^{(n,m)}}$$

$$= \frac{\sqrt{n}\left(\bar{T}^{(n,m)} - \mathbb{E}_{P^{(m)}}[T_1^{(m)} \mid \mathcal{D}_{tr}^{(m)}]\right)}{\hat{\sigma}^{(n,m)}} +$$

$$+ \frac{\sqrt{n}\mathbb{E}_{P^{(m)}}[\ell(\int \hat{g}^{(m)}(x, Z_1) d\hat{Q}_{X|Z}^{(m)}(x), Y_1) - \ell(\int \hat{g}^{(m)}(x, Z_1) dQ_{X|Z}^*(x), Y_1) \mid \mathcal{D}_{tr}^{(m)}]}{\hat{\sigma}^{(n,m)}}$$

$$+ \frac{\sqrt{n}\mathbb{E}_{P^{(m)}}[\ell(\int \hat{g}^{(m)}(x, Z_1) dQ_{X|Z}^*(x), Y_1) - \ell(\int g_{P^{(m)}}^*(x, Z_1) dQ_{X|Z}^*(x), Y_1) \mid \mathcal{D}_{tr}^{(m)}]}{\hat{\sigma}^{(n,m)}}$$

$$+ \frac{\sqrt{n}\mathbb{E}_{P^{(m)}}[\ell(g_{P^{(m)}}^*(X_1, Z_1), Y_1) - \ell(\hat{g}^{(m)}(X_1, Z_1), Y_1) \mid \mathcal{D}_{tr}^{(m)}]}{\hat{\sigma}^{(n,m)}}$$

$$+ \frac{\sqrt{n}}{\hat{\sigma}^{(n,m)}} \mathbb{E}_{P^{(m)}} \left[ \left[ \ell\left(\int g_{P^{(m)}}^*(x, Z_1) dQ_{X|Z}^*(x), Y\right) - \ell\left(\int g_{P^{(m)}}^*(x, Z_1) dP_{X|Z}^{(m)}(x), Y\right) \right] + \right.$$

$$\left. + \left[ \ell\left(\int g_{P^{(m)}}^*(x, Z_1) dP_{X|Z}^{(m)}(x), Y\right) - \ell(g_{P^{(m)}}^*(X_1, Z_1), Y_1) \right] \right]$$

$$= \frac{\sqrt{n} W_{1,P^{(m)}}^{(m)}}{\hat{\sigma}^{(n,m)}} + \frac{\sqrt{n} W_{2,P^{(m)}}^{(m)}}{\hat{\sigma}^{(n,m)}} + \frac{\sqrt{n} W_{3,P^{(m)}}^{(m)}}{\hat{\sigma}^{(n,m)}} + \frac{\sqrt{n} W_{4,P^{(m)}}^{(m)}}{\hat{\sigma}^{(n,m)}} + \frac{\sqrt{n} \Omega_{P^{(m)}}^{\text{RBPT}}}{\hat{\sigma}^{(n,m)}}$$

---

[21] Because of the definition of sup.

Given that $n$ is a function of $m$, we omit it when writing the $W_{j,P^{(m)}}^{(m)}$'s. Define $\sigma_{P^{(m)}}^{(m)} \triangleq \sqrt{\mathsf{Var}_{P^{(m)}}[T_1^{(m)} \mid \mathcal{D}_{tr}^{(m)}]}$ and see that

$$\mathbb{E}_{P^{(m)}}[\varphi_\alpha^{\mathrm{RBPT}}(\mathcal{D}_{te}^{(n)}, \mathcal{D}_{tr}^{(m)})] = \tag{B.4}$$

$$= \mathbb{P}_{P^{(m)}}[p(\mathcal{D}_{te}^{(n)}, \mathcal{D}_{tr}^{(m)}) \leq \alpha]$$

$$= \mathbb{P}_{P^{(m)}}\left[1 - \Phi\left(\Xi^{(n,m)}\right) \leq \alpha\right]$$

$$= \mathbb{P}_{P^{(m)}}\left[\Xi^{(n,m)} \geq \tau_\alpha\right]$$

$$= \mathbb{P}_{P^{(m)}}\left[\frac{\sqrt{n}W_{1,P^{(m)}}^{(m)}}{\hat{\sigma}^{(n,m)}} + \frac{\sqrt{n}W_{2,P^{(m)}}^{(m)}}{\hat{\sigma}^{(n,m)}} + \frac{\sqrt{n}W_{3,P^{(m)}}^{(m)}}{\hat{\sigma}^{(n,m)}} + \frac{\sqrt{n}W_{4,P^{(m)}}^{(m)}}{\hat{\sigma}^{(n,m)}} + \frac{\sqrt{n}\Omega_{P^{(m)}}^{\mathrm{RBPT}}}{\hat{\sigma}^{(n,m)}} \geq \tau_\alpha\right]$$

$$= \mathbb{P}_{P^{(m)}}\left[\frac{\sqrt{n}W_{1,P^{(m)}}^{(m)}}{\sigma_{P^{(m)}}} + \frac{\sqrt{n}W_{2,P^{(m)}}^{(m)}}{\sigma_{P^{(m)}}} + \frac{\sqrt{n}W_{3,P^{(m)}}^{(m)}}{\sigma_{P^{(m)}}} + \frac{\sqrt{n}W_{4,P^{(m)}}^{(m)}}{\sigma_{P^{(m)}}} + \frac{\sqrt{n}\Omega_{P^{(m)}}^{\mathrm{RBPT}}}{\sigma_{P^{(m)}}} + \tau_\alpha - \tau_\alpha\frac{\hat{\sigma}^{(n,m)}}{\sigma_{P^{(m)}}} \geq \tau_\alpha\right]$$

$$= \mathbb{P}_{P^{(m)}}\left[\frac{\sqrt{n}W_{1,P^{(m)}}^{(m)}}{\sigma_{P^{(m)}}}\frac{\sigma_{P^{(m)}}^{(m)}}{\sigma_{P^{(m)}}} + \frac{\sqrt{n}W_{2,P^{(m)}}^{(m)}}{\sigma_{P^{(m)}}} + \frac{\sqrt{n}W_{3,P^{(m)}}^{(m)}}{\sigma_{P^{(m)}}} + \frac{\sqrt{n}W_{4,P^{(m)}}^{(m)}}{\sigma_{P^{(m)}}} + \tau_\alpha - \tau_\alpha\frac{\hat{\sigma}^{(n,m)}}{\sigma_{P^{(m)}}^{(m)}}\frac{\sigma_{P^{(m)}}^{(m)}}{\sigma_{P^{(m)}}} \geq \tau_\alpha - \frac{\sqrt{n}\Omega_{P^{(m)}}^{\mathrm{RBPT}}}{\sigma_{P^{(m)}}}\right]$$

$$= 1 - \Phi\left(\tau_\alpha - \sqrt{\frac{n}{\sigma_{P^{(m)}}^2}}\Omega_{P^{(m)}}^{\mathrm{RBPT}}\right) + o(1) \tag{B.5}$$

Implying that

$$\sup_{P \in \mathcal{P}}\left|\mathbb{E}_P[\varphi_\alpha^{\mathrm{RBPT}}(\mathcal{D}_{te}^{(n)}, \mathcal{D}_{tr}^{(m)})] - 1 + \Phi\left(\tau_\alpha - \sqrt{\frac{n}{\sigma_P^2}}\Omega_P^{\mathrm{RBPT}}\right)\right| = o(1) \text{ as } m \to \infty$$

*Justifying step B.5.* First, from a central limit theorem for triangular arrays [6, Corollary 9.5.11], we have that

$$\sqrt{n}\left(\frac{W_{1,P^{(m)}}^{(m)}}{\sigma_{P^{(m)}}^{(m)}}\right) = \sqrt{n}\left(\frac{\bar{T}^{(n,m)} - \mathbb{E}_{P^{(m)}}[T_1^{(m)} \mid \mathcal{D}_{tr}^{(m)}]}{\sigma_{P^{(m)}}^{(m)}}\right) = \frac{1}{\sqrt{n}}\sum_{i=1}^{n}\left(\frac{T_i^{(m)} - \mathbb{E}_{P^{(m)}}[T_1^{(m)} \mid \mathcal{D}_{tr}^{(m)}]}{\sigma_{P^{(m)}}^{(m)}}\right) \Rightarrow N(0,1)$$

The conditions for the central limit theorem [6, Corollary 9.5.11] can be proven to hold like in Theorem 3.4's proof.

Second, we have that

$$\frac{\sqrt{n}W_{1,P^{(m)}}^{(m)}}{\sigma_{P^{(m)}}^{(m)}} - \left(\frac{\sqrt{n}W_{1,P^{(m)}}^{(m)}}{\sigma_{P^{(m)}}^{(m)}}\frac{\sigma_{P^{(m)}}^{(m)}}{\sigma_{P^{(m)}}} + \frac{\sqrt{n}W_{2,P^{(m)}}^{(m)}}{\sigma_{P^{(m)}}} + \frac{\sqrt{n}W_{3,P^{(m)}}^{(m)}}{\sigma_{P^{(m)}}} + \frac{\sqrt{n}W_{4,P^{(m)}}^{(m)}}{\sigma_{P^{(m)}}} + \tau_\alpha - \tau_\alpha\frac{\hat{\sigma}^{(n,m)}}{\sigma_{P^{(m)}}^{(m)}}\frac{\sigma_{P^{(m)}}^{(m)}}{\sigma_{P^{(m)}}}\right) =$$

$$= \frac{\sqrt{n}W_{1,P^{(m)}}^{(m)}}{\sigma_{P^{(m)}}^{(m)}}\left(1 - \frac{\sigma_{P^{(m)}}^{(m)}}{\sigma_{P^{(m)}}}\right) - \frac{\sqrt{n}W_{2,P^{(m)}}^{(m)}}{\sigma_{P^{(m)}}} - \frac{\sqrt{n}W_{3,P^{(m)}}^{(m)}}{\sigma_{P^{(m)}}} - \frac{\sqrt{n}W_{4,P^{(m)}}^{(m)}}{\sigma_{P^{(m)}}} +$$

$$+ \tau_\alpha\left(\frac{\hat{\sigma}^{(n,m)}}{\sigma_{P^{(m)}}^{(m)}} - 1\right)\left(\frac{\sigma_{P^{(m)}}^{(m)}}{\sigma_{P^{(m)}}} - 1\right) + \tau_\alpha\left(\frac{\sigma_{P^{(m)}}^{(m)}}{\sigma_{P^{(m)}}} - 1\right) + \tau_\alpha\left(\frac{\hat{\sigma}^{(n,m)}}{\sigma_{P^{(m)}}^{(m)}} - 1\right)$$

$$= o_p(1) \text{ as } m \to \infty$$

To see why the random quantity above converges to zero in probability, see that because of Assumption 3.3, Lemma[22] B.1, and continuous mapping theorem, we have that

$$\frac{\hat{\sigma}^{(n,m)}}{\sigma_{P^{(m)}}^{(m)}} - 1 = o_p(1) \text{ and } \frac{\sigma_{P^{(m)}}^{(m)}}{\sigma_{P^{(m)}}} - 1 = o_p(1) \text{ as } m \to \infty$$

Additionally, because of Assumptions 4.1, 4.2, and 4.3 and condition $n = o(m^\gamma)$, we have that

---

[22]We can apply this STFR's lemma because it still holds when we consider GCM's test statistic.

$$\left|\frac{\sqrt{n}}{\sigma_{P^{(m)}}}W_{2,P^{(m)}}^{(m)}\right| = \left|\frac{\sqrt{n}}{\sigma_{P^{(m)}}}\mathbb{E}_{P^{(m)}}\left[\ell\left(\int\hat{g}^{(m)}(x,Z_1)d\hat{Q}_{X|Z}^{(m)}(x),Y\right)-\ell\left(\int\hat{g}^{(m)}(x,Z_1)dQ_{X|Z}^*(x),Y\right)\mid\mathcal{D}_{tr}^{(m)}\right]\right|$$

$$\leq\frac{\sqrt{n}}{\sigma_{P^{(m)}}}\mathbb{E}_{P^{(m)}}\left[\left|\ell\left(\int\hat{g}^{(m)}(x,Z_1)d\hat{Q}_{X|Z}^{(m)}(x),Y\right)-\ell\left(\int\hat{g}^{(m)}(x,Z_1)dQ_{X|Z}^*(x),Y\right)\right|\mid\mathcal{D}_{tr}^{(m)}\right]$$

$$\leq\frac{L\sqrt{n}}{\sigma_{P^{(m)}}}\cdot\mathbb{E}_{P^{(m)}}\left[\left\|\int\hat{g}^{(m)}(x,Z_1)d\hat{Q}_{X|Z}^{(m)}(x)-\int\hat{g}^{(m)}(x,Z_1)dQ_{X|Z}^*(x)\right\|_2\mid\mathcal{D}_{tr}^{(m)}\right]$$

$$\leq\frac{L\sqrt{n}}{\sigma_{P^{(m)}}}\cdot\mathbb{E}_{P^{(m)}}\left[\left\|\int\hat{g}^{(m)}(x,Z_1)d\hat{Q}_{X|Z}^{(m)}(x)-\int\hat{g}^{(m)}(x,Z_1)dQ_{X|Z}^*(x)\right\|_1\mid\mathcal{D}_{tr}^{(m)}\right]$$

$$\leq\frac{L\sqrt{n}}{\sigma_{P^{(m)}}}\cdot\mathbb{E}_{P^{(m)}}\left[\int\left\|\hat{g}^{(m)}(x,Z_1)\right\|_1\cdot|\hat{q}_{X|Z}^{(m)}(x|Z)-q_{X|Z}^*(x|Z)|d\mu(x)\mid\mathcal{D}_{tr}^{(m)}\right]$$

$$\leq\frac{LM\sqrt{n}}{\sigma_{P^{(m)}}}\cdot\mathbb{E}_{P^{(m)}}\left[\int|\hat{q}_{X|Z}^{(m)}(x|Z)-q_{X|Z}^*(x|Z)|d\mu(x)\mid\mathcal{D}_{tr}^{(m)}\right]$$

$$=\frac{2LM\sqrt{n}}{\inf_{P\in\mathcal{P}}\sigma_P}\cdot\mathbb{E}_{P^{(m)}}\left[d_{\mathrm{TV}}(\hat{Q}_{X|Z}^{(m)},Q_{X|Z}^*)\mid\mathcal{D}_{tr}^{(m)}\right]$$

$$=\frac{2LM}{\inf_{P\in\mathcal{P}}\sigma_P}\cdot\left(o(m^\gamma)\mathcal{O}_p(m^{-2\gamma})\right)^{1/2}$$

$$=o_p(1)$$

$$\left|\frac{\sqrt{n}}{\sigma_{P^{(m)}}}W_{3,P^{(m)}}^{(m)}\right| = \left|\mathbb{E}_{P^{(m)}}\left[\ell\left(\int\hat{g}^{(m)}(x,Z_1)dQ_{X|Z}^*(x),Y\right)-\ell\left(\int g_{P^{(m)}}^*(x,Z_1)dQ_{X|Z}^*(x),Y\right)\mid\mathcal{D}_{tr}^{(m)}\right]\right|$$

$$\leq\frac{\sqrt{n}}{\sigma_{P^{(m)}}}\mathbb{E}_{P^{(m)}}\left[\left|\ell\left(\int\hat{g}^{(m)}(x,Z_1)dQ_{X|Z}^*(x),Y\right)-\ell\left(\int g_{P^{(m)}}^*(x,Z_1)dQ_{X|Z}^*(x),Y\right)\right|\mid\mathcal{D}_{tr}^{(m)}\right]$$

$$\leq\frac{L\sqrt{n}}{\sigma_{P^{(m)}}}\cdot\mathbb{E}_{P^{(m)}}\left[\left\|\int\hat{g}^{(m)}(x,Z_1)dQ_{X|Z}^*(x)-\int g_{P^{(m)}}^*(x,Z_1)dQ_{X|Z}^*(x)\right\|_2\mid\mathcal{D}_{tr}^{(m)}\right]$$

$$=\frac{L\sqrt{n}}{\sigma_{P^{(m)}}}\cdot\mathbb{E}_{P^{(m)}}\left[\left\|\int\hat{g}^{(m)}(x,Z_1)-g_{P^{(m)}}^*(x,Z_1)dQ_{X|Z}^*(x)\right\|_2\mid\mathcal{D}_{tr}^{(m)}\right]$$

$$\leq\frac{L\sqrt{n}}{\sigma_{P^{(m)}}}\cdot\mathbb{E}_{P^{(m)}}\left[\int\left\|\hat{g}^{(m)}(x,Z_1)-g_{P^{(m)}}^*(x,Z_1)\right\|_2 dQ_{X|Z}^*(x)\mid\mathcal{D}_{tr}^{(m)}\right]$$

$$=\frac{L\sqrt{n}}{\sigma_{P^{(m)}}}\cdot\mathbb{E}_{P^{(m)}}\left[\int\left\|\hat{g}^{(m)}(x,Z_1)-g_{P^{(m)}}^*(x,Z_1)\right\|_2\frac{dQ_{X|Z}^*}{dP_{X|Z}}(x)dP_{X|Z}(x)\mid\mathcal{D}_{tr}^{(m)}\right]$$

$$=\frac{L\sqrt{n}}{\sigma_{P^{(m)}}}\cdot\mathbb{E}_{P^{(m)}}\left[\left\|\hat{g}^{(m)}(X_1,Z_1)-g_{P^{(m)}}^*(X_1,Z_1)\right\|_2\frac{dQ_{X|Z}^*}{dP_{X|Z}}(X_1)\mid\mathcal{D}_{tr}^{(m)}\right]$$

$$\leq\frac{L\sqrt{n}}{\sigma_{P^{(m)}}}\cdot\left(\mathbb{E}_{P^{(m)}}\left[\left\|\hat{g}^{(m)}(X_1,Z_1)-g_{P^{(m)}}^*(X_1,Z_1)\right\|_2^2\mid\mathcal{D}_{tr}^{(m)}\right]\mathbb{E}_{P^{(m)}}\left[\left(\frac{dQ_{X|Z}^*}{dP_{X|Z}}(X_1)\right)^2\right]\right)^{1/2}$$

$$=\frac{L\sqrt{n}}{\sigma_{P^{(m)}}}\cdot\left(\mathbb{E}_{P^{(m)}}\left[\left\|\hat{g}^{(m)}(X_1,Z_1)-g_{P^{(m)}}^*(X_1,Z_1)\right\|_2^2\mid\mathcal{D}_{tr}^{(m)}\right]\mathbb{E}_{P^{(m)}}\left[\int\frac{dQ_{X|Z}^*}{dP_{X|Z}}(x)dQ_{X|Z}^*(x)\right]\right)^{1/2}$$

$$=\frac{L\sqrt{n}}{\sigma_{P^{(m)}}}\cdot\left(\mathbb{E}_{P^{(m)}}\left[\left\|\hat{g}^{(m)}(X_1,Z_1)-g_{P^{(m)}}^*(X_1,Z_1)\right\|_2^2\mid\mathcal{D}_{tr}^{(m)}\right]\mathbb{E}_{P^{(m)}}\left[\chi^2\left(Q_{X|Z}^*||P_{X|Z}\right)+1\right]\right)^{1/2}$$

$$\leq\frac{LR\sqrt{n}}{\inf_{P\in\mathcal{P}}\sigma_P}\cdot\left(\mathbb{E}_{P^{(m)}}\left[\left\|\hat{g}^{(m)}(X_1,Z_1)-g_{P^{(m)}}^*(X_1,Z_1)\right\|_2^2\mid\mathcal{D}_{tr}^{(m)}\right]\right)^{1/2}$$

$$=\frac{LR}{\inf_{P\in\mathcal{P}}\sigma_P}\cdot\left(o(m^\gamma)\mathcal{O}_p(m^{-\gamma})\right)^{1/2}$$

$$=o_p(1)$$

$$\left|\frac{\sqrt{n}}{\sigma_{P^{(m)}}}W_{4,P^{(m)}}^{(m)}\right| = \left|\mathbb{E}_{P^{(m)}}\left[\ell\left(\hat{g}^{(m)}(X_1,Z_1),Y_1\right) - \ell\left(g_{P^{(m)}}^*(X_1,Z_1),Y_1\right) \mid \mathcal{D}_{tr}^{(m)}\right]\right|$$

$$\leq \frac{\sqrt{n}}{\sigma_{P^{(m)}}}\mathbb{E}_{P^{(m)}}\left[\left|\ell\left(\hat{g}^{(m)}(X_1,Z_1),Y_1\right) - \ell\left(g_{P^{(m)}}^*(X_1,Z_1),Y_1\right)\right| \mid \mathcal{D}_{tr}^{(m)}\right]$$

$$\leq \frac{L\sqrt{n}}{\sigma_{P^{(m)}}}\cdot\mathbb{E}_{P^{(m)}}\left[\left\|\hat{g}^{(m)}(X_1,Z_1) - g_{P^{(m)}}^*(X_1,Z_1)\right\|_2 \mid \mathcal{D}_{tr}^{(m)}\right]$$

$$\leq \frac{L\sqrt{n}}{\inf_{P\in\mathcal{P}}\sigma_P}\cdot\left(\mathbb{E}_{P^{(m)}}\left[\left\|\hat{g}^{(m)}(X_1,Z_1) - g_{P^{(m)}}^*(X_1,Z_1)\right\|_2^2 \mid \mathcal{D}_{tr}^{(m)}\right]\right)^{1/2}$$

$$= \frac{L}{\inf_{P\in\mathcal{P}}\sigma_P}\cdot\left(o(m^\gamma)\mathcal{O}_p(m^{-\gamma})\right)^{1/2}$$

$$= o_p(1)$$

where $M$ is an upper bound for the sequence $\left(\left\|\hat{g}^{(m)}(x,z)\right\|_1\right)_{m\in\mathbb{N}}$ (Assumption 4.3) and $R$ is an upper bound for the sequence $\left(\left(\mathbb{E}_{P^{(m)}}\left[\chi^2\left(Q_{X|Z}^*||P_{X|Z}\right)+1\right]\right)^{1/2}\right)_{m\in\mathbb{N}}$ (Assumption 4.2).

Finally,

$$\frac{\sqrt{n}W_{1,P^{(m)}}^{(m)}}{\sigma_{P^{(m)}}^{(m)}}\frac{\sigma_{P^{(m)}}^{(m)}}{\sigma_{P^{(m)}}} + \frac{\sqrt{n}W_{2,P^{(m)}}^{(m)}}{\sigma_{P^{(m)}}} + \frac{\sqrt{n}W_{3,P^{(m)}}^{(m)}}{\sigma_{P^{(m)}}} + \frac{\sqrt{n}W_{4,P^{(m)}}^{(m)}}{\sigma_{P^{(m)}}} + \tau_\alpha - \tau_\alpha\frac{\hat{\sigma}^{(n,m)}}{\sigma_{P^{(m)}}^{(m)}}\frac{\sigma_{P^{(m)}}^{(m)}}{\sigma_{P^{(m)}}} =$$

$$= \frac{\sqrt{n}W_{1,P^{(m)}}^{(m)}}{\sigma_{P^{(m)}}^{(m)}} - \left[\frac{\sqrt{n}W_{1,P^{(m)}}^{(m)}}{\sigma_{P^{(m)}}^{(m)}} - \left(\frac{\sqrt{n}W_{1,P^{(m)}}^{(m)}}{\sigma_{P^{(m)}}^{(m)}}\frac{\sigma_{P^{(m)}}^{(m)}}{\sigma_{P^{(m)}}} + \frac{\sqrt{n}W_{2,P^{(m)}}^{(m)}}{\sigma_{P^{(m)}}} + \frac{\sqrt{n}W_{3,P^{(m)}}^{(m)}}{\sigma_{P^{(m)}}} + \frac{\sqrt{n}W_{4,P^{(m)}}^{(m)}}{\sigma_{P^{(m)}}} + \tau_\alpha - \tau_\alpha\frac{\hat{\sigma}^{(n,m)}}{\sigma_{P^{(m)}}^{(m)}}\frac{\sigma_{P^{(m)}}^{(m)}}{\sigma_{P^{(m)}}}\right)\right.$$

$$= \frac{\sqrt{n}W_{1,P^{(m)}}^{(m)}}{\sigma_{P^{(m)}}^{(m)}} + o_p(1) \Rightarrow N(0,1)$$

by Slutsky's theorem. Because $N(0,1)$ is a continuous distribution, we have uniform convergence of the distribution function [27][Chapter 8, Exercise 5], and we do not have to worry about the fact that $\tau_\alpha - \frac{\sqrt{n}\Omega_{P^{(m)}}^{\text{RBPT}}}{\sigma_{P^{(m)}}}$ depends on $m$.

$\square$

# C  Experiments

## C.1  Running times

**Artificial-data experiments.**  Regarding running times (average per iteration), RBPT took $5 \cdot 10^{-4}s$ to run, RBPT2 took $1.9s$, STFR took $7.5^{-4}s$, RESIT took $1.3 \cdot 10^{-1}s$, GCM took $6 \cdot 10^{-4}s$, CRT took $1.7 \cdot 10^{-2}s$, and CPT took $6.2 \cdot 10^{-1}s$, all in a MacBook Air 2020 M1.

**Real-data experiments.**  Regarding running times (average per iteration), RBPT took $1.1 \cdot 10^{-1}s$ to run, RBPT2 took $2.4 \cdot 10^{-1}s$, STFR took $10^{-3}s$, GCM took $10^{-3}s$, CRT took $2.5 \cdot 10^{-2}s$, and CPT took $7.2 \cdot 10^{-1}s$, all in a MacBook Air 2020 M1.

## C.2  Extra results

We include extra experiments in which $Y \mid X, Z$ has skewed normal distributions with location $\mu = cX + a^\top Z + \gamma(b^\top Z)^2$, scale $\sigma = 1$, and shape $s = 3$ (shape $s = 0$ lead to the normal distribution). From the following plots, we can learn that the skewness often impacts negatively in Type-I error control.

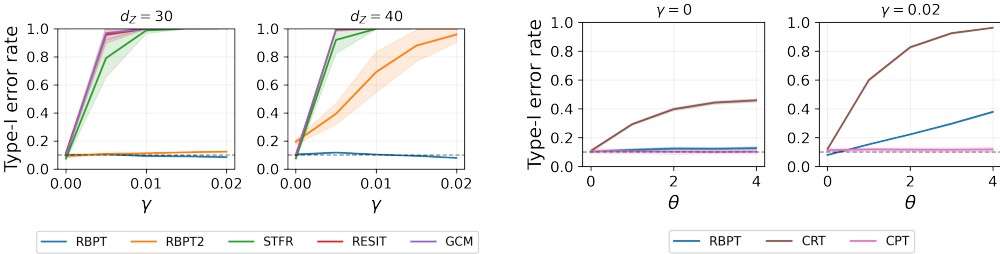

Figure 7: Type-I error rates ($c = 0$). In the first two plots, we set $\theta = 0$ for RBPT, permitting Type-I error control across different $d_Z$ values (Theorem 4.4), while RBPT2 allows Type-I error control for moderate $d_Z$. All the baselines fail to control Type-I errors regardless of $d_Z$. The last two plots illustrate that CRT emerges as the least robust test in this context, succeeded by RBPT and CPT.

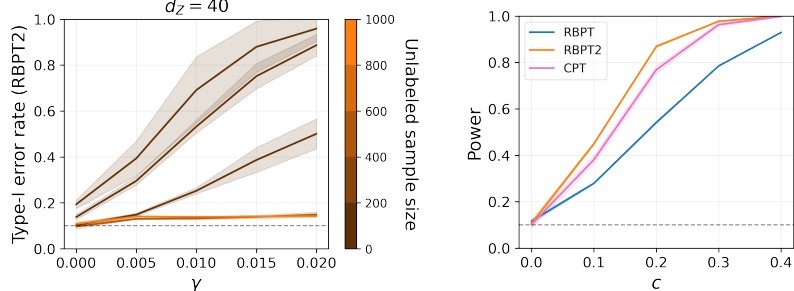

Figure 8: (i) Making RBPT2 more robust using unlabeled data. With $d_Z = 40$, we gradually increase the unlabeled sample size from 0 to 1000 when fitting $\hat{h}$. The results show that a larger unlabeled sample size leads to effective Type-I error control. Even though we present this result for RBPT2, the same pattern is expected for RBPT in the presence of unlabeled data. (ii) Power curves for different methods. We compare our methods with CPT (when $\theta = 0$), which seems to have practical robustness against misspecified inductive biases. RBPT2 and CPT have similar power while RBPT is slightly more conservative.

