# OpenReview forum: "Conditional independence testing under misspecified inductive biases"
_NeurIPS.cc/2023/Conference — NeurIPS 2023 spotlight_

### Official Review · Reviewer_hFX5 · 2023-06-26

**Soundness:** 4 excellent
**Presentation:** 3 good
**Contribution:** 3 good
**Rating:** 7
**Confidence:** 3

**Summary:**

The paper proposes a new Rao-Blackwellized Predictor Test which is a regression-based conditional independence test. It was shown to be robust in misspecification settings.

**Strengths:**

The paper is in general well-written and considers an important scenario which is not covered by standard conditional independence tests. The new approach guarantees Type-1 error control (asymptotically).

**Weaknesses:**

The paper is quite technical (what is good), however, a number of important details are in the Appendix what is not always practical.

**Questions:**

I would like a clarification for the phrase "even models capable of universal approximation may not accurately estimate the Bayes predictor...". What is exactly meant? Do you mean that if some hyper-parameters are not well-chosen then even a well-specified model does not approach the Bayes error? Even asymptotically? However, in an asymptotic case hyper-parameters do not play any role?

On page 3, Figure 4 is mentioned. Where is it?

In Section 5, "real data experiments": X represents the driving risk and X is a minority indicator, but what is Y? Is it the price? It would be helpful to write it explicitly.

"For RBPT2, we use a CatBoost regressor...": could you detail how you apply the CatBoost to get the Rao-Blackwellized predictor? What kind of modifications are made to the original CatBoost?

---

> ### Author Rebuttal · Authors · 2023-08-03
>
> ## Thank you
>
> We want to thank you for your time and work on our paper.
>
> We believe we have addressed your questions in our comments. Please let us know if you have further questions or comments.
>
> Best regards,
>
> Authors
>
> --------------
>
> ## Rebuttal
>
> **Question about universal approximators.** We intend to communicate here that regardless of the model's capacity for universal approximation, the training algorithm essentially narrows the potential model class to a smaller class of "feasible" models (perhaps not in the hard-constraints sense), even in the asymptotic case. This bias can make the Bayes error unreachable. Therefore, choosing hyperparameters is a crucial step.
>
> **Figure 4 missing.** It was a typo that will be corrected. We wanted to mention Figure 1 instead. Thanks for pointing this out.
>
> **Meaning of $Y$ in the real data experiments.** In that case, $Y$ is the insurance price. We will make it more explicit.
>
> **Question about CatBoost.** We appreciate your interest in the implementation details of the Rao-Blackwellized predictor using CatBoost in our RBPT2 methodology. In this context, CatBoost is employed to predict the output of the initial predictor, $\\hat{g}(X,Z)$, utilizing $Z$ as the input to the boosting algorithm. We have not made any modifications to the standard CatBoost algorithm, but we apply it in a way that aids the execution of the RBPT2 test when estimating $P_{X| Z}$ proves challenging. For a more comprehensive explanation of this process and the underlying rationale, we invite you to consult the paper subsection "Running RBPT when it is hard to estimate $P_{X| Z}$: the RBPT2."

---

> > ### Comment · Reviewer_hFX5 · 2023-08-12
> >
> > I acknowledge that I read the authors response.

---

### Official Review · Reviewer_Hib1 · 2023-07-06

**Soundness:** 3 good
**Presentation:** 3 good
**Contribution:** 3 good
**Rating:** 8
**Confidence:** 4

**Summary:**

The authors study model-based tests of conditional independence under conditions where the models may be biased. They propose new approximations and bounds on testing errors resulting from bias in model-based CI tests. They also introduce RBPT, a new model-based CI test and evaluate it against several alternatives.


**Strengths:**

The paper is well-written and provides substantial theoretical derivation of the Type I error rate and power of the proposed test.

The authors address a very important problem that is under-appreciated. Many researchers would assume (mistakenly) that non-parametric conditional independence testing is a solved problem. In reality, there exists no generally accepted solution, despite substantial prior work.

Furthermore, the authors work on a very important aspect of the problem. Model misspecification, as they term it, appears to be both ubiquitous and difficult to detect and diagnose. Making model-based CI tests that are robust to this source of bias would represent a major step forward.

The authors correctly point out that, despite a variety of deep models being *capable* of being universal function approximators, that does not mean that unbiased models are *actually learned in practice*.

Empirical evaluation of CI tests is difficult, at best. However, the authors provide both simulation and real-data experiments that provide some evidence that their approach delivers on its promises.

**Weaknesses:**

As the authors note, the term “model misspecification” has a long history in statistics (and some parts of machine learning), and it seems a stretch to expand that term to include non-optimal parameter settings due to the limitations of a given training algorithm. There is certainly an issue here, but calling it “model misspecification” is likely to cause some initial confusion for readers. The authors are quite clear that they are redefining the term, but seeking an alternative term may be more helpful to readers. Some options include “non-optimal training”, “underspecification”, and “non-identifiability”, though I suspect there is an even better term.

In the caption of Figure 3, the authors state that “RBPT and RBPT2 have better Type-I error control compared to other methods.” However, STFR fairly clearly outperforms RBPT2 in terms of absolute error with respect to the 0.10 gold standard and STFR usually outperforms RBPT.

One minor correction: The text describing Figure 1 says that the Type I error rate was set to 0.01, but the figure shows that it was set to 0.10. Furthermore, that text refers to Figure 4, which does not appear to exist in the paper.

**Questions:**

(none)

**Limitations:**

There appear to be relatively few negative impacts to the authors' work. Indeed, it facilitates *avoiding* important classes of errors that could have substantial negative impacts.

---

> ### Author Rebuttal · Authors · 2023-08-03
>
> ## Thank you
>
> We greatly appreciate your work on our paper.
>
> Thank you for the raised points. We have addressed them below. Please let us know if you have any other comments or questions.
>
> Best regards,
>
> Authors
>
> ---------------------
>
> ## Rebuttal
>
> **Superiority of RBPT/RBPT2 over STFR in the real data experiment.** Indeed, if we consider the "absolute error" as the metric, your observation holds. However, significance testing procedures often prioritize maintaining control over the Type-I error (having a Type-I error rate less than the significance level $\\alpha$). Consequently, falling below the $\\alpha=10\\%$ threshold (as RBPT/RBPT2 in most cases) is more desirable than above (as  STFR  in most cases).
>
> **Terminology.** We acknowledge your valuable input regarding the possible confusion that might arise from our use of the term "model misspecification," given its traditional connotations in the field. We will certainly consider your advice and reassess our terminology choices in the next version of this paper.
>
> **Typos.** Thank you for pointing out the typos. We will correct the main text.

---

> > ### Comment · Reviewer_Hib1 · 2023-08-21
> >
> > I acknowledge that I read the authors response, and I remain strongly supportive of acceptance.

---

### Official Review · Reviewer_8LZA · 2023-07-06

**Soundness:** 4 excellent
**Presentation:** 3 good
**Contribution:** 3 good
**Rating:** 7
**Confidence:** 2

**Summary:**

This paper presents new robustness results for three regression-based conditional independence (CI) tests, namely, Significance Test of Feature Relevance (STFR), Generalized Covariance Measure (GCM) test, and REgression with Subsequent Independence Test (RESIT). It introduces a new CI test, the Rao-Blackwellized Predictor Test (RBPT), designed to be more robust to model misspecification. Theoretical findings and experimental results show that the RBPT is able to control Type-I error effectively even when models are misspecified.

**Strengths:**

- Well written and structured. Thanks
- Great intro to related work.
- Addresses a significant problem
- Provides theoretical and empirical insights.

**Weaknesses:**

- The authors make a long and good argument, that misspecification needs to account for the training algorithm. It seems that this is a natural result of the model posing a non-convex or underspecified optimization problem for training. Therefore, I'm not sure this needs to be demonstrated in such depth. (Please correct me, if I'm misunderstanding this)
- Could have more empirical validation.

**Questions:**

- In the eq. between lines 76, and 77. Why do you write " Y | X, Z ~ ... " and not just " Y | Z " ?


**Limitations:**

n.a.

---

> ### Author Rebuttal · Authors · 2023-08-03
>
> ## Thank you
>
> We greatly appreciate your feedback on our paper.
>
> Thank you for your questions. We have addressed them below. Please let us know if you have any other comments or questions.
>
> Best regards,
>
> Authors
>
> ------------------
>
> ## Rebuttal
>
> **Misspecification discussion.** You're correct that it is indeed a natural outcome. However, given the centrality of model misspecification in our paper, we felt a comprehensive discussion was necessary, especially since our definition of "misspecification" differs from the conventional one in the statistics literature.
>
> **Question about math notation.** The notation $Y|X, Z \sim$ was used as opposed to $Y|Z  \sim$ to highlight that $Y$ and $X$ are conditionally independent when given $Z$. That is evident because the distribution of $Y|X, Z$ does not depend on $X$ even though we included that variable in "$Y|X, Z$."
>
> **More empirical evaluation.**  We have conducted three additional experiments, the details of which can be found below and in the uploaded PDF document.
>
> 1. *High-dimensional $Z$ and use of unlabeled data:* With $d_Z=40$ (a challenging scenario for RBPT2, as seen in the appendix), we gradually increased the unlabeled sample size from 0 to 500. The results, detailed in the uploaded PDF, show that a larger unlabeled sample size effectively controls the Type-I error rate, indicating a successful approximation of the Rao-Blackwellized predictor even when $d_Z$ is big.
>
> 2. *RBPT robustness in challenging scenarios:* Simultaneously adjusting $\gamma$ and $\theta$ across a range of values, we observed that RBPT outperformed CPT in robustness for $\theta\leq 4$. However, for $\theta>4$, RBPT encountered challenges controlling the Type-I error rate for large $\gamma$ values. Despite this, the full set of experiments underscores RBPT's robustness, making it an especially favorable approach in instances where it is challenging to estimate the functional form of $p_{X|Z}$ (density of $P_{X|Z}$). Contrary to CPT, RBPT sidesteps this challenge using Monte Carlo integration combined with GAN-like sampling techniques. Furthermore, CPT is more sensitive in experiments with real data.
>
> 3. *Detecting discrimination against minorities using fully real data.* In this experiment, we compare the output of different conditional independence tests regarding power. Roughly speaking, the methods point to the same qualitative result that discrimination against minorities in ZIP codes is most evident in Illinois, followed by Texas, Missouri, and California. These findings corroborate Angwin and colleagues' findings, indicating that our methodology has satisfactory power while maintaining Type-I error control (which the benchmarks violate).

---

> > ### Comment · Reviewer_8LZA · 2023-08-14
> > **Ack**
> >
> > Acknowledged. Thank you.

---

### Official Review · Reviewer_t2AA · 2023-07-09

**Soundness:** 2 fair
**Presentation:** 3 good
**Contribution:** 2 fair
**Rating:** 6
**Confidence:** 1

**Summary:**

This paper discusses the challenges of conditional independence (CI) testing in modern statistics and machine learning. The authors note that many modern methods for CI testing rely on supervised learning methods to learn regression functions or Bayes predictors, but these methods can be unreliable when the models are misspecified. The paper proposes new approximations or upper bounds for the testing errors of three regression-based tests that depend on misspecification errors. The authors present new robustness results for three relevant regression-based CI tests and derive approximations or upper bounds for the testing errors that explicitly depend on the misspecification of the prediction models. The paper concludes that more attention should be given to theoretically understanding the effects of misspecification on CI hypothesis testing and that current regression-based methods are usually not designed to be robust against misspecification errors, making CI testing less reliable.

**Strengths:**

The writing is good.

The introduction on model misspecification is good.

**Weaknesses:**

Lack of introducing conditional independence testing.

RBPT2 is powerful but it cost much more time compared with other methods, such as STFR, GCM. Lack of analysis the time complexity of each designed method.

Only one real dataset. This method should be applied on more real application and dataset, like STFR which is applied on MNIST handwritten digits, Mechanisms of Action (MoA) prediction for new drugs and Chest X-rays for pneumonia diagnosis.

**Questions:**

Why are regression-based and simulation-based tests more appealing than the other two tests?

What is type-I error?

Why there are no result on Illinois and Texas in p-value of Figure 3?

In Figure 2, why you did not have results of RBPT2 on \theta VS Type-I error rate?

Since you compared your designed method with STFP, why not choose the same dataset as STFP to show the efficiency of your method?

---

> ### Author Rebuttal · Authors · 2023-08-03
>
> ## Thank you
>
> Thank you for your time and work on our paper.
>
> We believe we have addressed your concerns and questions in our comments. Please let us know if you have any other questions.
>
> Best regards,
>
> Authors
>
> ______________
>
> ## Rebuttal
> **Time complexity of RBPT and RBPT2.** The running times of RBPT and RBPT2 primarily hinge on the underlying supervised or generative models we employ, mirroring the situation in STFR, GCM, CRT, CPT, and similar methods. Once the training step is completed, testing is conducted in linear time (in the number of testing samples) for RBPT, RBPT2, STFR, GCM, etc.
>
> **Results for Illinois and Texas.** The p-values for Illinois and Texas are nearly zero, rendering them practically invisible on the plot.
>
> **Results for RBPT in the second plot of Figure 2.** This is because $\theta$ is intrinsically tied to the misspecification of the model for $P_{X|Z}$. Fitting that model is not required for RBPT2, hence the omission.
>
> **Regression and simulation-based tests.** Regression-based and simulation-based tests hold appeal as they take advantage of the most recent and powerful supervised and generative methods. These tests are also known for their scalability with large datasets in terms of features and samples.
>
> **Definition for Type-I error.** Type-I error arises in hypothesis testing when a true null hypothesis gets wrongly rejected, leading to false discovery.
>
> **Real data applications.** Our goal in the experiments section is to benchmark RBPT/RBPT2 against a range of methods, including STFR, GCM, RESIT, CPT, and CRT, each of which uses different datasets in their original papers. We decided to use the car insurance dataset, which represents a key application area for conditional independence tests and is compatible with almost all these methods (except RESIT).

---

> > ### Comment · Reviewer_t2AA · 2023-08-21
> >
> > Thank you for your rebuttal. I have increased the score.

---

### Official Review · Reviewer_zPGB · 2023-07-25

**Soundness:** 4 excellent
**Presentation:** 1 poor
**Contribution:** 3 good
**Rating:** 7
**Confidence:** 3

**Summary:**

The paper under review offers a thoughtful exploration of how model misspecification influences conditional independence testing. The significance and practical applications of such testing are aptly underscored in the introduction. The use of a toy example to demonstrate the impact of model misspecification on Type I error is a compelling approach that effectively propels their study. However, the paper's structure could be improved for better readability. While the work is technically exhaustive and appears sound, the layout requires refinement. For instance, the discussion and conclusion section seems truncated due to an overflow of technical details, presumably in an attempt to conform to the page limit.

Additionally, the paper presents an insightful study on Type I error bounds and illuminates how factors in model misspecification might influence the performance of our model. This is examined across three leading methods of regression-based conditional independence testing, STFR, GCM, and RESIT; a commendable breadth of investigation. On the other hand, there lies a lot of  ingenuity in the application of sufficient statistics and the Rao-Blackwell theorem to devise a conditional independence test - an approach that stands out as particularly clever.

Despite the paper's notable theoretical contribution and illumination of model misspecification implications, its current state warrants a rejection recommendation. The primary rationale lies in the sparse experimental validation, which leaves room for further expansion. While a more comprehensive set of experimental results could potentially sway my assessment towards endorsement, the manuscript, as it stands, along with the limited set of experiments, doesn't quite meet the high standards necessary for acceptance. Thus, my score, on balance, falls on the lower side.


**Strengths:**

1. The paper exhibits a robust theoretical foundation, ingeniously leveraging the principles of sufficient statistics and the Rao-Blackwell theorem to address conditional independence testing. This innovative application, to the best of my understanding, introduces a novel concept and forms a significant contribution.
2. The paper provides extensive misspecification bounds that comprehensively span across various regression-based methods. Moreover, the experiments also benchmark against simulation-based approaches.
3. The exploration of examples in Appendix A.2 is highly instructive, where model misspecification can influence and Type I error of our test significantly. In addition, the choice of synthetic experiments adapted from Berrett et al. serves as a fitting means of demonstrating the concept in this study.
4. In an innovative deviation from the traditional setting, this paper presents proofs across a series of models that converge to a pre-established class of models, thereby aligning well with contemporary machine learning paradigms


**Weaknesses:**

1. Acknowledging that the paper's paramount contribution resides in its theoretical development, I must express my reservations regarding the experimental element, which appears somewhat insufficient. To bolster the validation process, I've proposed additional experimental situations in the "Questions" section. As it stands, the validation primarily leans on a single toy experimental setup, and a comprehensive benchmarking across different methods in a real-world setup is lacking.
2. In terms of presentation, the paper could benefit from some refinement. Notably, a significant portion of the proof concepts across different Conditional Independence Testing methods appears to overlap. A majority of these proofs could have been concisely presented by adopting lemmas that generalize across multiple methods, thus increasing reader engagement. Also, the main body is currently dense with scattered technical terms that could be better consolidated.
3. Just a minor point of clarification, I'd like to delve into the details of the conducted toy experiments. Developing a unified framework that frames functions as a sequential path towards a distinct model group is truly commendable. Nonetheless, there's scope for refining its presentation. Specifically, the introduction gave the impression that this was a broad generalization of model misspecification, which it doesn't appear to be.
Upon incorporating regularization terms during training, it's important to note that while the overarching model class remains constant, the subset of models that surface must adhere to the regularization-imposed limits. Consequently, the model's prime focus may deviate from merely minimizing the Bayes predictor loss.
This deviation could result from either integrating regularization terms into the optimization formula, thus modifying the loss characteristics, or from imposing these terms as constraints, effectively limiting the model class.
Considering these factors, I'd suggest positioning your model misspecifications as a tool to tailor the concept to real-world machine learning environments, rather than representing it as a broad generalization of the concept of model misspecification itself.
4. One of the method's most significant practical implications, in my view, is the potential for semi-supervised learning on unlabeled data - a scenario where RBPT is likely to outshine RBPT2. Given its importance in practical real-world settings, this deserves an additional set of experiments. An easy experiment to consider would be to add a large set of unlabeled data to the toy experiment.


**Questions:**

1. Regarding eq. (3.1), it seems to me that $\hat{\sigma}$ mismatches the sample standard deviation of $T_i^{(m)}$; it is incorrectly expressed as a biased estimator. For simplicity, you might want to just denote $\bar{T}^{(n,m)}$ and $\hat{\sigma}^{(n, m)}$ as the sample mean and variance estimators and not include any formulas, referencing Dei et al. for looking up the exact derivations.
2. In the experiments section, especially concerning figure 2, I have several points of interest:
* It would be insightful to observe how RBPT behaves in terms of Type I error when $Q$ is misspecified. For instance, a scenario where $c=0$, $\theta=1$, and $\gamma$ is progressively increased could be worth examining. Alternatively, to evaluate how sensitive RBPT is to such misspecification, setting $\gamma$ to a larger value (say 0.02), and sweeping through different $\theta$ values to assess Type I error might be valuable. I believe this experiment is of extreme importance since, in most real-world situations, the total variation between $Q^*(X|Z)$ and $P(X|Z)$ is significant and may not be bounded by the Jensen gap. Assessing this sensitivity is pivotal to demonstrating the robustness of the setting. The second plot, where $\gamma = 0$, may not adequately reflect this robustness.
* It would be beneficial to include the values of other hyperparameters set in the caption of Figure 2. This would prevent the need to navigate back and forth through the text to understand the setup; also, I assumed that for the third plot $\gamma = \theta = 0$, but it was not mentioned in the main text.
3. Why hasn't the power of RBPT been directly compared with the baselines for the real-world examples? The reasons for not contrasting RBPT are somewhat elusive to me. While I concur that both RBPT and RBPT2 theoretically establish bounds for Type I error control under model misspecification, a quantitative comparison with baselines is still essential due to the trade-off between Type I and Type II errors. Although we don't expect RBPT to outperform other models in terms of power, it would be interesting to see if it possesses comparable power, as suggested by the third plot in Figure 2.
4. I am perplexed by the stark disparity between the performance of RBPT2 in low versus high-dimensional settings. It appears that an increase in the dimension of Z invariably leads to rejection of the null hypothesis, resulting in a rise in both power and Type I error rate. Could this be attributed to the need for a dimension normalization term in $b$? How are $b$ and $a$ generated per experiment, and how does $b^T Z$ scale with the dimension? Why doesn't this impact RBPT or CPT? After all, one of the main applications of regression-based conditional independence tests are high-dimensional settings, and it is important to address such concerns.


**Limitations:**

1. In the RBPT methodology, model misspecification is conceptualized as the difference between $Q^*(X|Z)$ and $P(X|Z)$, and between $g^*(x, y)$ and $f^*(x, y)$. This deviates from prior methods where the misspecification is solely defined between $g_1^*$ and $f_1^*$, and $g_2^*$ and $f_2^*$. While the latter defines misspecification between functions, the former is defined on an entire distribution.
Notably, even if we posit that the disparity between the predicted $\hat{Q}(X|Z)$ and $Q^*(X|Z)$ is infinitesimal, the divergence between $Q^*(X|Z)$ and $P(X|Z)$ remains far from negligible. Furthermore, the effectiveness of the entire method hinges on this misspecification gap ($\Omega_{P,1}^{\text{RBPT}}$) being smaller than the Jensen's gap, which places stringent restrictions on its applicability. While theoretically, this method relaxes the bound on the expected total variation between $Q^*$ and $P$ and sets it to $\frac{\Omega_{P,2}^{\text{RBPT}}}{M \cdot L}$, the $M$ and $L$ coefficients will be pretty large in practice. On the other hand, while RBPT2 appears to circumvent this issue, its theoretical grounding isn't as robust as that of RBPT, thereby reducing its perceived solidity.
I believe this issue will become more prominent in real-world settings and the toy experiments do not reveal this core flaw of the method. Therefore, the authors should have mentioned this limitation or run extra experiments to show the effect of this gap.
2. A prominent constraint associated with RBPT2 appears to be its performance when scaling $Z$. Unfortunately, this aspect is not addressed within the main paper text. It would be beneficial to incorporate some commentary on this issue, as it pertains to the most plausible plug-and-play application. Furthermore, I harbor concerns regarding the potential for error from the initial regression to percolate into the subsequent one, which could potentially escalate the overall margin of error. This is pointed out in Appendix A.6, and with the empirical results acting strangely with scaling (I also mentioned in the “Questions” section), I am concerned as to how much this can hinder the performance of RBPT2.

---

> ### Author Rebuttal · Authors · 2023-08-04
>
> ## Thank you
>
> Thank you for your time and constructive feedback. Your suggestions made our paper more complete.
>
> We believe we have addressed your requests and questions in our comments. Please let us know if you have further concerns.
>
> Best regards,
>
> Authors
>
> --------------
>
> ## Rebuttal
>
> **Semi-supervised learning.** Both RBPT and RBPT2 have the potential to improve performance when a large unlabeled dataset is available. This also applies to RBPT2 because fitting the second predictor does not necessitate any labels. Please check the new experiments.
>
> **Dimensionality of $Z$ and its effects on the testing procedure.** You have astutely noted that escalating the dimension of $Z$ could detrimentally influence RBPT2. Nevertheless, as demonstrated in our new experiments, this complication can be mitigated by employing a large unlabeled dataset to train the second predictor. Moreover, it is crucial to emphasize that a growing $d_Z$ is also likely to adversely impact STFR, GCM, CPT, CRT, and any other regression or simulation-based conditional testing procedures. This effect arises primarily because these methods initially necessitate the estimation of regression functions or conditional distributions, tasks that become increasingly complex with higher dimensions. In our older experiments, the impact of a growing $d_Z$ on RBPT and CPT is not apparent, largely due to our approach of not estimating $P_{X|Z}$ in artificial data experiments. Instead, we introduce "misspecification" via modifications in $\theta$. We acknowledge the importance of addressing this point and will ensure to include a discussion about it in our manuscript.
>
> **On the percolation of initial regression errors.** While your concerns about the potential cascading of errors from the initial regression to the subsequent one are understandable, it's important to note that this issue may not necessarily pose a problem given a sufficiently competent second model. This is a direct implication of Jensen's inequality: for an *arbitrary* initial model $g(x,z)$ and a convex loss function $\ell$, under $H_0$ we find that
> $$
> \\mathbb{E}[\\ell(h(Z),Y)] \\leq \\mathbb{E}[\\ell(g(X,Z),Y)]
> $$
>
> where $h(z)\triangleq \\mathbb{E}[g(X,Z)|Z=z]$ is the Rao-Blackwellized predictor. If our second predictor performs well, $h$ will be estimated accurately, and the Type-I error will be controlled. This concept is empirically validated in our new experiment, where we demonstrate that by increasing the size of the unlabeled sample - thus keeping the first predictor unaffected - the second predictor's performance improves, and the Type-I error rate is effectively controlled.
>
> **Comparing power in real data experiment.** We added the benchmarks when analyzing power; please check our third new experiment.
>
> **New experiments:** In response to your valuable feedback, we have conducted three additional experiments, the details of which can be found below and in the uploaded PDF document. These experiments address your concerns, shedding light on the capabilities and robustness of RBPT/RBPT2.
>
> 1. *High-dimensional $Z$ and use of unlabeled data:* With $d_Z=40$ (a challenging scenario for RBPT2, as seen in the appendix), we gradually increased the unlabeled sample size from 0 to 500. The results, detailed in the uploaded PDF, show that a larger unlabeled sample size effectively controls the Type-I error rate, indicating a successful approximation of the Rao-Blackwellized predictor even when $d_Z$ is big.
>
> 2. *RBPT robustness in challenging scenarios:* Simultaneously adjusting $\gamma$ and $\theta$ across a range of values, we observed that RBPT outperformed CPT in robustness for $\theta\leq 4$. However, for $\theta>4$, RBPT encountered challenges controlling the Type-I error rate for large $\gamma$ values. Despite this, the full set of experiments underscores RBPT's robustness, making it an especially favorable approach in instances where it is challenging to estimate the functional form of $p_{X|Z}$ (density of $P_{X|Z}$). Contrary to CPT, RBPT sidesteps this challenge using Monte Carlo integration combined with GAN-like sampling techniques. Furthermore, CPT is more sensitive in experiments with real data.
>
> 3. *Detecting discrimination against minorities using fully real data.* In this experiment, we compare the output of different conditional independence tests regarding power. Roughly speaking, the methods point to the same qualitative result that discrimination against minorities in ZIP codes is most evident in Illinois, followed by Texas, Missouri, and California. These findings corroborate Angwin and colleagues' findings, indicating that our methodology has satisfactory power while maintaining Type-I error control (which the benchmarks violate).
>
> **About RBPT2.** We acknowledge the limitations of RBPT2 you have highlighted (in point 1 of the "Limitations" section) and will ensure they are discussed in the subsequent version of our paper. We appreciate your attention to this detail.
>
> **Caption of Figure 2.** We are going to incorporate the values of the parameters within the figure caption and will also embed the observation "$\gamma=\theta=0$", which you accurately inferred, into the body of the manuscript. Thank you for your input!
>
> **Presentation and general clarifications.** We greatly value your input on refining our paper's presentation. We'll certainly consider your suggestions for enhancing readability and conciseness in the paper's next version.
>
> **Observation about sample standard deviation.** The bias in $\\hat{\\sigma}^{(n,m)}$ is asymptotically negligible and does not impact our results. Still, we appreciate your suggestion of referencing Dai's paper and will consider it for the upcoming revision.
>
> --------------------
> ### References
>
> [1] Julia Angwin, Jeff Larson, Lauren Kirchner, and Surya Mattu. Minority neighborhoods pay higher car insurance premiums than white areas with the same risk, April 2017.

---

> > ### Comment · Reviewer_zPGB · 2023-08-16
> >
> > Dear authors,
> >
> > After reviewing the rebuttal and considering the new experimental settings with unlabeled data, I find my concerns about scalability and dimensionality to be addressed. The inclusion of baselines in real-world experiments and promising RBPT robustness results have contributed to my satisfaction. Therefore, I am pleased to increase my score to a 7.
> >
> > Best,

---

> > > ### Author Response · Authors · 2023-08-19
> > >
> > > Thank you once more for your valuable feedback and now for raising your score.
> > >
> > > Best regards,
> > >
> > > Authors

---

### Author Rebuttal · Authors · 2023-08-04

We extend our sincere thanks to all reviewers and chairs. We have responded to all the points raised by reviewers individually and have attached a PDF containing additional experiments. The first of these experiments demonstrate that the robustness of RBPT2 in higher dimensions can be improved through the use of unlabeled data (data pairs consisting of $X$ and $Z$ only). The second experiment examines RBPT in more challenging scenarios and supports the assertion that it offers a practical and robust solution for conditional independence testing. The third one shows that RBPT/RBPT2 output on the car insurance experiment aligns with other methods and previous literature, indicating that our methodology has satisfactory power while maintaining Type-I error control (which the benchmarks violate).

---

### Decision · Program_Chairs · 2023-09-21

**Decision:**

Accept (spotlight)

**Comment:**

The paper studies model-based tests of conditional independence when models may be biased (misspecified). They derive bounds on testing errors resulting from forms of misspecification in model-based CI tests and also introduce RBPT, a robust, model-based CI test. Some initial reviews highlighted the need for more empirical evaluation particularly when the conditioning set is high-dimensional; the authors during the discussion phase provided an updated empirical evaluation studying the effect of the dimensionality of the conditioning variable, the robustness of the test and the performance of baselines on real data.

This is a well written paper that tackles an important problem that is often brushed under the rug. Please do include the additional results in the pdf presented and incorporate a discussion of the same into the main paper/supplement. With that, I think this will be a very strong addition to the conference proceedings.